# Towards Minimax Optimal Reinforcement Learning in Factored Markov Decision Processes

**Yi Tian**[*]
Department of EECS
MIT
Cambridge, MA 02139
yitian@mit.edu

**Jian Qian**[*]
Department of EECS
MIT
Cambridge, MA 02139
jianqian@mit.edu

**Suvrit Sra**
Department of EECS
MIT
Cambridge, MA 02139
suvrit@mit.edu

## Abstract

We study minimax optimal reinforcement learning in episodic factored Markov decision processes (FMDPs), which are MDPs with conditionally independent transition components. Assuming the factorization is known, we propose two model-based algorithms. The first one achieves minimax optimal regret guarantees for a rich class of factored structures, while the second one enjoys better computational complexity with a slightly worse regret. A key new ingredient of our algorithms is the design of a bonus term to guide exploration. We complement our algorithms by presenting several structure-dependent lower bounds on regret for FMDPs that reveal the difficulty hiding in the intricacy of the structures.

## 1 Introduction

In reinforcement learning (RL) an agent interacts with an unknown environment seeking to maximize its cumulative reward. The dynamics of the environment and the agent's interaction with it are typically modeled as a Markov decision process (MDP). We consider the specific setting of *episodic MDPs* with a fixed interaction horizon. Here, at each step the agent observes the current state of the environment, takes an action, and receives a reward. Given the agent's action, the environment then transitions to the next state. The interaction horizon is the number of steps in an episode. Both the transitions and rewards may be unknown to the agent.

The agent's performance can be quantified using *regret*: the gap between the expected cumulative rewards the agent receives and those obtainable by following an optimal policy. An optimal RL algorithm is then one that incurs the minimum regret. For episodic MDPs, the minimax regret bound is known to be $\tilde{\mathcal{O}}(\sqrt{HSAT})$ [1] (the notation $\tilde{\mathcal{O}}(\cdot)$ hides log factors), where $H, S, A, T$ denote the horizon, the size of the state space, the size of the action space, and the total number of steps, respectively.

**Background and our focus.** In problems with large state and action spaces even the $\tilde{\mathcal{O}}(\sqrt{SA})$ dependence of the regret is impractical. We focus on problems where one can circumvent this dependence. Specifically, we focus on problems with conditional independence structure that can be modeled via *factored MDPs* (FMDPs) [2, 3]. The state for an FMDP is represented as a tuple, where each component is determined by a potentially small portion of the state-action space, termed its "scope". For example, for a home robot, whether the table is clean has nothing to do with whether the robot vacuums the floor. FMDPs also arise naturally from cooperative RL [15, 32] and constrained RL [7]; furthermore, they are useful to model subtask dependencies in hierarchical RL [36].

A key benefit of FMDPs is their more compact representation that requires exponentially smaller space to store state transitions than corresponding nonfactored MDPs. Early works [16, 24, 37, 39] show that FMDPs can also reduce the sample complexity exponentially, albeit with some higher order terms. Osband and Van Roy [28] and Xu and Tewari [43] develop near-optimal regret bounds for

---

[*]These authors contributed equally to this work.

FMDPs in the episodic and nonepisodic settings, respectively. For an FMDP with $m$ conditionally independent components, both works bound the regret by $\tilde{\mathcal{O}}(\sum_{i=1}^{m} \sqrt{D^2 S_i X[I_i]T})$, [2] where $D$ is the diameter of the MDP [20] and $X[I_i]$ is the size of the scope of the $i$th state component.

**Main difficulty.** For (nonfactored) MDPs, the above regret bounds from [28, 43] reduce to $\tilde{\mathcal{O}}(\sqrt{D^2 S^2 AT})$, also translated to $\tilde{\mathcal{O}}(\sqrt{H^2 S^2 AT})$ in the episodic setting. Either bound has a gap to the corresponding lower bound, leading one to wonder whether we can adapt the techniques from MDPs to obtain tight bounds for FMDPs. Answering this question turns out to be nontrivial.

Indeed, the key technique that yields a minimax regret bound for MDPs is a careful design of the bonus term that ensures optimism by applying scalar concentration instead of $L_1$-norm concentration (Lemma 36) on a vector [1]. To adapt the same idea to FMDPs, perhaps the most natural choice is to estimate the transition of each component separately, and then to sum up the component-wise bonuses to guide overall exploration. With the bonuses derived from $L_1$-norm concentration, this method ensures optimism, leading to the regret bound in [28, 43]. But with the bonuses derived from scalar concentration (which was key to [1]), it is hard, if not impossible, to show that this naive sum of component-wise bonuses ensures optimism (see (5.1)), and thus is the major obstacle towards removing the suboptimal $\sqrt{S_i}$ factor from the regret bound.

On the other hand, there is no uniform nondegenerate lower bound that holds for every structure. As a result, the dependence on structure has to be specified, making the lower bound statements subtle due to the intricacy of the structure involved.

**Our contributions.** Our main contribution is to present two provably efficient algorithms for episodic FMDPs, including one with tight regret guarantees for a rich class of structures. We overcome the above noted difficulty by introducing an additional bonus term via an inverse telescoping technique. Our first algorithm F-UCBVI, uses Hoeffding-style bonuses and achieves the $\tilde{\mathcal{O}}(\sum_{i=1}^{m} \sqrt{H^2 X[I_i]T})$ regret (Theorem 1). Our second algorithm F-EULER, uses Bernstein-style bonuses and achieves a problem-dependent regret (Theorem 2) that reduces to $\tilde{\mathcal{O}}(\sum_{i=1}^{m} \sqrt{H X[I_i]T})$ in the worst case (Corollary 3), and thus closes the gap to the $\Omega(\sqrt{HSAT})$ lower bound (see, e.g., [21]) when reduced to the nonfactored case. Despite an extra $\sqrt{H}$ dependence, F-UCBVI has a much simpler form of bonus and enjoys a lower computational complexity.

Furthermore, we identify some reasonably general factored structures for which we prove lower bounds (Propositions 4, 5 and Theorems 6, 7). Our construction builds upon the lower bounds for multi-armed bandits (MABs) [26], in a similar manner to [8, 21]. As a side remark, the JAO MDP, used to establish the $\Omega(\sqrt{DSAT})$ lower bound for nonepisodic MDPs [20], only establishes a $\Omega(\sqrt{HSAT/\log T})$ lower bound through a direct extension (Appendix D). This result misses a log factor resulting from the episodic resetting, and it is not clear whether it can be tightened with the same construction.

## 1.1 Related work

**Regret bound for episodic MDPs.** For episodic MDPs, using the principle of optimism in the face of uncertainty, recent works establish the $\tilde{\mathcal{O}}(\sqrt{HSAT})$ worst-case regret bound [1, 13, 23, 35, 44], matching the $\Omega(\sqrt{HSAT})$ lower bound (see, e.g., [21]) up to log factors. For model-free algorithms, the state-of-the-art worst-case regret is $\tilde{\mathcal{O}}(\sqrt{H^2 SAT})$ [45]. Sharper problem-dependent regret bounds [35, 44] also exist in this setting. Posterior sampling for RL (PSRL) [30] offers an alternative to optimism-based methods and can perform well empirically [29]. The best existing Bayesian regret bound of PSRL for episodic MDPs is $\tilde{\mathcal{O}}(\sqrt{H^2 SAT})$ [29]. Among these works, our work is mostly motivated by [1, 44]. Specifically, for nonfactored MDPs, F-UCBVI and F-EULER reduce to the UCBVI-CH [1] and EULER [44] algorithms, respectively, after which we name our algorithms.

**Planning in FMDPs.** For nonepisodic FMDPs, various works exploit the factored structure to develop efficient (approximate) dynamic programming methods [4, 14, 17, 25, 34]. See [10] for an overview. For episodic FMDPs, however, efficient approximate planning is yet to be understood. Planning is a subroutine of model-based RL algorithms. The focus of this work is the sample efficiency, and we adopt a value iteration procedure that has the same computational complexity as that for a nonfactored MDP.

**Reinforcement learning in FMDPs.** To efficiently find the optimal policy assuming known factorization yet unknown state transitions or rewards, early works aim to develop algorithms with PAC guarantees [16, 24, 37, 39]. More recent works focus on the regret analysis and establish $\tilde{\mathcal{O}}(\sum_{i=1}^{m} \sqrt{D^2 S_i X[I_i] T})$ regret in either the episodic or nonepisodic setting [28, 43], also translating to $\tilde{\mathcal{O}}(\sum_{i=1}^{m} \sqrt{H^2 S_i X[I_i] T})$ in terms of horizon $H$ for episodic problems. Our work considers minimax optimal regret guarantees in the episodic setting. It provides the first $\tilde{\mathcal{O}}(\sum_{i=1}^{m} \sqrt{H X[I_i] T})$ regret bound and the first formal treatment of lower bounds, by which we show that our algorithm is minimax optimal for a rich class of factored structures. Subsequently, Chen et al. [7] improve the upper bound to $\tilde{\mathcal{O}}(\sqrt{\sum_{i=1}^{m} H X[I_i] T})$ by a more delicate variance decomposition. Their bound is tighter than that of our F-EULER algorithm by at most $\sqrt{m}$, a log factor in the size of the state space.

**Structure learning in FMDPs.** Assuming unknown structure except for known state factorization, various structure learning algorithms exist in the literature, e.g., SPITI [11], SLF-Rmax [38], Met-Rmax [12] and LSE-Rmax [6]. See [5, Chapter 7] for an overview of these methods. More recent works include [18, 19, 33], among which Rosenberg and Mansour [33] provide the first regret analysis for nonepisodic FMDPs in this setting.

## 2 Preliminaries and problem formulation

### 2.1 Learning in episodic MDPs

An episodic MDP is described by the quintuple $(\mathcal{S}, \mathcal{A}, P, r, H)$, where $\mathcal{S}$ is the state space, $\mathcal{A}$ is the action space, $P : \mathcal{S} \times \mathcal{A} \times \mathcal{S} \to [0, 1]$ is the transition function such that $P(\cdot|s, a)$ is in the probability simplex on $\mathcal{S}$, denoted by $\Delta(\mathcal{S})$, $r : \mathcal{S} \times \mathcal{A} \to [0, 1]$ is the reward function, and $H$ is the horizon or the length of an episode. The focus in this paper is the stationary case, where both the transition function $P$ and the reward function $r$ remain unchanged across steps or episodes.

An RL agent interacts episodically with the environment starting from arbitrary initial states. The role of the agent is abstracted into a sequence of policies. For any natural number $n$, we use $[n]$ to denote the set $\{1, 2, \cdots, n\}$. Then for finite-horizon MDPs, the policy $\pi : \mathcal{S} \times [H] \to \mathcal{A}$ maps a state and a time step to an action. Let $\pi_k$ denote the policy in the $k$th episode. An RL algorithm, therefore, specifies the updates of the sequence $\{\pi_k\}$. Our algorithms use deterministic policies, while the lower bounds hold for the more general class of stochastic policies.

The Q-value function $Q_h^{\pi}$ and state-value function $V_h^{\pi}$ measure the goodness of a state-action pair and of a state following the policy $\pi$ starting from the $h$th step in an episode, respectively. For state $s \in \mathcal{S}$ and action $a \in \mathcal{A}$, these functions are defined by

$$Q_h^{\pi}(s, a) = \mathbb{E}\Big[\sum\nolimits_{t=h}^{H} r(s_t, a_t) \,\big|\, s_h = s, a_h = a\Big], \quad V_h^{\pi}(s) = Q_h^{\pi}(s, \pi(s, h)).$$

There exists an optimal policy $\pi^*$ such that $V_h^{\pi^*}(s) \geq V_h^{\pi}(s)$ for all $s \in \mathcal{S}, h \in [H]$ [31]. Let $V_h^*$ denote the optimal state-value function $V_h^{\pi^*}$, and let $s_{k,1}$ be the initial state at the $k$th episode; we then define the regret over $K$ episodes ($T = KH$ steps) as

$$\text{Regret}(K) := \sum\nolimits_{k=1}^{K} \left(V_1^*(s_{k,1}) - V_1^{\pi_k}(s_{k,1})\right).$$

### 2.2 Factored MDPs

Episodic FMDPs inherit the above definitions, but we also need additional notation that we adopt mainly from [28] to describe the factored structure. Let $\mathcal{X} := \mathcal{S} \times \mathcal{A}$ be the state-action space, so that $x = (s, a) \in \mathcal{X}$ denotes a state-action pair. Let $m, n, l$ be natural numbers. The state space is factorized as $\mathcal{S} = \mathcal{S}_1 \times \cdots \times \mathcal{S}_m$, using which we write a state $s \in \mathcal{S}$ as the tuple $(s[1], \cdots, s[m])$. Similarly, the state-action space is factorized as $\mathcal{X} = \mathcal{X}_1 \times \cdots \times \mathcal{X}_n$. Below we also use "$\bigotimes$" to denote Cartesian products, as needed.

**Definition 1** (Scope operation). *For any natural number $n$, any index set $I \subset [n]$ and any factored set $\mathcal{X} = \bigotimes_{i=1}^{n} \mathcal{X}_i$, define the scope operation $\mathcal{X}[I] := \bigotimes_{i \in I} \mathcal{X}_i$, where the indices in the Cartesian product are in ascending order by default. Correspondingly, define the scope variable $x[I] \in \mathcal{X}[I_i]$ as the tuple of $x_i$ where $i \in I$.*

The transition factored structure refers to the conditional independence of the transitions of the $m$ state components. Mathematically, $P(s'|s, a) = P(s'|x) = \prod_{i=1}^{m} P_i(s'[i]|x)$, where $P_i(\cdot|x) \in \Delta(\mathcal{S}_i)$.

Moreover, the transition of the $i$th state component is determined only by its scope, a subset of the state-action components. Let $I_i \subset [n]$ be the scope index set for $i \in [m]$. Then, with a slight abuse of notation, $P_i(s'[i]|x) \equiv P_i(s'[i]|x[I_i])$ where "$\equiv$" denotes identity, and the factored transition is given by $P(s'|x) = \prod_{i=1}^{m} P_i(s'[i]|x[I_i])$.

The reward function $r$ is stochastic and unknown in general, and can also be factored. Let $J_i \subset [n]$ be the reward scope index set for $i \in [l]$. The factored reward is the sum of $l$ independent components $r_i(x) \equiv r_i(x[J_i])$, given by $r(x) = \sum_{i=1}^{l} r_i(x[J_i])$. Let $R := \mathbb{E}[r]$ and $R_i := \mathbb{E}[r_i]$ be their expectations. Then $R = \sum_{i=1}^{l} R_i : \mathcal{X} \to [0,1]$ is deterministic. With the above notation, an FMDP model is described in general by

$$\mathcal{M} = \left( \{\mathcal{X}_i\}_{i=1}^{n}, \{\mathcal{S}_i, I_i, P_i\}_{i=1}^{m}, \{J_i, r_i\}_{i=1}^{l}, H \right). \tag{2.1}$$

In this paper, we assume unknown transition with a known factorization and consider two cases about the knowledge of reward: (1) the reward $r$ is unknown with a known factorization and the agent receives the factored rewards $\{r_i\}_{i=1}^{l}$ [28, 43]; or (2) the reward $r$ is known and not necessarily factored, in which case the expected reward $R$ is also known [1].

**Notation.** Throughout the paper, we use $S, A, X, S_i, X[I]$ to denote the finite cardinalities of the corresponding $\mathcal{S}, \mathcal{A}, \mathcal{X}, \mathcal{S}_i, \mathcal{X}[I]$, respectively. Then $S = \prod_{i=1}^{m} S_i$ and $X[I] \leq X = SA$ for any $I \subset [n]$. Since the abstraction of a state component $s[i]$ is meaningful only if $\mathcal{S}_i \geq 2$, the number of state components $m$ is at most $\log_2 S$, which is treated as a negligible log factor whenever necessary. For $x = (s,a) \in \mathcal{X}$, we use $Q(s,a)$ and $Q(x)$ interchangeably to denote a function $Q$ defined on $\mathcal{X} := \mathcal{S} \times \mathcal{A}$. A function $V : \mathcal{S} \to \mathbb{R}$ is also seen as a real vector indexed by $\mathcal{S}$, denoted by $V \in \mathbb{R}^{\mathcal{S}}$. For vectors in $\mathbb{R}^{\mathcal{S}}$, let $|\cdot|, (\cdot)^2$ denote entrywise absolute values and squares, respectively.

Define $P(x) := P(\cdot|x) \in \Delta(\mathcal{S})$. Moreover, for $P_i(x) \equiv P_i(x[I_i]) := P_i(\cdot|x[I_i]) \in \Delta(\mathcal{S}_i)$ and natural numbers $a \leq b$, define $P_{a:b}(x) := \prod_{i=a}^{b} P_i(x) \in \Delta(\bigotimes_{i=a}^{b} \mathcal{S}_i)$. In particular, $\prod_{i=1}^{m} P_i(x) = P(x) \in \Delta(\mathcal{S})$. For $P \in \Delta(\mathcal{S})$, $V \in \mathbb{R}^{\mathcal{S}}$, define their inner product $\langle P, V \rangle := \sum_{s \in \mathcal{S}} P(s)V(s) = \mathbb{E}_P[V(s)] \equiv \mathbb{E}_P[V]$. The inner product of two elements in $\Delta(\mathcal{S}_i)$ and $\mathbb{R}^{\mathcal{S}_i}$ respectively takes sum over $\mathcal{S}_i$. For $P_i \in \Delta(\mathcal{S}_i)$ and $V \in \mathbb{R}^{\mathcal{S}}$, their inner product takes sum over $\mathcal{S}$. In this case, note that $\langle P_i, V \rangle \neq \mathbb{E}_{P_i}[V]$, which is given by $\mathbb{E}_{P_i}[V] \equiv \mathbb{E}_{P_i}[V(s)] = \sum_{s[i] \in \mathcal{S}} P_i(s[i])V(s) \in \mathbb{R}^{\mathcal{S}[-i]}$, where $\mathcal{S}[-i] := \bigotimes_{j=1, j\neq i}^{m} \mathcal{S}_j$.

## 3  Factored UCBVI and regret bounds

We are now ready to present the two algorithms for solving FMDPs that we propose. In this section, We introduce the general procedures (Algorithms 1 and 2) and discuss an instantiation called factored UCBVI (F-UCBVI) using Hoeffding-style optimism. In the next section, we use a Bernstein-style instantiation called factored EULER (F-EULER) that obtains better regret guarantees. The general optimistic model-based RL framework for FMDPs is described in F-OVI (Algorithm 1), which maintains an empirical estimate of the transition function, typically the maximum likelihood (ML) estimate. For FMDPs, although a direct ML estimate of the overall transition is still possible, the product of the ML estimates of component-wise transition functions is preferable to exploit the factored structure. The same argument applies to the empirical expected reward function.

As a subroutine of F-OVI (Algorithm 1), VI_Optimism (Algorithm 2) ensures an upper confidence bound (UCB) on the optimal state-value function, achieved by introducing bonuses derived from concentration inequalities. The $L_1$-norm concentration (Lemma 36) on the transition estimation leads to the UCRL-Factored algorithm, achieving the $\tilde{\mathcal{O}}(\sum_{i=1} \sqrt{D^2 S_i X[I_i]T})$ regret [28]. Scalar concentration on the transition estimation leads to the two new algorithms proposed in this work.

In the case of known rewards, there is no need to estimate the reward $\hat{R}$ or to collect the reward history $\mathcal{H}_r$ in F-OVI (Algorithm 1). Neither is there the need to introduce the reward bonus $\beta$ in VI_Optimism (Algorithm 2). In both procedures, the empirical $\hat{R}$ is replaced by the true $R$.

**Hoeffding-style bonus.** Let $L = \log(16mlSXT/\delta)$ be a log factor. With a slight abuse of notation, let $N_i(x) \equiv N_i(x[I_i])$. The transition bonus of F-UCBVI is given by

$$b(x) := \sum_{i=1}^{m} H\sqrt{\tfrac{L}{2N_i(x)}} + \sum_{i=1}^{m}\sum_{j=i+1}^{m} 2HL\sqrt{\tfrac{S_i S_j}{N_i(x)N_j(x)}}, \tag{3.1}$$

---

**Algorithm 1** F-OVI (Factored Optimistic Value Iteration)

---

**Require:** Factored structure $\{\mathcal{X}_i\}_{i=1}^n$, $\{\mathcal{S}_i, I_i\}_{i=1}^m$, $\{J_i\}_{i=1}^l$, and horizon $H$
1: Initialize counters $N_i(x[I_i]) = N_i'(x[I_i], s[i]) = 0, \forall (i \in [m], x[I_i] \in \mathcal{X}[I_i], s[i] \in \mathcal{S}_i)$
2: Initialize counters $M_i(x[J_i]) = M_i'(x[J_i]) = 0, \forall (i \in [l], x[J_i] \in \mathcal{X}[J_i])$ and history $\mathcal{H}_r = \emptyset$
3: **for** episode $k = 1, 2, \cdots, K$ **do**
4:     Estimate transitions $\hat{P}_i(s[i]|x[I_i]) = \frac{N_i'(x[I_i], s[i])}{\max\{1, N_i(x[I_i])\}}, \forall (i \in [m], x[I_i] \in \mathcal{X}[I_i], s[i] \in \mathcal{S}_i)$
5:     Estimate expected rewards $\hat{R}_i(x[J_i]) = \frac{M_i'(x[J_i])}{\max\{1, M_i(x[J_i])\}}, \forall (i \in [l], x[J_i] \in \mathcal{X}[J_i])$
6:     Estimate expected overall reward $\hat{R}(x) = \sum_{i=1}^l \hat{R}_i(x[J_i]), \forall x \in \mathcal{X}$
7:     Obtain policy $\pi = $ VI_Optimism$(\{\hat{P}_i\}_{i=1}^m, \hat{R}, \{N_i\}_{i=1}^m, \{M_i\}_{i=1}^l, \mathcal{H}_r)$
8:     Observe initial state $s_1$ from environment
9:     **for** step $h = 1, 2, \cdots, H$ **do**
10:         Take action $a_h = \pi(s_h, h)$ and let $x_h = (s_h, a_h)$
11:         Receive reward $\{r_{i,h}\}_{i=1}^l$ and observe next state $s_{h+1}$
12:         Update transition counters $N_i(x_h[I_i]) \mathrel{+}= 1$ and $N_i'(x_h[I_i], s_{h+1}[i]) \mathrel{+}= 1, \forall i \in [m]$
13:         Update reward counters $M_i(x_h[J_i]) \mathrel{+}= 1$ and $M_i'(x_h[J_i]) \mathrel{+}= r_{i,h}, \forall i \in [l]$
14:         Update reward history $\mathcal{H}_r = \mathcal{H}_r \cup \{(x_h, \{r_{i,h}\}_{i=1}^l)\}$
15:     **end for**
16: **end for**

---

**Algorithm 2** VI_Optimism (Value Iteration with Optimism)

---

**Require:** Empirical transitions $\{\hat{P}_i\}_{i=1}^m$, empirical expected reward $\hat{R}$, transition counters $\{N_i\}_{i=1}^m$, reward counters $\{M_i\}_{i=1}^l$, reward history $\mathcal{H}_r$
1: Initialize UCB and LCB $\overline{V}_{H+1}(s) = \underline{V}_{H+1}(s) = 0, \forall s \in \mathcal{S}$; let $\hat{P}(x) = \prod_{i=1}^m \hat{P}_i(x), \forall x \in \mathcal{X}$
2: **for** step $h = H, H-1, \cdots, 1$ **do**
3:     **for** state $s \in \mathcal{S}$ **do**
4:         **for** action $a \in \mathcal{A}$, with $x = (s, a)$ **do**
5:             $\overline{Q}_h(x) = \min\left\{H - h + 1, \hat{R}(x) + \left\langle \hat{P}(x), \overline{V}_{h+1} \right\rangle + b(x) + \beta(x)\right\}$
6:         **end for**
7:         Let $\pi(s, h) = \operatorname{argmax}_{a \in \mathcal{A}} \overline{Q}_h(s, a)$ and $x = (s, \pi(s, h))$
8:         Update UCB $\overline{V}_h(s) = \overline{Q}_h(x)$
9:         Update LCB $\underline{V}_h(s) = \max\left\{0, \hat{R}(x) + \left\langle \hat{P}(x), \underline{V}_{h+1} \right\rangle - b(x) - \beta(x)\right\}$
10:     **end for**
11: **end for**
12: **return** policy $\pi$

---

which consists of both a component-wise bonus term and a cross-component bonus term. Intuitively, since the transition and reward factorizations in an FMDP may differ arbitrarily, the sum of component-wise transition bonuses alone may not suffice to ensure optimism. It turns out that combining the cross-component transition bonuses suffices. In the case of unknown rewards, the reward bonus of F-UCBVI is given by

$$\beta(x) := \sum_{i=1}^l \sqrt{\frac{L}{2M_i(x)}}. \tag{3.2}$$

For the transition and reward bonuses here and below, zero-valued $N_i(x)$ and $M_i(x)$ in the denominators are replaced by 1 to prevent zero division, as in F-OVI (Algorithm 1). For F-UCBVI, the reward history $\mathcal{H}_r$ and the lower confidence bound (LCB) $\underline{V}_h$ are redundant and the related updates are removable. F-UCBVI has the following regret guarantees.

**Theorem 1** (Worst-case regret bounds of F-UCBVI). *For an FMDP specified by* (2.1), *in the case of unknown rewards, with probability $1 - \delta$, the regret of F-UCBVI in $K$ episodes is upper bounded by*

$$\tilde{\mathcal{O}}\left(\sum_{i=1}^m \sqrt{H^2 X[I_i] T} + \sum_{i=1}^l \sqrt{X[J_i] T}\right). \tag{3.3}$$

*In the case of known rewards, the term $\sum_{i=1}^l \sqrt{X[J_i] T}$ can be dropped.*

In Theorem 1 (and Theorem 2, Corollary 3 below), the $\sqrt{T}$ terms dominate the upper bounds under the assumption that $T \geq \text{poly}(m, l, \max_i S_i, \max_i X[I_i], H)$, which can be exponentially smaller than $\text{poly}(S, A, H)$, required in general by nonfactored algorithms [1, 13, 22, 23, 35, 44, 45]. This weaker requirement on $T$ is also true for [28] and can be viewed as another benefit from the factored structure. The lower-order terms in Section 5 are also under this assumption.

In (3.3), the regret bound consists of a transition-induced term and a reward-induced term, which correspond to the sum over time of the component-wise transition bonus term in (3.1) and the reward bonus (3.2), respectively. The cross-component transition bonus term in (3.1) diminishes fast and its sum over time turns out to be negligible. Nevertheless, this bonus term plays a significant role by stablizing early-stage exploration, when it dominates the transition bonus due to the dependence on $S_i$. Concretely, initially, the cross-component transition bonus term encourages exploration of the $i$th and $j$th components with large $S_i S_j$. Then as the counters of some components (say, the $i$th) grow faster such that $N_i(x) \geq S_i$, its cross-component transition bonuses (say, with the $j$th) satisfy $HL\sqrt{S_i/N_i(x) \cdot S_j/N_j(x)} \leq HL\sqrt{S_j/N_j(x)}$, which prioritize exploration of underexplored components until all counters satisfy $N_i(x) \geq S_i$. In the next stage this bonus term balances the exploration among different components until all counters satisfy $\max\{N_i(x), N_j(x)\} \geq S_i S_j$ and the component-wise transition bonus term takes over to dominate the overall bonus.

Note that the cross-component form is not absolutely necessary, e.g., by the AM-GM inequality,

$$\sum_{i=1}^{m} \sum_{j=i+1}^{m} 2HL\sqrt{\tfrac{S_i S_j}{N_{i,k}(x) N_{j,k}(x)}} \leq (m-1)HL \sum_{i=1}^{m} \tfrac{S_i}{N_{i,k}(x)}.$$

In this work, we use the cross-component form, as it is tighter and implies an intertwined behavior among the state components.

The component-wise transition bonus term in (3.1) is exactly the bonus in the UCBVI-CH algorithm [1]. It may also be feasible to derive a factored version of UCBVI-BF [1], but we switch to an upper-lower bound exploration strategy like EULER [44] for the Bernstein-style optimism, due in part to its problem-dependent regret.

## 4   Factored EULER and regret bounds

To describe the Bernstein-style bonus, we first add some convenient notation. We omit the dependence of $P_i(x)$ and $P(x)$ on $x$; for $P_i \in \Delta(\mathcal{S}_i)$, $P = \prod_{i=1}^{m} P_i \in \Delta(\mathcal{S})$ and $V \in \mathbb{R}^{\mathcal{S}}$, define

$$g_i(P, V) := 2\sqrt{L}\sqrt{\text{Var}_{P_i} \mathbb{E}_{P_{-i}}[V]}, \tag{4.1}$$

where $\mathbb{E}_{P_{-i}}[V] := \mathbb{E}_{P_{1:i-1}P_{i+1:m}}[V] \in \mathbb{R}^{\mathcal{S}_i}$ takes expectation over the $m-1$ components. With the notation from [44], let $\|X\|_{2,P} = \sqrt{\mathbb{E}_P[X^2]}$ be the $L_2$-norm on the equivalence classes of almost surely the same random variables.

**Bernstein-style bonus.** The transition bonus of F-EULER is defined by

$$b(x) := \sum_{i=1}^{m} \frac{g_i(\hat{P}(x), \overline{V}_{h+1})}{\sqrt{N_i(x)}} + \sum_{i=1}^{m} \frac{\sqrt{2L}\|\overline{V}_{h+1} - \underline{V}_{h+1}\|_{2,\hat{P}(x)}}{\sqrt{N_i(x)}}$$
$$+ \sum_{i=1}^{m} \sum_{j=i+1}^{m} 11HL\sqrt{\tfrac{S_i S_j}{N_i(x) N_j(x)}} + \sum_{i=1}^{m} \tfrac{5HL}{N_i(x)}. \tag{4.2}$$

In the case of unknown rewards, the reward bonus of F-EULER is defined by

$$\beta(x) := \sum_{i=1}^{l} \sqrt{\tfrac{4\mathbb{S}[\hat{r}_i(x)]L}{M_i(x)}} + \sum_{i=1}^{l} \tfrac{14L}{3M_i(x)}, \tag{4.3}$$

where $\mathbb{S}[\hat{r}_i(x)]$ is sample variance of the reward component, calculated using the history $\mathcal{H}_r$. Online algorithms like Welford's method [42] can efficiently compute $\hat{R}_i$ and $\mathbb{S}[\hat{r}_i(x)]$ jointly. F-EULER has the following problem-dependent regret guarantees.

**Theorem 2** (Problem-dependent regret bounds of F-EULER). *For an FMDP specified by* (2.1), *let*

$$\mathcal{Q}_i := \max_{x \in \mathcal{X}, h \in H} \left\{ \text{Var}_{P_i(x)} \mathbb{E}_{P_{-i}(x)}[V_{h+1}^*] \right\}, \quad \mathcal{R}_i := \max_{x \in \mathcal{X}} \left\{ \text{Var}[r_i(x)] \right\}.$$

*Let $\mathcal{G}$ be the upper bound of $\sum_{h=1}^{H} R(x_h)$ for any initial state and any policy. In the case of unknown rewards, with probability $1 - \delta$, the regret of F-EULER in $K$ episodes is upper bounded by*

$$\min\{\tilde{\mathcal{O}}(\sum_{i=1}^{m} \sqrt{\mathcal{Q}_i X[I_i]T}), \tilde{\mathcal{O}}(\sum_{i=1}^{m} \sqrt{\mathcal{G}^2 X[I_i]K})\} \tag{4.4}$$

$$+ \min\{\tilde{\mathcal{O}}(\sum_{i=1}^{l} \sqrt{\mathcal{R}_i X[J_i]T}), \tilde{\mathcal{O}}(\sum_{i=1}^{l} \sqrt{\mathcal{G}^2 X[J_i]K})\}. \tag{4.5}$$

*In the case of known rewards, the term* (4.5) *is dropped.*

F-EULER does not assume knowledge of the actual $\mathcal{Q}_i, \mathcal{R}_i$ and $\mathcal{G}$ and its regret bounds (Theorem 2) enjoy the same problem-dependent advantages as discussed in the nonfactored case [44]. $r(\cdot) \in [0, 1]$ yields $\mathcal{G} \leq H$ and $\mathcal{R}_i \leq 1$ for all $i \in [l]$. Hence, Theorem 2 implies the following worst-case regret.

**Corollary 3** (Worst-case regret bounds of F-EULER)**.** *For an FMDP specified by* (2.1)*, if the rewards are unknown, with probability* $1 - \delta$*, the regret of F-EULER in $K$ episodes is upper bounded by*

$$\tilde{\mathcal{O}}\left(\sum_{i=1}^{m} \sqrt{HX[I_i]T} + \sum_{i=1}^{l} \sqrt{X[J_i]T}\right). \tag{4.6}$$

*It the rewards are known, the term* $\sum_{i=1}^{l} \sqrt{X[J_i]T}$ *is dropped.*

In (4.6), the regret bound again consists of a transition-induced term and a reward-induced term, resulting from the sum over time of the first term in (4.2) and the first term in (4.3), respectively. For nonfactored MDPs, the reward-induced term $\tilde{\mathcal{O}}(\sqrt{SAT})$ is absorbed into the transition-induced term $\tilde{\mathcal{O}}(\sqrt{HSAT})$, while for FMDPs, these two terms depend on their factorization structures, and do not absorb each other in general. Corollary 3 indicates that F-EULER improves the transition-induced term of F-UCBVI by a $\sqrt{H}$ factor. On the other hand, F-EULER has a worse computational complexity than F-UCBVI. Even if neglecting the computation on the reward bonus (since efficient methods exist), in each run of the VI_Optimism subroutine, F-EULER has $\mathcal{O}(mS^2AH)$ computational complexity, $m$ times more than F-UCBVI.

## 5 Proof sketch

We present here the proof ideas of F-UCBVI in the case of known rewards, which sheds light on our main new techniques. The full proof is deferred to Appendix A. We add a subcript $k$ to $\hat{P}_i, \hat{P}, N_i, M_i, \overline{V}_h, b$ in Algorithms 1 and 2 to denote the corresponding quantities in the $k$th episode. For inductive purpose, let $V_{H+1}^\pi(s) = 0$ for all $\pi \in \Pi, s \in \mathcal{S}$, so that $V_h^\pi(s) = R(x) + \left\langle P(x), V_{h+1}^\pi\right\rangle$ with $x = (s, \pi(s, h))$ for all $\pi \in \Pi, s \in \mathcal{S}, h \in [H]$, where $\Pi$ is the set of all deterministic policies.

**The mechanism of optimism.** The purpose of the bonus is to ensure optimism, which means obtaining an entrywise UCB $\overline{V}_{k,h}$ on $V_h^*$ for all $k \in [K], h \in [H]$.

For $h = H + 1$, this is certainly the case, since they are both entrywise 0. For $h \in [H], \forall s \in \mathcal{S}$, with $x_h^* = (s, \pi^*(s, h))$,

$$\overline{V}_{k,h}(s) - V_h^*(s) \geq \underbrace{\left\langle \hat{P}_k(x_h^*), V_{k,h+1} - V_{h+1}^*\right\rangle}_{\geq\ 0\text{ by inductive assumption}} + \underbrace{\left\langle \hat{P}_k(x_h^*) - P(x_h^*), V_{h+1}^*\right\rangle}_{\text{transition estimation error}} + b_k(x_h^*).$$

To form a valid backward induction argument, we expect the bonus to be a UCB of the absolute value of the transition estimation error. For nonfactored MDPs, with $N_k(x)$ being the number of visits to $x$ before the $k$th episode, a direct application of Hoeffding's inequality (Lemma 37) yields that for any $x \in \mathcal{X}$,

$$\left|\left\langle \hat{P}_k(x) - P(x), V_{h+1}^*\right\rangle\right| \leq H\sqrt{\frac{L}{2N_k(x)}},$$

which determines the transition bonus. For FMDPs, a natural extension is to expect

$$\left|\left\langle \hat{P}_k(x) - P(x), V_{h+1}^*\right\rangle\right| \stackrel{?}{\leq} \sum_{i=1}^{m}\left|\left\langle \hat{P}_{i,k}(x) - P_i(x), V_{h+1}^*\right\rangle\right| \leq \sum_{i=1}^{m} H\sqrt{\frac{L}{2N_{i,k}(x)}}, \tag{5.1}$$

which can then be used as the transition bonus. Although it holds that

$$\left|\left\langle \hat{P}_k(x) - P(x), V_{h+1}^*\right\rangle\right| \leq \sum_{i=1}^{m}\left\langle \left|\hat{P}_{i,k}(x) - P_i(x)\right|, V_{h+1}^*\right\rangle,$$

the inequality in question in (5.1) fails to hold (e.g., using $S_i = m = 2$ one obtains a quick contradiction). To address this difficulty, we adopt an inverse telescoping technique. Omitting the dependence of the transitions $\hat{P}_k(x), P(x), \hat{P}_i(x), P_i(x)$ on $x$, we rewrite $\left\langle \hat{P}_k - P, V_{h+1}^*\right\rangle$ as

$$\left\langle \prod_{i=1}^{m} \hat{P}_{i,k} - \prod_{i=1}^{m} P_i, V_{h+1}^*\right\rangle = \left\langle \sum_{i=1}^{m}(\hat{P}_{i,k} - P_i)P_{1:i-1}\hat{P}_{i+1:m,k}, V_{h+1}^*\right\rangle$$

$$= \sum_{i=1}^{m}\left\langle \hat{P}_{i,k} - P_i, \mathbb{E}_{P_{1:i-1}}\mathbb{E}_{P_{i+1:m}}[V_{h+1}^*]\right\rangle \tag{5.2}$$

$$+ \sum_{i=1}^{m}\left\langle \hat{P}_{i,k} - P_i, \mathbb{E}_{P_{1:i-1}}(\mathbb{E}_{\hat{P}_{i+1:m,k}} - \mathbb{E}_{P_{i+1:m}})[V_{h+1}^*]\right\rangle, \tag{5.3}$$

where the first equality is referred to as the "inverse telescoping" technique, and the second equality adds then subtracts the same term (5.2), which can be bounded by scalar concentration, leading to the component-wise bonus term in (3.1). (5.3) is bounded by $L_1$-norm concentration (Lemma 36) with another use of the inverse telescoping technique, leading to the cross-component bonus term in (3.1).

**Regret bound.** In the sequel, we use "$\lesssim, \approx$" to represent "$\leq, =$" hiding constants and lower-order terms. Let $w_{k,h}(x)$ denote the visit probability of $x \in \mathcal{X}$ at step $h \in [H]$ following policy $\pi_k$ during the interaction. Then the optimism ensures that the regret in $K$ episodes is upper bounded by

$$\sum_{k=1}^{K} \overline{V}_{k,1}(s_{k,1}) - V_1^{\pi_k}(s_{k,1})$$

$$\approx \sum_{k=1}^{K} \sum_{h=1}^{H} \sum_{x \in L_k} w_{k,h}(x) \Big\{ b_k(x) + \underbrace{\langle \hat{P}_k(x) - P(x), V_{h+1}^* \rangle}_{\text{transition estimation error}} + \underbrace{\langle \hat{P}_k(x) - P(x), \overline{V}_{k,h+1} - V_{h+1}^* \rangle}_{\text{correction, sum is lower-order (Lemma 22)}} \Big\},$$

where the "good" set $L_k$ is a notion of sufficient visits before the $k$th episode and the sum out of $L_k$ is a lower-order term (Lemma 13). The cumulative transition estimation error is upper bounded by

$$\sum_{k=1}^{K} \sum_{h=1}^{H} \sum_{x \in L_k} w_{k,h}(x) \Big\{ \sum_{i=1}^{m} H \sqrt{\frac{L}{2N_{i,k}(x)}} + \underbrace{\sum_{i=1}^{m} \sum_{j=i+1}^{m} 2HL \sqrt{\frac{S_i S_j}{N_{i,k}(x) N_{j,k}(x)}}}_{\text{sum is lower-order (Lemma 16)}} \Big\} \lesssim \sum_{i=1}^{m} H \sqrt{X[I_i]TL}.$$

The upper bound of the cumulative transition bonus takes exactly the same form. Thus, we obtain

$$\text{Regret}(K) \lesssim \sum_{i=1}^{m} H \sqrt{X[I_i]TL} = \tilde{\mathcal{O}}\Big( \sum_{i=1}^{m} \sqrt{H^2 X[I_i]T} \Big).$$

**Extension to other conditions.** The treatment of unknown rewards is standard [44]. The same mechanism of optimism applies to F-EULER, except that the scalar concentration uses Bernstein's inequality (Lemma 38). Please see Appendix B for a complete proof.

# 6 Lower bounds

In this section, we present some information theoretic lower bounds for different factored structures of FMDPs, which illustrate the structure-dependent nature of the lower bounds. Even in the most general lower bound (Theorem 7), we assume a normal factored structure (Definition 2); a full characterization is still an open problem.

Recall the formal statement of the lower bound for MDPs that for any given natural numbers $S, A, H$, there is an MDP that has $S$ states, $A$ actions and horizon $H$ with unknown transition $P$ and possibly known rewards $R$, such that the expected regret of any algorithm in $K$ episodes is $\Omega(\sqrt{HSAT})$. Ideally, the lower bound of FMDPs should be stated in the same way, specified for any natural numbers $m, n, l, H$ and any factored structure $\{\mathcal{S}_i\}_{i=1}^{m}, \{\mathcal{X}_i\}_{i=1}^{n}, \{I_i\}_{i=1}^{m}, \{J_i\}_{i=1}^{l}$. A simple $\Omega(\sum_{i=1}^{m} \sqrt{HX[I_i]T} + \sum_{i=1}^{l} \sqrt{X[J_i]T})$ lower bound is tempting, yet far from the truth. To avoid excessive subtlety, our lower bound discussion is restricted to the following normal factored structure, where the state-action factorization is an extension to the state factorization.

**Definition 2** (Normal factored structure). *For natural numbers $m < n$, let the state space $\mathcal{S} = \bigotimes_{i=1}^{m} \mathcal{S}_i$ and the action space $\mathcal{A} = \bigotimes_{i=1}^{n-m} \mathcal{A}_i$. The factored structure of an FMDP is normal if the state-action space $\mathcal{X} = \bigotimes_{i=1}^{n} \mathcal{X}_i$ where $\mathcal{X}_i = \mathcal{S}_i$ for $i \in [m]$ and $\mathcal{X}_{m+i} = \mathcal{A}_i$ for $i \in [n-m]$.*

We now state the lower bounds for two degenerate structures.

**Proposition 4** (Degenerate case 1). *For any algorithm, under the assumption of unknown rewards, for the normal factored structure that satisfies $I_i \subset [m]$ for all $i \in [m]$ and any $J_i$ for $i \in [l]$, there is an FMDP with the specified structure such that for some initial states, the expected regret in $K$ episodes is at least $\Omega(\max_i \sqrt{X[J_i']T})$ where $J_i' = J_i \cap \{m+1, \cdots, n\}$.*

**Proposition 5** (Degenerate case 2). *For any algorithm, under the assumption of unknown rewards, for the normal factored structure that satisfies $J_i \subset \{m+1, \cdots, n\}$ for all $i \in [l]$, there is an FMDP with the specified structure such that for some initial states, the expected regret in $K$ episodes is at least $\Omega(\max_i \sqrt{X[J_i]T})$.*

Proposition 4 considers the degenerate case where the transitions do not depend on the action space at all. Furthermore, the regret is always 0 if the rewards do not depend on the action space either ($J_i' = \emptyset$ for all $i \in [l]$). Proposition 5 considers the degenerate case where the rewards have no dependence on the state space, and $X[I_i]$ vanishes in the lower bound for this MAB problem. The following theorem identifies some nondegenerate constraints on the factored structures.

**Theorem 6** (Lower bound, a nondegenerate case). *For any algorithm, under the assumption of known rewards, for the normal factored structure that satisfies $S_i \geq 3$, $i \in I_i$, $\exists j \in \{m+1, \cdots, n\}$ such that $j \in I_i$ for all $i \in [m]$ and $H \geq 2$, there is an FMDP with the specified structure such that for some initial states, the expected regret in $K$ episodes is at least $\Omega(\max_i \sqrt{HX[I_i]T})$.*

The regret achieved by F-EULER is minimax optimal up to log factors for the structures specified in Theorem 6. The main assumption here is that the transition of each state component depends on itself and at least one action component. A major obstacle to show the lower bound for a general factored structure is that some structures lack the self-loop property, i.e., the transition of a state component depends on itself so that we can say it *stays* in the same state with a certain probability, which is key to the lower bound constructions for nonfactored MDPs [8, 20, 21]. By generalizing the self-loop property to the loop property, we can show a more general version of the nondegenerate lower bound.

For an FMDP with the normal factored structure, for $i, j \in [m]$, if $i \in I_j$, then we say the $j$th state component *depends* on the $i$th state component, or the $i$th state component *influences* the $j$th state component, denoted by $\mathcal{S}_i \rightarrow \mathcal{S}_j$. We say the $i$th state component has the *loop* property if the $i$th state component influences itself either directly ($\mathcal{S}_i \rightarrow \mathcal{S}_i$, self-loop) or through some intermediate state components ($\mathcal{S}_i \rightarrow \cdots \rightarrow \mathcal{S}_j \rightarrow \cdots \rightarrow \mathcal{S}_i$ for some $j$'s $\in [m]$). The loop is referred to as an *influence loop*. Let $\mathcal{I}$ be the set of the indices of the state components that have the loop property and some action dependence, i.e.,

$$\mathcal{I} := \{i \in [m] : \mathcal{S}_i \text{ has the loop property, and there exists } j \in \{m+1, \cdots, n\} \text{ such that } j \in I_i\}.$$

Then we have the following theorem that subsumes Theorem 6.

**Theorem 7** (Lower bound, normal factored structure with known rewards). *For any algorithm, under the assumption of known rewards, for the normal factored structure that satisfies $S_i \geq 3, \forall i \in [m]$ and $H \geq 2$, there is an FMDP with the specified structure such that for some initial states, the expected regret in $K$ episodes is at least $\Omega(\max_{i \in \mathcal{I}} \sqrt{HX[I_i]T})$.*

Theorem 7 indicates that for a rich class of factored structures satisfying $\max_{i \in \mathcal{I}} X[I_i] = \max_{i \in [m]} X[I_i]$, F-EULER is minimax optimal up to log factors. We defer to Appendix C the proofs of Propositions 4, 5 and Theorems 6, 7, relying on the lower bounds for MABs [26] and the construction of MAB-like FMDPs (FMDPs whose cumulative rewards depend only on the action in the first step), similar to the MAB-like MDPs in [8, 21].

# 7 Conclusion

In this paper, we study reinforcement learning in the tabular episodic setting of FMDPs, that is, MDPs which enjoy a factored structure in their dynamics. Such a factorization is typically derived via conditional independence structures in the transition, and by using it effectively one can greatly reduce the complexity of learning. However, a straightforward adaptation of the usual minimax optimal regret analysis for MDPs turns out not to be possible. We uncover the difficulties posed by such an adaptation, and subsequently develop two new algorithms (called F-UCBVI and F-EULER, both motivated by their known nonfactored analogues) that carefully exploit the factored structure to obtain minimax optimal regret bounds for a rich class of factored structures. The key algorithmic technique that we develop is a careful design of a "cross-component" bonus term to ensure optimism and guide exploration.

We present the lower bounds for FMDPs under certain structure constraints. The problem of characterizing lower bounds for FMDPs with arbitrary structures turns out to be more subtle, and remains an open problem. In addition, our methods are model-based, and the computational acceleration of planning in episodic FMDPs is a practical direction; on the other hand, developing efficient model-free algorithms for FMDPs is also worth exploring.

## Broader impact

This work focuses on a theoretical problem about efficient learning in factored MDPs. The work itself is ethically neutral. Future societal consequences heavily depend on the specific domain of application. For example, applications to home robots can potentially alleviate the housework in a middle-class family and contribute to the harmony therein.

## Acknowledgments and Disclosure of Funding

YT acknowledges partial support from an MIT Presidential Fellowship and a graduate research assistantship from the NSF BIGDATA grant (number 1741341). SS acknowledges partial support from NSF-BIGDATA (1741341) and NSF-TRIPODS+X (1839258).

## Footnotes

[2]The dependence on reward factorization is omitted here and below, unless otherwise stated. The omission also corresponds to the case of known rewards.

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
