[Supplementary Material]

# A  Regret analysis of F-UCBVI

## A.1  Failure event

Before we define the failure event that causes our regret guarantee to fail, we introduce some additional notation. As a shorthand, for any natural number $n$, any factored set $\mathcal{X} = \bigotimes_{i=1}^{n} \mathcal{X}_i$ and any given index set $I \subset [n]$, let $\mathcal{X}[-I] := \bigotimes_{i=1, i \notin I}^{n} \mathcal{X}_i$ and $x[-I] \in \mathcal{X}[-I]$ be a tuple of $x[j]$ for $j \in [n]$ and $j \notin I$. For a singleton $\{i\}$, let $\mathcal{S}_{-i} \equiv \mathcal{S}[-\{i\}] := (\bigotimes_{j=1}^{i-1} \mathcal{S}_j) \times (\bigotimes_{j=i+1}^{m} \mathcal{S}_j)$. Let $s[-i] \in \mathcal{S}_{-i}$ be a tuple of $s[j]$ for $j \in [m]$ and $j \neq i$. For vector $V \in \mathbb{R}^{\mathcal{S}}$,

$$V(s[-i]) := V((s[1], \cdots, s[i-1], \cdot, s[i+1], \cdots, s[m])) \in \mathbb{R}^{\mathcal{S}_i},$$
$$V(s[i]) := V((\cdot, \cdots, \cdot, s[i], \cdot, \cdots, \cdot)) \in \mathbb{R}^{\mathcal{S}_{-i}}.$$

Recall that $w_{k,h}(x)$ is the visit probability to $x$ at step $h$ of episode $k$. Overloading the notation, we make the following definitions.

**Definition 3** (Visit probabilities). *Define*

$$w_{i,k,h}(x) := w_{i,k,h}(x[I_i]) = \sum\nolimits_{x[-I_i] \in \mathcal{X}[-I_i]} w_{k,h}(x),$$
$$v_{i,k,h}(x) := v_{i,k,h}(x[J_i]) = \sum\nolimits_{x[-J_i] \in \mathcal{X}[-J_i]} w_{k,h}(x).$$

*Then let $w_k(x) := \sum_{h=1}^{H} w_{k,h}(x)$, $w_{i,k}(x) := \sum_{h=1}^{H} w_{i,k,h}(x)$ and $v_{i,k}(x) := \sum_{h=1}^{H} v_{i,k,h}(x)$.*

Recall that $L = \log(16mlSXT/\delta)$. Then we define the failure event below.

**Definition 4** (Failure event). *Define the events*

$$\mathcal{F}_1 := \left\{ \exists (i \in [m], k \in [K], h \in [H], x \in \mathcal{X}), \right.$$
$$\left. \left| \left\langle \hat{P}_{i,k}(x) - P_i(x), \mathbb{E}_{P_{-i}(x)}[V_{h+1}^*] \right\rangle \right| > H\sqrt{\frac{L}{2N_{i,k}(x)}} \right\},$$

$$\mathcal{F}_2 := \left\{ \exists (i \in [m], k \in [K], x \in \mathcal{X}), \quad \left\| \hat{P}_{i,k}(x) - P_i(x) \right\|_1 > \sqrt{\frac{2S_i L}{N_{i,k}(x)}} \right\},$$

$$\mathcal{F}_3 := \left\{ \exists (i \in [l], k \in [K], x \in \mathcal{X}), \quad \left| \hat{R}_{i,k}(x) - R_i(x) \right| > \sqrt{\frac{L}{2M_{i,k}(x)}} \right\},$$

$$\mathcal{F}_4 := \left\{ \exists (i \in [m], k \in [K], x \in \mathcal{X}), \quad N_{i,k}(x) < \frac{1}{2} \sum_{\kappa < k} w_{i,\kappa}(x) - HL \right\},$$

$$\mathcal{F}_5 := \left\{ \exists (i \in [l], k \in [K], x \in \mathcal{X}), \quad M_{i,k}(x) < \frac{1}{2} \sum_{\kappa < k} v_{i,\kappa}(x) - HL \right\},$$

$$\mathcal{F}_6 := \left\{ \exists (i \in [m], k \in [K], x \in \mathcal{X}, s' \in \mathcal{S}), \right.$$
$$\left. \left| \hat{P}_{i,k}(s'[i]|x) - P_i(s'[i]|x) \right| > \frac{2L}{3N_{i,k}(x)} + \sqrt{\frac{2P_i(s'[i]|x)L}{N_{i,k}(x)}} \right\}.$$

*Then the failure event for F-UCBVI is defined by $\mathcal{F} := \bigcup_{i=1}^{6} \mathcal{F}_i$.*

The following lemma shows that the failure event $\mathcal{F}$ happens with low probability.

**Lemma 8** (Failure probability). *For any FMDP specified by (2.1), during the running of F-UCBVI for $K$ episodes, the failure event $\mathcal{F}$ happens with probability at most $\delta$.*

*Proof.* By Hoeffding's inequality (Lemma 37) and the union bound, $\mathcal{F}_1$ happens with probability at most $\delta/8$. The same argument applies to $\mathcal{F}_3$. By $L_1$-norm concentration (Lemma 36) and the union bound, $\mathcal{F}_2$ happens with probability at most $\delta/8$. By the same argument regarding failure event $\mathcal{F}_N$ in [9, Lemma 6, Section B.1], $\mathcal{F}_4$ and $\mathcal{F}_5$ happen with probability at most $\delta/16$ respectively. By the same Bernstein's inequality argument in [1, Lemma 1, Section B.4], $\mathcal{F}_6$ happens with probability at most $\delta/8$. Finally, applying the union bound on $\mathcal{F}_i$ for $i \in [6]$ yields that the failure event $\mathcal{F}$ happens with probability at most $5\delta/8 \leq \delta$. $\qquad\square$

The deduction in the rest of this section and hence the regret bound hold outside the failure event $\mathcal{F}$, with probability at least $1 - \delta$. From the above derivation, note that we can actually use a smaller

$$L_0 = \log(10ml \max\{\max_i(S_i X[I_i]), \max_i X[J_i]\} T/\delta)$$

to replace $L$ for F-UCBVI. We use $L$ for simplicity.

### A.2 Upper confidence bound on the optimal state-value function

The transition estimation error refers to a term incurred by the difference between the estimated transition and the true one. To apply scalar concentration, we use the standard technique that bounds the inner product of their difference and the optimal value function. Specifically, we have the following lemma.

**Lemma 9** (Transition estimation error, Hoeffding-style). *Outside the failure event $\mathcal{F}$, for any episode $k \in [K]$, step $h \in [H]$ and state-action pair $x \in \mathcal{X}$, the transition estimation error satisfies that*

$$\left| \left\langle \hat{P}_k(x) - P(x), V_{h+1}^* \right\rangle \right| \leq \sum_{i=1}^m H \sqrt{\frac{L}{2N_{i,k}(x)}} + \sum_{i=1}^m \sum_{j=i+1}^m 2HL \sqrt{\frac{S_i S_j}{N_{i,k}(x) N_{j,k}(x)}}. \quad \text{(A.1)}$$

*Proof.* Omitting the dependence of $\hat{P}_k(x), P(x), \hat{P}_{i,k}(x), P_i(x)$ on $x$,

$$\left\langle \hat{P}_k - P, V_{h+1}^* \right\rangle = \left\langle \prod_{i=1}^m \hat{P}_{i,k} - \prod_{i=1}^m P_i, V_{h+1}^* \right\rangle$$

$$= \left\langle \sum_{i=1}^m (\hat{P}_{i,k} - P_i) P_{1:i-1} \hat{P}_{i+1:m,k}, V_{h+1}^* \right\rangle$$

$$= \sum_{i=1}^m \left\langle \hat{P}_{i,k} - P_i, \mathbb{E}_{P_{1:i-1} \hat{P}_{i+1:m,k}}[V_{h+1}^*] \right\rangle$$

$$= \sum_{i=1}^m \left\langle \hat{P}_{i,k} - P_i, \mathbb{E}_{P_{1:i-1}} \mathbb{E}_{\hat{P}_{i+1:m,k}}[V_{h+1}^*] \right\rangle$$

$$= \sum_{i=1}^m \left\langle \hat{P}_{i,k} - P_i, \mathbb{E}_{P_{1:i-1}} \mathbb{E}_{P_{i+1:m}}[V_{h+1}^*] \right\rangle \quad \text{(A.2)}$$

$$+ \sum_{i=1}^m \left\langle \hat{P}_{i,k} - P_i, \mathbb{E}_{P_{1:i-1}}(\mathbb{E}_{\hat{P}_{i+1:m,k}} - \mathbb{E}_{P_{i+1:m}})[V_{h+1}^*] \right\rangle, \quad \text{(A.3)}$$

where the second equality adopts an inverse telescoping technique (add and subtract a sequence of terms), essential to our analysis. Outside the failure event $\mathcal{F}$ (specifically, $\mathcal{F}_1$), (A.2) is upper bounded by

$$\left| \left\langle \hat{P}_{i,k}(x) - P_i(x), \mathbb{E}_{P_{1:i-1}(x)} \mathbb{E}_{P_{i+1:m}(x)}[V_{h+1}^*] \right\rangle \right| \leq H \sqrt{\frac{L}{2N_{i,k}(x)}}. \quad \text{(A.4)}$$

By Lemma 10, outside the failure event $\mathcal{F}$, (A.3) is upper bounded by

$$\left| \left\langle \hat{P}_{i,k}(x) - P_i(x), \mathbb{E}_{P_{1:i-1}(x)}(\mathbb{E}_{\hat{P}_{i+1:m,k}(x)} - \mathbb{E}_{P_{i+1:m}(x)})[V_{h+1}^*] \right\rangle \right|$$

$$\leq \sum_{j=i+1}^m 2HL \sqrt{\frac{S_i S_j}{N_{i,k}(x) N_{j,k}(x)}}. \quad \text{(A.5)}$$

Combining (A.4) and (A.5) yields the transition estimation error bound (A.1).  □

The following Lemma 10 brings in the cross-component term, also as part of the transition bonus later, which results from applying inverse telescoping once and $L_1$-norm concentration (lemma 36) twice.

**Lemma 10** (Holder's argument). *Outside the failure event $\mathcal{F}$, for any index $i \in [m]$, episode $k \in [K]$, step $h \in [H]$ and state-action pair $x \in \mathcal{X}$,*

$$\left| \left\langle \hat{P}_{i,k}(x) - P_i(x), \mathbb{E}_{P_{1:i-1}(x)}(\mathbb{E}_{\hat{P}_{i+1:m,k}(x)} - \mathbb{E}_{P_{i+1:m}(x)})[V_{h+1}^*] \right\rangle \right|$$

$$\leq \sum_{j=i+1}^{m} 2HL\sqrt{\frac{S_i S_j}{N_{i,k}(x)N_{j,k}(x)}}.$$

*Proof.* By Holder's inequality,

$$\left| \left\langle \hat{P}_{i,k}(x) - P_i(x), \mathbb{E}_{P_{1:i-1}(x)}(\mathbb{E}_{\hat{P}_{i+1:m,k}(x)} - \mathbb{E}_{P_{i+1:m}(x)})[V_{h+1}^*] \right\rangle \right|$$

$$\leq \left\| \hat{P}_{i,k}(x) - P_i(x) \right\|_1 \cdot \left\| \mathbb{E}_{P_{1:i-1}(x)}(\mathbb{E}_{\hat{P}_{i+1:m,k}(x)} - \mathbb{E}_{P_{i+1:m}(x)})[V_{h+1}^*] \right\|_\infty$$

$$\leq \sqrt{\frac{2S_i L}{N_{i,k}(x)}} \cdot \left\| \mathbb{E}_{P_{1:i-1}(x)}(\mathbb{E}_{\hat{P}_{i+1:m,k}(x)} - \mathbb{E}_{P_{i+1:m,h}(x)})[V_{h+1}^*] \right\|_\infty, \qquad (\text{A.6})$$

where the second inequality holds outside the failure event $\mathcal{F}$ (specifically, $\mathcal{F}_2$). We proceed to bound the $L_\infty$-norm term by applying the inverse telescoping technique. Omitting the dependence of $\hat{P}_k(x), P(x), \hat{P}_{i,k}(x), P_i(x)$ on $x$ and defining the empty product (e.g., $P_{i+1:i}$) to be 1, for any $i \in [m]$,

$$\mathbb{E}_{P_{1:i-1}}(\mathbb{E}_{\hat{P}_{i+1:m,k}} - \mathbb{E}_{P_{i+1:m}})[V_{h+1}^*]$$

$$= \left\langle \prod_{j=i+1}^{m} \hat{P}_{j,k} - \prod_{j=i+1}^{m} P_j, \mathbb{E}_{P_{1:i-1}}[V_{h+1}^*] \right\rangle$$

$$= \left\langle \sum_{j=i+1}^{m} (\hat{P}_{j,k} - P_j)P_{i+1:j-1}\hat{P}_{j+1:m,k}, \mathbb{E}_{P_{1:i-1}}[V_{h+1}^*] \right\rangle$$

$$= \sum_{j=i+1}^{m} \left\langle \hat{P}_{j,k} - P_j, \mathbb{E}_{P_{1:i-1}}\mathbb{E}_{P_{i+1:j-1}}\mathbb{E}_{\hat{P}_{j+1:m,k}}[V_{h+1}^*] \right\rangle \in \mathbb{R}^{\mathcal{S}_i}.$$

Therefore, for any $i \in [m]$ and $s'[i] \in \mathcal{S}_i$,

$$\left| \mathbb{E}_{P_{1:i-1}}(\mathbb{E}_{\hat{P}_{i+1:m,k}} - \mathbb{E}_{P_{i+1:m}})[V_{h+1}^*](s'[i]) \right|$$

$$= \left| \mathbb{E}_{P_{1:i-1}}(\mathbb{E}_{\hat{P}_{i+1:m,k}} - \mathbb{E}_{P_{i+1:m}})[V_{h+1}^*(s'[i])] \right|$$

$$\leq \sum_{j=i+1}^{m} \left| \left\langle \hat{P}_{j,k} - P_j, \mathbb{E}_{P_{1:i-1}}\mathbb{E}_{P_{i+1:j-1}}\mathbb{E}_{\hat{P}_{j+1:m,k}}[V_{h+1}^*(s'[i])] \right\rangle \right|$$

$$\leq \sum_{j=i+1}^{m} \left\| \hat{P}_{j,k} - P_j \right\|_1 \cdot \left\| \mathbb{E}_{P_{i+1:j-1}}\mathbb{E}_{\hat{P}_{j+1:m,k}}[V_{h+1}^*(s'[i])] \right\|_\infty$$

$$\leq \sum_{j=i+1}^{m} \sqrt{\frac{2S_j L}{N_{j,k}}} \cdot H,$$

where the last inequality holds outside the failure event $\mathcal{F}$ (specifically, $\mathcal{F}_2$). Substituting the above into (A.6) yields

$$\left| \left\langle \hat{P}_{i,k}(x) - P_i(x), \mathbb{E}_{P_{1:i-1}(x)}(\mathbb{E}_{\hat{P}_{i+1:m,k}(x)} - \mathbb{E}_{P_{i+1:m}(x)})[V_{h+1}^*] \right\rangle \right|$$

$$\leq \sum_{j=i+1}^{m} \sqrt{\frac{2S_i L}{N_{i,k}(x)}} \cdot H\sqrt{\frac{2S_j L}{N_{j,k}(x)}}$$

$$= \sum_{j=i+1}^{m} 2HL\sqrt{\frac{S_i S_j}{N_{i,k}(x)N_{j,k}(x)}}. \qquad \square$$

Recall that our Hoeffding-style transition bonus (3.1) is exactly the transition estimation error bound in (A.1). Add a subscript $k$ to $\hat{R}, \hat{R}_i$ to denote the corresponding quantities in the $k$th episode. Recall that our choice of the reward bonus upper bounds the reward estimation error outside the failure event $\mathcal{F}$ (specifically, $\mathcal{F}_3$), i.e.,

$$\left| \hat{R}_k(x) - R(x) \right| \leq \sum_{i=1}^{l} \left| \hat{R}_{i,k}(x) - R_i(x) \right| \leq \sum_{i=1}^{l} \sqrt{\frac{L}{2M_{i,k}(x)}} := \beta_k(x).$$

The following lemma shows that these choices ensure the optimism. Specifically, $\overline{V}_{k,h}$ is an entrywise UCB of $V_h^*$ for all $k \in [K], h \in [H]$.

**Lemma 11** (Upper confidence bound). *Outside the failure event $\mathcal{F}$, for the choices of bonuses in (3.1) and (3.2), $V_h^*(s) \leq \overline{V}_{k,h}(s)$ for any episode $k \in [K]$, step $h \in [H]$ and state $s \in \mathcal{S}$.*

*Proof.* For $h = H + 1$, $V_{H+1}^*(s) = V_{k,H+1}(s) = 0$ for all $k \in [K]$ and $s \in \mathcal{S}$. We proceed by backward induction. For all $k \in [K]$, for a given $h \in [H]$, for all $s \in \mathcal{S}$, with $x_{k,h} = (s, \pi_k(s,h))$ and $x_h^* = (s, \pi^*(s,h))$,

$$\overline{V}_{k,h}(s) - V_h^*(s)$$
$$= \hat{R}(x_{k,h}) + \beta_k(x_{k,h}) + \left\langle \hat{P}_k(x_{k,h}), \overline{V}_{k,h+1} \right\rangle + b_k(x_{k,h}) - R(x_h^*) - \left\langle P(x_h^*), V_{h+1}^* \right\rangle$$
$$\geq \hat{R}(x_h^*) + \beta_k(x_h^*) + \left\langle \hat{P}_k(x_h^*), \overline{V}_{k,h+1} \right\rangle + b_k(x_h^*) - R(x_h^*) - \left\langle P(x_h^*), V_{h+1}^* \right\rangle$$
$$\geq \left\langle \hat{P}_k(x_h^*), \overline{V}_{k,h+1} - V_{h+1}^* \right\rangle + \hat{R}(x_h^*) - R(x_h^*) + \beta_k(x_h^*)$$
$$\quad + \left\langle \hat{P}_k(x_h^*) - P(x_h^*), V_{h+1}^* \right\rangle + b_k(x_h^*),$$

where the first equality corresponds to the nontrivial case where $\overline{V}_{k,h}(s) < H - h + 1$. Since $\left\langle \hat{P}_k(x_h^*), \overline{V}_{k,h+1} - V_{h+1}^* \right\rangle \geq 0$ by the inductive assumption, we have $\overline{V}_{k,h}(s) - V_h^*(s) \geq 0$ outside the failure event $\mathcal{F}$. Therefore, $V_h^*(s) \leq \overline{V}_{k,h}(s)$ for all $k \in [K], h \in [H], s \in \mathcal{S}$. $\qquad \square$

Refer to the difference between the optimistic value function than the optimal value function as the *confidence radius*. We now bound the confidence radius in the following lemma. After the introduction of "good" sets, we then bound the sum over time of the squared confidence radius, which is useful to prove that the cumulative correction term is lower-order (polylog in $T$, Lemma 22).

**Lemma 12** (Confidence radius, Hoeffding-style). *Let $F_0 := 5mH \max_i S_i L$ be a lower-order term. Let $s_{k,t} \in \mathcal{S}$ denote the state at step $t$ of episode $k$ and $x_{k,t} = (s_{k,t}, \pi_k(s_{k,t}, t))$. Outside the failure event $\mathcal{F}$, for any episode $k \in [K]$, step $h \in [H]$ and state $s \in \mathcal{S}$, the confidence radius of F-UCBVI satisfies that*

$$\overline{V}_{k,h}(s) - V_h^*(s) \leq \min\left\{ \sum_{t=h}^{H} \mathbb{E}_{\pi_k}\left[ \sum_{i=1}^{m} \frac{F_0}{\sqrt{N_{i,k}(x_{k,t})}} + \sum_{i=1}^{l} \sqrt{\frac{2L}{M_{i,k}(x_{k,t})}} \,\middle|\, s_{k,h} = s \right], H \right\}.$$

*Proof.* By definition, for any $k \in [K], h \in [H], s \in \mathcal{S}$,

$$\overline{V}_{k,h}(s) - V_h^*(s)$$

$$\leq \hat{R}(x_{k,h}) + \beta_k(x_{k,h}) + \left\langle \hat{P}_k(x_{k,h}), \overline{V}_{k,h+1} \right\rangle + b_k(x_{k,h}) - R(x_h^*) - \left\langle P(x_h^*), V_{h+1}^* \right\rangle$$

$$\leq \hat{R}(x_{k,h}) + \beta_k(x_{k,h}) + \left\langle \hat{P}_k(x_{k,h}), \overline{V}_{k,h+1} \right\rangle + b_k(x_{k,h}) - R(x_{k,h}) - \left\langle P(x_{k,h}), V_{h+1}^* \right\rangle$$

$$\leq 2\beta_k(x_{k,h}) + \left\langle \hat{P}_k(x_{k,h}) - P(x_{k,h}), \overline{V}_{k,h+1} \right\rangle + \left\langle P(x_{k,h}), \overline{V}_{k,h+1} - V_{h+1}^* \right\rangle + b_k(x_{k,h})$$

$$\leq \left\langle P(x_{k,h}), \overline{V}_{k,h+1} - V_{h+1}^* \right\rangle + \sum_{i=1}^{m} H\sqrt{\frac{2S_i L}{N_{i,k}(x)}} + b_k(x_{k,h}) + 2\beta_k(x_{k,h}). \tag{A.7}$$

For $b_k(x_{k,h})$, we have

$$b_k(x_{k,h}) = \sum_{i=1}^{m} H\sqrt{\frac{L}{2N_{i,k}(x_{k,h})}} + \sum_{i=1}^{m} \sum_{j=i+1}^{m} 2HL\sqrt{\frac{S_i S_j}{N_{i,k}(x_{k,h}) N_{j,k}(x_{k,h})}}$$

$$\leq \sum_{i=1}^{m} H\sqrt{\frac{L}{2N_{i,k}(x_{k,h})}} + \sum_{i=1}^{m} 2m \max_i S_i HL \frac{1}{\sqrt{N_{i,k}(x_{k,h})}}$$

$$\leq 3m \max_i S_i HL \sum_{i=1}^{m} \frac{1}{\sqrt{N_{i,k}(x_{k,h})}}.$$

Substituting the above into (A.7) yields

$$\overline{V}_{k,h}(s) - V_h^*(s)$$

$$\leq \left\langle P(x_{k,h}), \overline{V}_{k,h+1} - V_{h+1}^* \right\rangle + 5mH \max_i S_i L \sum_{i=1}^{m} \frac{1}{\sqrt{N_{i,k}(x)}} + \sum_{i=1}^{l} \sqrt{\frac{2L}{M_{i,k}(x_{k,h})}}$$

$$= \left\langle P(x_{k,h}), \overline{V}_{k,h+1} - V_{h+1}^* \right\rangle + \sum_{i=1}^{m} \frac{F_0}{\sqrt{N_{i,k}(x)}} + \sum_{i=1}^{l} \sqrt{\frac{2L}{M_{i,k}(x_{k,h})}}.$$

By backward induction over the subscript $h$, we have

$$\overline{V}_{k,h}(s) - V_h^*(s) \leq \min \left\{ \sum_{t=h}^{H} \mathbb{E}_{\pi_k} \left[ \sum_{i=1}^{m} \frac{F_0}{\sqrt{N_{i,k}(x_{k,t})}} + \sum_{i=1}^{l} \sqrt{\frac{2L}{M_{i,k}(x_{k,t})}} \middle| s_{k,h} = s \right], H \right\}.$$

$\square$

## A.3 Good sets of state-action pairs

The following "good sets" [44] are a notion of sufficient visits so that the estimations are meaningful, the introduction of which is seminal to the sum-over-time analysis.

**Definition 5** (Good sets). *Define the good sets of state-action components for transition and reward estimations as*

$$L_{i,k} := \left\{ x[I_i] \in \mathcal{X}[I_i] : \frac{1}{4} \sum_{\kappa < k} w_{i,\kappa}(x[I_i]) \geq HL + H \right\},$$

$$\Lambda_{i,k} := \left\{ x[J_i] \in \mathcal{X}[J_i] : \frac{1}{4} \sum_{\kappa < k} v_{i,\kappa}(x[J_i]) \geq HL + H \right\}.$$

*Then the corresponding good sets of state-action pairs are defined by*

$$L_k := \{x \in \mathcal{X} : x[I_i] \in L_{i,k} \text{ for all } i \in [m]\},$$
$$\Lambda_k := \{x \in \mathcal{X} : x[J_i] \in \Lambda_{i,k} \text{ for all } i \in [l]\}.$$

We shall restrict our attention to the state-action pairs in the good sets (with sufficient visits). To this end, we show the sums of visit probabilities to the state-action pairs outside the good sets are lower-order terms in the following lemma.

**Lemma 13** (Sum out of good sets). *The sums of the visit probabilities of the state-action pairs out of the good sets and over time satisfy that*

$$\sum_{k=1}^{K}\sum_{h=1}^{H}\sum_{x\notin L_k} w_{k,h}(x) \le 8\sum_{i=1}^{m} X[I_i]HL.$$

$$\sum_{k=1}^{K}\sum_{h=1}^{H}\sum_{x\notin \Lambda_k} v_{k,h}(x) \le 8\sum_{i=1}^{l} X[J_i]HL.$$

*Proof.* If $x[I_i] \notin L_{i,k}$, then by definition,

$$\frac{1}{4}\sum_{\kappa\le k} w_{i,\kappa}(x[I_i]) < HL + H + H = H(L+2).$$

Therefore,

$$\sum_{k=1}^{K}\sum_{h=1}^{H}\sum_{x\notin L_k} w_{k,h}(x) \le \sum_{k=1}^{K}\sum_{h=1}^{H}\sum_{x\in\mathcal{X}} w_{k,h}(x)\mathbb{I}(x\notin L_k)$$

$$\le \sum_{k=1}^{K}\sum_{h=1}^{H}\sum_{x\in\mathcal{X}} w_{k,h}(x)\sum_{i=1}^{m}\mathbb{I}(x[I_i]\notin L_{i,k})$$

$$\le \sum_{i=1}^{m}\sum_{x[I_i]\in\mathcal{X}[I_i]}\sum_{k=1}^{K}\sum_{h=1}^{H} w_{i,k,h}(x[I_i])\mathbb{I}(x[I_i]\notin L_{i,k})$$

$$= \sum_{i=1}^{m}\sum_{x[I_i]\in\mathcal{X}[I_i]}\sum_{k=1}^{K} w_{i,k}(x[I_i])\mathbb{I}(x[I_i]\notin L_{i,k})$$

$$\le 4\sum_{i=1}^{m} X[I_i]H(L+2),$$

where in the third inequality we write $\sum_{x\in\mathcal{X}}$ as $\sum_{x[I_i]\in\mathcal{X}[I_i]}\sum_{x[-I_i]\in\mathcal{X}[-I_i]}$ and use the definition of $w_{i,k,h}$ (Definition 3). Since $L = \log(16mlSXT/\delta) \ge 2$, we have

$$\sum_{k=1}^{K}\sum_{h=1}^{H}\sum_{x\notin L_k} w_{k,h}(x) \le 8\sum_{i=1}^{m} X[I_i]HL.$$

The same argument applies to $v_{k,h}(x)$. $\qquad\square$

The following lemma bridges the visit probabilities $w_{i,k}$ and $v_{i,k}$ to the actual numbers of visits $N_{i,k}$ and $M_{i,k}$ for the state-action pairs in the good sets.

**Lemma 14** (Visit number and visit probability). *Outside the failure event $\mathcal{F}$, the numbers of visits $N_{i,k}$ and $M_{i,k}$ to the state-action pairs in the good sets satisfy that*

$$N_{i,k}(x) \ge \frac{1}{4}\sum_{\kappa\le k} w_{i,\kappa}(x) \quad \text{for all } i\in[m] \text{ and } x\in L_k,$$

$$M_{i,k}(x) \ge \frac{1}{4}\sum_{\kappa\le k} w_{i,\kappa}^r(x) \quad \text{for all } i\in[l] \text{ and } x\in\Lambda_k.$$

*Proof.* Outside the failure event $\mathcal{F}$ (specifically, $\mathcal{F}_4$), for all $i \in [m]$,

$$
\begin{aligned}
N_{i,k}(x) &\geq \frac{1}{2} \sum_{\kappa < k} w_{i,\kappa}(x) - HL \\
&= \frac{1}{4} \sum_{\kappa < k} w_{i,\kappa}(x) + \frac{1}{4} \sum_{\kappa < k} w_{i,\kappa}(x) - HL \\
&\geq \frac{1}{4} \sum_{\kappa < k} w_{i,\kappa}(x) + H \\
&\geq \frac{1}{4} \sum_{\kappa \leq k} w_{i,\kappa}(x),
\end{aligned}
$$

where the second inequality results from the definition of good sets (Definition 5). Outside the failure event $\mathcal{F}$ (specifically, $\mathcal{F}_5$), the same argument applies to $M_{i,k}(x)$ for all $i \in [l]$. □

By Lemma 14 and the definition of the good sets (Definition 5), for all $k \in [K]$, $N_{i,k}(x) \geq HL + H \geq 2$ for all $x \in L_k$, and $M_{i,k}(x) \geq HL + H \geq 2$ for all $x \in \Lambda_k$. Therefore, the regret analysis out of the good sets automatically precludes the cases of zero denominators in Algorithms 1 and 2, where we replace the zeros by ones for algorithmic completeness.

Refer to the ratio of visit probability $w_{k,h}$ to visit number $N_{i,k}$ or $M_{i,k}$ as the *visit ratio*. Then the accumulation of the visit ratios turns out to be a lower-order term, as shown in the following lemma.

**Lemma 15** (Sum of visit ratio in good sets). *Outside the failure event $\mathcal{F}$, the sums of the visit ratios of the state-action pairs within the good sets and over time satisfy that*

$$
\sum_{k=1}^{K} \sum_{h=1}^{H} \sum_{x \in L_k} \frac{w_{k,h}(x)}{N_{i,k}(x)} = \sum_{k=1}^{K} \sum_{x \in L_k} \frac{w_k(x)}{N_{i,k}(x[I_i])} \leq 4X[I_i]L \quad \text{for all } i \in [m],
$$

$$
\sum_{k=1}^{K} \sum_{h=1}^{H} \sum_{x \in \Lambda_k} \frac{w_{k,h}(x)}{M_{i,k}(x)} = \sum_{k=1}^{K} \sum_{x \in \Lambda_k} \frac{w_k(x)}{M_{i,k}(x[J_i])} \leq 4X[J_i]L \quad \text{for all } i \in [l].
$$

*Proof.* Outside the failure event $\mathcal{F}$, for any $i \in [m]$, by Lemma 14,

$$
\begin{aligned}
\sum_{k=1}^{K} \sum_{x \in L_k} \frac{w_k(x)}{N_{i,k}(x[I_i])} &\leq \sum_{k=1}^{K} \sum_{x \in \mathcal{X}} \frac{w_k(x)}{N_{i,k}(x[I_i])} \mathbb{I}(x[I_i] \in L_{i,k}) \\
&\leq \sum_{k=1}^{K} \sum_{x[I_i] \in \mathcal{X}[I_i]} \frac{w_{i,k}(x[I_i])}{N_{i,k}(x[I_i])} \mathbb{I}(x[I_i] \in L_{i,k}) \\
&\leq 4 \sum_{k=1}^{K} \sum_{x[I_i] \in \mathcal{X}[I_i]} \frac{w_{i,k}(x[I_i])}{\sum_{\kappa \leq k} w_{i,\kappa}(x[I_i])} \mathbb{I}(x[I_i] \in L_{i,k}) \\
&\leq 4X[I_i]L,
\end{aligned}
$$

where the last inequality is shown by the proof of Lemma 13 in [44]. The same argument applies to the visit ratio $w_{k,h}(x)/M_{i,k}(x)$ for any $i \in [l]$. □

As to be shown below, the cross-component transition bonus term brings in the mixed visit ratio $w_{k,h}(x)/\sqrt{N_{i,k}(x)N_{j,k}(x)}$ in the analysis. By the Cauchy-Schwarz inequality, we immediately have the following control on the accumulation of the mixed visit ratios.

**Lemma 16** (Sum of mixed visit ratio in good sets). *Outside the failure event $\mathcal{F}$, the sum of the mixed visit ratios of the state-action pairs within the good set $L_k$ and over time satisfies that*

$$
\sum_{k=1}^{K} \sum_{h=1}^{H} \sum_{x \in L_k} \frac{w_{k,h}(x)}{\sqrt{N_{i,k}(x)N_{j,k}(x)}} \leq 4\sqrt{X[I_i]X[I_j]}L \quad \text{for all } i, j \in [m].
$$

*Proof.* Outside the failure event $\mathcal{F}$, by the Cauchy-Schwarz inequality and Lemma 15, for any $i, j \in [m]$,

$$\sum_{k=1}^{K}\sum_{h=1}^{H}\sum_{x \in L_k} \frac{w_k(x)}{\sqrt{N_{i,k}(x)N_{j,k}(x)}} = \sum_{k=1}^{K}\sum_{x \in L_k} \frac{w_k(x)}{\sqrt{N_{i,k}(x[I_i])N_{j,k}(x[I_j])}}$$

$$\leq \sqrt{\sum_{k=1}^{K}\sum_{x \in L_k} \frac{w_k(x)}{N_{i,k}(x[I_i])}} \cdot \sqrt{\sum_{k=1}^{K}\sum_{x \in L_k} \frac{w_k(x)}{N_{j,k}(x[I_j])}}$$

$$\leq 4\sqrt{X[I_i]X[I_j]L}. \qquad \square$$

The following lemma bounds the sum over time of the expected squared confidence radius, through the proof of which we can see an initial application of the above lemmas obtained with the notion of good sets. For clarification, by "sum over time", we mean the $w_{k,h}(x)$-weighted sum over $k \in [K], h \in [H]$ and $x \in L_k$ or $x \in \Lambda_k$ henceforth.

**Lemma 17** (Cumulative confidence radius, Hoeffding-style). *Define the lower-order term*

$$G_0 := 208m^4 H^4 (\max_i S_i)^2 \max_i X[I_i]L^3 + 24l^2 H^3 \max_i X[J_i]L^2.$$

*Then outside the failure event $\mathcal{F}$, for all $i \in [m]$, the sum over time of the following expected squared confidence radius of F-UCBVI satisfies that*

$$\sum_{k=1}^{K}\sum_{h=1}^{H}\sum_{x \in \mathcal{X}} w_{k,h}(x) \left( \mathbb{E}_{P_i} \left( \mathbb{E}_{P_{-i}}[\overline{V}_{k,h+1} - V_{h+1}^*] \right)^2 \right)$$

$$\leq \sum_{k=1}^{K}\sum_{h=1}^{H}\sum_{x \in \mathcal{X}} w_{k,h}(x) \left( \mathbb{E}_P[(\overline{V}_{k,h+1} - V_{h+1}^*)^2] \right) \leq G_0.$$

*Proof.* Let $s_{k,h} \in \mathcal{S}$ denote the state at step $h$ of episode $k$. Since $(\mathbb{E}[X])^2 \leq \mathbb{E}[X^2]$ for any random variable $X$, we have that for all $i \in [m]$,

$$\sum_{k=1}^{K}\sum_{h=1}^{H}\sum_{x \in \mathcal{X}} w_{k,h}(x) \left( \mathbb{E}_{P_i} \left( \mathbb{E}_{P_{-i}}[\overline{V}_{k,h+1} - V_{h+1}^*] \right)^2 \right)$$

$$\leq \sum_{k=1}^{K}\sum_{h=1}^{H}\sum_{x \in \mathcal{X}} w_{k,h}(x) \left( \mathbb{E}_{P_i}\mathbb{E}_{P_{-i}}[(\overline{V}_{k,h+1} - V_{h+1}^*)^2] \right)$$

$$= \sum_{k=1}^{K}\sum_{h=1}^{H}\sum_{x \in \mathcal{X}} w_{k,h}(x) \left( \sum_{s' \in \mathcal{S}} P(s'|x)(\overline{V}_{k,h+1}(s') - V_{h+1}^*(s'))^2 \right)$$

$$= \sum_{k=1}^{K}\sum_{h=1}^{H} \mathbb{E}_{\pi_k} \left[ (\overline{V}_{k,h+1}(s_{k,h+1}) - V_{h+1}^*(s_{k,h+1}))^2 \Big| s_{k,1} \right]$$

$$\leq \sum_{k=1}^{K}\sum_{h=1}^{H} \mathbb{E}_{\pi_k} \left[ (\overline{V}_{k,h}(s_{k,h}) - V_h^*(s_{k,h}))^2 \Big| s_{k,1} \right]. \tag{A.8}$$

By the confidence radius lemma (Lemma 12), for any $k \in [K], h \in [H]$,

$$\mathbb{E}_{\pi_k}\left[\left(\overline{V}_{k,h}(s_{k,h}) - V_h^*(s_{k,h})\right)^2\Big|s_{k,1}\right]$$

$$\leq \mathbb{E}_{\pi_k}\left[\left(\left(\sum_{t=h}^{H}\mathbb{E}_{\pi_k}\left[\sum_{i=1}^{m}\frac{F_0}{\sqrt{N_{i,k}(x_{k,t})}} + \sum_{i=1}^{l}\sqrt{\frac{2L}{M_{i,k}(x_{k,t})}}\Big|s_{k,h}\right]\right)^2\Big|s_{k,1}\right]$$

$$\leq 2mHF_0^2\sum_{i=1}^{m}\sum_{t=h}^{H}\mathbb{E}_{\pi_k}\left[\frac{1}{N_{i,k}(x_{k,t})}\Big|s_{k,1}\right] + 4lHL\sum_{i=1}^{l}\sum_{t=h}^{H}\mathbb{E}_{\pi_k}\left[\frac{1}{M_{i,k}(x_{k,t})}\Big|s_{k,1}\right]$$

$$\leq 2mH^2F_0^2\sum_{i=1}^{m}\mathbb{E}_{\pi_k}\left[\frac{1}{N_{i,k}(x_{k,h})}\Big|s_{k,1}\right] + 4lH^2L\sum_{i=1}^{l}\mathbb{E}_{\pi_k}\left[\frac{1}{M_{i,k}(x_{k,t})}\Big|s_{k,1}\right],$$

where in the second inequality we use the inequality $\left(\sum_{i=1}^{n}a_i\right)^2 \leq n\sum_{i=1}^{n}a_i^2$ for multiple times ($n = 2, m, l, H$). The confidence radius lemma (Lemma 12) also guarantees that

$$\mathbb{E}_{\pi_k}\left[\left(\overline{V}_{k,h}(s_{k,h}) - V_h^*(s_{k,h})\right)^2\Big|s_{k,1}\right] \leq H^2.$$

Therefore, substituting the above two bounds into (A.8) and by Lemmas 13 and 15, we have

$$\sum_{k=1}^{K}\sum_{h=1}^{H}\mathbb{E}_{\pi_k}\left[\left(\overline{V}_{k,h}(s_{k,h}) - V_h^*(s_{k,h})\right)^2\Big|s_{k,1}\right]$$

$$\leq 2mH^2F_0^2\sum_{i=1}^{m}\sum_{k=1}^{K}\sum_{h=1}^{H}\sum_{x\in L_k}\frac{w_{k,h}(x)}{N_{i,k}(x)} + 4lH^2L\sum_{i=1}^{l}\sum_{k=1}^{K}\sum_{h=1}^{H}\sum_{x\in\Lambda_k}\frac{w_{k,h}(x)}{M_{i,k}(x)}$$

$$+ \sum_{k=1}^{K}\sum_{h=1}^{H}\sum_{x\notin L_k}w_{k,h}(x)H^2 + \sum_{k=1}^{K}\sum_{h=1}^{H}\sum_{x\notin\Lambda_k}w_{k,h}(x)H^2$$

$$\leq 2mH^2F_0^2\sum_{i=1}^{m}4X[I_i]L + 4lH^2L\sum_{i=1}^{l}4X[J_i]L + 8H^3\sum_{i=1}^{m}X[I_i]L + 8H^3\sum_{i=1}^{l}X[J_i]L$$

$$\leq 8mH^2F_0^2\sum_{i=1}^{m}X[I_i]L + 16lH^2\sum_{i=1}^{l}X[J_i]L^2 + 8H^3\sum_{i=1}^{m}X[I_i]L + 8H^3\sum_{i=1}^{l}X[J_i]L$$

$$= 208m^4H^4(\max_i S_i)^2\max_i X[I_i]L^3 + 24l^2H^3\max_i X[J_i]L^2,$$

where in the last equality we use the definition of $F_0$ (Lemma 12). $\qquad\square$

## A.4 Regret decomposition

We decompose the regret in the following standard way [44], and then bound the sum over time of the individual terms in the next few subsections. Here we assume the general transition bonus $b$ to be a function of step $h$, as in F-EULER.

**Lemma 18** (Regret decomposition). *Let $L_k, \Lambda_k$ be the good sets defined in Definition 5. Then for any given FMDP specified in (2.1), outside the failure event $\mathcal{F}$, the regret of F-UCBVI in $K$ episodes satisfies that*

$$\text{Regret}(K)$$

$$\leq \sum_{k=1}^{K}\sum_{x\in\mathcal{X}}w_{k,h}(x)\min\left\{\left(\left\langle\hat{P}_k(x) - P(x), \overline{V}_{k,h+1}\right\rangle + b_{k,h}(x) + \hat{R}_k(x) - R(x) + \beta_k(x)\right), H\right\}$$

$$\leq \sum_{k=1}^{K}\sum_{h=1}^{H}\sum_{x\in L_k}w_{k,h}(x)\left(\underbrace{\left\langle\hat{P}_k(x) - P(x), V_{h+1}^*\right\rangle}_{\text{transition estimation error}} + b_{k,h}(x) + \underbrace{\left\langle\hat{P}_k(x) - P(x), \overline{V}_{k,h+1} - V_{h+1}^*\right\rangle}_{\text{correction term}}\right)$$

$$+ \sum_{k=1}^{K}\sum_{h=1}^{H}\sum_{x\in\Lambda_k}w_{k,h}(x)2\beta_k(x) + 8H^2\sum_{i=1}^{m}X[I_i]L + 8H^2\sum_{i=1}^{l}X[J_i]L,$$

*where the $b_{k,h}(x)$ term is referred to as "transition optimism" and $2\beta_k(x)$ term is referred to as "reward estimation error and optimism".*

*Proof.* Add a subscript $k$ to $\overline{Q}_h$ in the VI_Optimism procedure (Algorithm 2) to denote the corresponding optimistic Q-value function in episode $k$. Since $\overline{V}_{k,h}$ is an entrywise UCB of $V_h^*$, we upper bound the regret by

$$\text{Regret}(K) \leq \sum_{k=1}^{K} \overline{V}_{k,1}(s_{k,1}) - V_1^{\pi_k}(s_{k,1}) \tag{A.9}$$

$$= \sum_{k=1}^{K} \sum_{x \in \mathcal{X}} w_{k,1}(x)(Q_{k,1}(x) - Q_1^{\pi_k}(x))$$

$$= \sum_{k=1}^{K} \sum_{x \in \mathcal{X}} w_{k,1}(x)\Big(\min\Big\{\hat{R}_k(x) + \beta_k(x) + \Big\langle \hat{P}_k(x), \overline{V}_{k,2}\Big\rangle + b_{k,1}(x), H\Big\}$$

$$- R(x) - \langle P(x), V_2^{\pi_k}\rangle\Big) \tag{A.10}$$

$$\leq \sum_{k=1}^{K} \sum_{x \in \mathcal{X}} w_{k,1}(x)\Big(\min\Big\{\Big(\hat{R}_k(x) - R(x) + \beta_k(x) + \Big\langle \hat{P}_k(x) - P(x), \overline{V}_{k,2}\Big\rangle$$

$$+ b_{k,1}(x)\Big), H\Big\} + \Big\langle P(x), \overline{V}_{k,2} - V_2^{\pi_k}\Big\rangle\Big)$$

$$= \sum_{k=1}^{K} \Big(\sum_{x \in \mathcal{X}} w_{k,1}(x)\min\Big\{\Big(\hat{R}_k(x) - R(x) + \beta_k(x) + \Big\langle \hat{P}_k(x) - P(x), \overline{V}_{k,2}\Big\rangle$$

$$+ b_{k,1}(x)\Big), H\Big\} + \sum_{x \in \mathcal{X}} w_{k,1}(x) \sum_{s'} P(s'|x)\big(\overline{V}_{k,2}(s') - V_2^{\pi_k}(s')\big)\Big). \tag{A.11}$$

Let $x' = (s', a')$. By definition, the visit probability $w_{k,h}(x)$ has the property that $w_{k,h+1}(x') = \sum_{x \in \mathcal{X}} w_{k,h}(x) P(s'|x) \mathbb{P}(\pi_k(s', h) = a')$, where $\mathbb{P}(\cdot)$ denotes an appropriate probability measure. Hence,

$$\sum_{x \in \mathcal{X}} w_{k,1}(x) \sum_{s'} P(s'|x)\big(\overline{V}_{k,2}(s') - V_2^{\pi_k}(s')\big)$$

$$= \sum_{x \in \mathcal{X}} w_{k,1}(x) \sum_{x' \in \mathcal{X}} P(s'|x)\mathbb{P}(\pi_k(s', 1) = a')\big(Q_{k,2}(x') - Q_2^{\pi_k}(x')\big)$$

$$= \sum_{x' \in \mathcal{X}} w_{k,2}(x')\big(Q_{k,2}(x') - Q_2^{\pi_k}(x')\big).$$

Substituting the above into (A.11) yields

$$\text{Regret}(K) \leq \sum_{k=1}^{K} \sum_{x \in \mathcal{X}} w_{k,1}(x)(Q_{k,1}(x) - Q_1^{\pi_k}(x))$$

$$\leq \sum_{k=1}^{K} \Big(\sum_{x \in \mathcal{X}} w_{k,1}(x)\min\Big\{\Big(\hat{R}_k(x) - R(x) + \beta_k(x) + \Big\langle \hat{P}_k(x) - P(x), \overline{V}_{k,2}\Big\rangle$$

$$+ b_{k,1}(x)\Big), H\Big\} + \sum_{x \in \mathcal{X}} w_{k,2}(x)\big(Q_{k,2}(x) - Q_2^{\pi_k}(x)\big)\Big).$$

Inductively, we have

$$\text{Regret}(K)$$

$$\leq \sum_{k=1}^{K} \sum_{x \in \mathcal{X}} w_{k,h}(x)\min\Big\{\Big(\hat{R}_k(x) - R(x) + \beta_k(x) + \Big\langle \hat{P}_k(x) - P(x), \overline{V}_{k,h+1}\Big\rangle + b_{k,h}(x)\Big), H\Big\}.$$

Outside the failure event $\mathcal{F}$ (specifically, $\mathcal{F}_3$ for F-UCBVI),

$\text{Regret}(K)$

$$\leq \sum_{k=1}^{K}\sum_{h=1}^{H}\sum_{x\in\mathcal{X}} w_{k,h}(x)\min\left\{\left(\left\langle \hat{P}_k(x)-P(x),\overline{V}_{k,h+1}\right\rangle + b_{k,h}(x)+2\beta_k(x)\right),H\right\}$$

$$= \sum_{k=1}^{K}\sum_{h=1}^{H}\sum_{x\in\mathcal{X}} w_{k,h}(x)\min\left\{\left(\left\langle \hat{P}_k(x)-P(x),V_{h+1}^*\right\rangle + b_{k,h}(x)\right.\right.$$
$$\left.\left. + \left\langle \hat{P}_k(x)-P(x),\overline{V}_{k,h+1}-V_{h+1}^*\right\rangle + 2\beta_k(x)\right),H\right\},$$

$$\leq \sum_{k=1}^{K}\sum_{h=1}^{H}\sum_{x\in L_k} w_{k,h}(x)\left(\left\langle \hat{P}_k(x)-P(x),V_{h+1}^*\right\rangle + b_{k,h}(x)+\left\langle \hat{P}_k(x)-P(x),\overline{V}_{k,h+1}-V_{h+1}^*\right\rangle\right)$$

$$+ \sum_{k=1}^{K}\sum_{h=1}^{H}\sum_{x\in\Lambda_k} w_{k,h}(x)2\beta_k(x) + \sum_{k=1}^{K}\sum_{h=1}^{H}\sum_{x\notin L_k} w_{k,h}(x)H + \sum_{k=1}^{K}\sum_{h=1}^{H}\sum_{x\notin\Lambda_k} w_{k,h}(x)H$$

$$\leq \sum_{k=1}^{K}\sum_{h=1}^{H}\sum_{x\in L_k} w_{k,h}(x)\left(\left\langle \hat{P}_k(x)-P(x),V_{h+1}^*\right\rangle + b_{k,h}(x)+\left\langle \hat{P}_k(x)-P(x),\overline{V}_{k,h+1}-V_{h+1}^*\right\rangle\right)$$

$$+ \sum_{k=1}^{K}\sum_{h=1}^{H}\sum_{x\in\Lambda_k} w_{k,h}(x)2\beta_k(x) + 8H^2\sum_{i=1}^{m} X[I_i]L + 8H^2\sum_{i=1}^{l} X[J_i]L. \qquad \square$$

## A.5 Bounds on the individual terms in regret

To prove the following bounds on the individual terms in regret (Lemma 18), we heavily use the lemmas derived from the notion of the good sets (Section A.3).

**Lemma 19** (Cumulative transition estimation error, Hoeffding-style). *For F-UCBVI, outside the failure event $\mathcal{F}$, the sum over time of the transition estimation error satisfies that*

$$\sum_{k=1}^{K}\sum_{h=1}^{H}\sum_{x\in L_k} w_{k,h}(x)\langle \hat{P}_k(x)-P(x),V_{h+1}^*\rangle \leq \sum_{i=1}^{m} H\sqrt{2X[I_i]TL} + 4m^2 H \max_i S_i \max_i X_i L^2.$$

*Proof.* Outside the failure event $\mathcal{F}$, by lemma 9,

$$\sum_{k=1}^{K}\sum_{h=1}^{H}\sum_{x\in L_k} w_{k,h}(x)\left\langle \hat{P}_k(x)-P(x),V_{h+1}^*\right\rangle$$

$$\leq \sum_{k=1}^{K}\sum_{h=1}^{H}\sum_{x\in L_k} w_{k,h}(x)\left(\sum_{i=1}^{m} H\sqrt{\frac{L}{2N_{i,k}(x)}} + \sum_{i=1}^{m}\sum_{j=i+1}^{m} 2HL\sqrt{\frac{S_iS_j}{N_{i,k}(x)N_{j,k}(x)}}\right), \quad \text{(A.12)}$$

For the first term in (A.12), by the Cauchy-Schwarz inequality and Lemma 15,

$$\sum_{k=1}^{K}\sum_{h=1}^{H}\sum_{x\in L_k} w_{k,h}(x)\left(\sum_{i=1}^{m} H\sqrt{\frac{L}{2N_{i,k}(x)}}\right)$$

$$\leq H\sqrt{\frac{L}{2}}\sum_{i=1}^{m}\sqrt{\sum_{k=1}^{K}\sum_{h=1}^{H}\sum_{x\in L_k} w_{k,h}(x)}\cdot\sqrt{\sum_{k=1}^{K}\sum_{h=1}^{H}\sum_{x\in L_k}\frac{w_{k,h}(x)}{N_{i,k}(x)}}$$

$$\leq \sum_{i=1}^{m} H\sqrt{2X[I_i]TL}.$$

For the second term in (A.12), by Lemma 16,

$$
\sum_{k=1}^{K}\sum_{h=1}^{H}\sum_{x\in L_k} w_{k,h}(x)\left(\sum_{i=1}^{m}\sum_{j=i+1}^{m} 2HL\sqrt{\frac{S_iS_j}{N_{i,k}(x)N_{j,k}(x)}}\right)
$$

$$
\leq m^2 HL \max_i S_i \cdot 4\sqrt{X[I_i]X[I_j]L}
$$

$$
\leq 4m^2 H \max_i S_i \max_i X_i L^2.
$$

Substituting the above two bounds into (A.12) completes the proof. $\qquad\square$

**Lemma 20** (Cumulative transition optimism, Hoeffding-style). *For F-UCBVI, outside the failure event $\mathcal{F}$, the sum over time of the transition optimism satisfies that*

$$
\sum_{k=1}^{K}\sum_{h=1}^{H}\sum_{x\in L_k} w_{k,h}(x)b_k(x) \leq \sum_{i=1}^{m} H\sqrt{2X[I_i]TL} + 4m^2 H \max_i S_i \max_i X_i L^2.
$$

*Proof.* The proof is exactly the same as that of Lemma 19 by noting

$$
b_k(x) = \sum_{i=1}^{m} H\sqrt{\frac{L}{2N_{i,k}(x)}} + \sum_{i=1}^{m}\sum_{j=i+1}^{m} 2HL\sqrt{\frac{S_iS_j}{N_{i,k}(x)N_{j,k}(x)}}\,\square
$$

**Lemma 21** (Cumulative reward estimation error and optimism, Hoeffding-style). *For F-UCBVI, outside the failure event $\mathcal{F}$, the sum over time of the reward estimation error and optimism satisfies that*

$$
\sum_{k=1}^{K}\sum_{h=1}^{H}\sum_{x\in \Lambda_k} w_{k,h}(x)2\beta_k(x) \leq \sum_{i=1}^{l} 2\sqrt{2X[J_i]TL}
$$

*Proof.* Outside the failure event $\mathcal{F}$, by the Cauchy-Schwarz inequality,

$$
\sum_{k=1}^{K}\sum_{h=1}^{H}\sum_{x\in \Lambda_k} w_{k,h}(x)\beta_k(x) = \sum_{k=1}^{K}\sum_{h=1}^{H}\sum_{x\in \Lambda_k} w_{k,h}(x)\sum_{i=1}^{l}\sqrt{\frac{L}{2M_{i,k}(x)}}
$$

$$
\leq \sqrt{\frac{L}{2}}\sum_{i=1}^{l}\sqrt{\sum_{k=1}^{K}\sum_{h=1}^{H}\sum_{x\in \Lambda_k}\frac{w_{k,h}(x)}{M_{i,k}(x)}}\sqrt{\sum_{k=1}^{K}\sum_{h=1}^{H}\sum_{x\in \Lambda_k} w_{k,h}(x)}
$$

$$
\leq \sum_{i=1}^{l}\sqrt{2X[J_i]TL},
$$

where the last inequality is due to Lemma 15. $\qquad\square$

**Lemma 22** (Cumulative correction term, Hoeffding-style). *For F-UCBVI, outside the failure event $\mathcal{F}$, the sum over time of the correction term satisfies that*

$$
\sum_{k=1}^{K}\sum_{h=1}^{H}\sum_{x\in L_k} w_{k,h}(x)\left\langle \hat{P}_k(x) - P(x), \overline{V}_{k,h+1} - V_{h+1}^*\right\rangle
$$

$$
\leq 45m^3 H^2 (\max_i S_i)^{1.5}\max_i X[I_i]L^{2.5} + 14mlH^{1.5}(\max_i S_i)^{0.5}(\max_i X[I_i])^{0.5}(\max_i X[J_i])^{0.5}L^2.
$$

*Proof.* Since $\overline{V}_{k,h+1}$ is a random vector, we cannot apply scalar concentration as in bounding the transition estimation error (Lemma 9). However, some techniques there are useful here, including the inverse telescoping technique and the Holder's argument (Lemma 10).

Omitting the dependence of $\hat{P}_k(x), P(x), \hat{P}_{i,k}(x), P_i(x)$ on $x$, for any fixed $i$ and $s'[i] \in \mathcal{S}_i$, by the inverse telescoping technique,

$$\left\langle \hat{P}_k - P, \overline{V}_{k,h+1} - V^*_{h+1} \right\rangle$$

$$= \left\langle \prod_{i=1}^m \hat{P}_{i,k} - \prod_{i=1}^m P_i, \overline{V}_{k,h+1} - V^*_{h+1} \right\rangle$$

$$= \left\langle \sum_{i=1}^m (\hat{P}_{i,k} - P_i) P_{1:i-1} \hat{P}_{i+1:m,k}, \overline{V}_{k,h+1} - V^*_{h+1} \right\rangle$$

$$= \sum_{i=1}^m \left\langle \hat{P}_{i,k} - P_i, \mathbb{E}_{P_{1:i-1}} \mathbb{E}_{\hat{P}_{i+1:m,k}} [\overline{V}_{k,h+1} - V^*_{h+1}] \right\rangle$$

$$= \sum_{i=1}^m \left\langle \hat{P}_{i,k} - P_i, \mathbb{E}_{P_{1:i-1}} \mathbb{E}_{P_{i+1:m}} [\overline{V}_{k,h+1} - V^*_{h+1}] \right\rangle \tag{A.13}$$

$$+ \sum_{i=1}^m \left\langle \hat{P}_{i,k} - P_i, \mathbb{E}_{P_{1:i-1}} (\mathbb{E}_{\hat{P}_{i+1:m,k}} - \mathbb{E}_{P_{i+1:m}}) [\overline{V}_{k,h+1} - V^*_{h+1}] \right\rangle. \tag{A.14}$$

For (A.13), outside the failure event $\mathcal{F}$ (specifically, $\mathcal{F}_6$),

$$\left\langle \hat{P}_{i,k} - P_i, \mathbb{E}_{P_{1:i-1}} \mathbb{E}_{P_{i+1:m}} [\overline{V}_{k,h+1} - V^*_{h+1}] \right\rangle$$

$$\leq \sum_{s'[i] \in S_i} \left( \frac{2L}{3N_{i,k}(x)} + \sqrt{\frac{2P_i(s'[i]|x)L}{N_{i,k}(x)}} \right) \mathbb{E}_{P_{1:i-1}} \mathbb{E}_{P_{i+1:m}} [\overline{V}_{k,h+1} - V^*_{h+1}]$$

$$\leq \frac{2HS_iL}{3N_{i,k}(x)} + \sum_{s'[i] \in S_i} \sqrt{\frac{2P_i(s'[i]|x)L}{N_{i,k}(x)}} \mathbb{E}_{P_{-i}} [\overline{V}_{k,h+1} - V^*_{h+1}],$$

the sum over time of which are both bounded, respectively, by

$$\sum_{k=1}^K \sum_{h=1}^H \sum_{x \in L_k} w_{k,h}(x) \frac{2HS_iL}{3N_{i,k}(x)} \leq \frac{8}{3} HS_i X[I_i] L^2 \leq 3HS_i X[I_i] L^2,$$

and

$$\sum_{k=1}^K \sum_{h=1}^H \sum_{x \in L_k} w_{k,h}(x) \sum_{s'[i] \in S_i} \sqrt{\frac{2P_i(s'[i]|x)L}{N_{i,k}(x)}} \mathbb{E}_{P_{-i}} [\overline{V}_{k,h+1} - V^*_{h+1}]$$

$$\leq \sqrt{2S_iL} \sum_{k=1}^K \sum_{h=1}^H \sum_{x \in L_k} w_{k,h}(x) \sqrt{\frac{\mathbb{E}_{P_i} (\mathbb{E}_{P_{-i}} [\overline{V}_{k,h+1} - V^*_{h+1}])^2}{N_{i,k}(x)}}$$

$$\leq \sqrt{2S_iL} \sqrt{\sum_{k=1}^K \sum_{h=1}^H \sum_{x \in L_k} \frac{w_{k,h}(x)}{N_{i,k}(x)}} \cdot \sqrt{\sum_{k=1}^K \sum_{h=1}^H \sum_{x \in L_k} w_{k,h}(x) \mathbb{E}_{P_i} (\mathbb{E}_{P_{-i}} [\overline{V}_{k,h+1} - V^*_{h+1}])^2}$$

$$\leq \sqrt{2S_iL} \cdot 2\sqrt{X[I_i]L} \cdot \sqrt{G_0}$$

$$\leq 41 m^2 H^2 (\max_i S_i)^{1.5} \max_i X[I_i] L^{2.5} + 14l H^{1.5} (\max_i S_i)^{0.5} (\max_i X[I_i])^{0.5} (\max_i X[J_i])^{0.5} L^2,$$

where the first and second inequalities are due to the Cauchy-Schwarz inequality, and the third inequality is due to Lemma 15 and Lemma 17. With the same Holder's argument as in Lemma 10, (A.14) is upper bounded by

$$\sum_{i=1}^m \left\langle \hat{P}_{i,k} - P_i, \mathbb{E}_{P_{1:i-1}} (\mathbb{E}_{\hat{P}_{i+1:m,k}} - \mathbb{E}_{P_{i+1:m}}) [\overline{V}_{k,h+1} - V^*_{h+1}] \right\rangle$$

$$\leq \sum_{i=1}^m \sum_{j=i+1}^m 2HL \sqrt{\frac{S_i S_j}{N_{i,k}(x) N_{j,k}(x)}},$$

the sum over time of which is upper bounded by $4m^2 H \max_i S_i \max_i X_i L^2$ due to Lemma 16. $\square$

### A.6 Regret bounds (proof of Theorem 1)

*Proof.* Outside the failure event $\mathcal{F}$, combining Lemmas 18, 19, 20, 21 and 22, we obtain

$$
\begin{aligned}
\text{Regret}(K) &\leq 2\left(\sum_{i=1}^{m} H\sqrt{2X[I_i]TL} + 4m^2 H \max_i S_i \max_i X_i L^2\right) \\
&\quad + 45m^3 H^2 (\max_i S_i)^{1.5} \max_i X[I_i] L^{2.5} \\
&\quad + 14ml H^{1.5} (\max_i S_i)^{0.5} (\max_i X[I_i])^{0.5} (\max_i X[J_i])^{0.5} L^2 + 4\sum_{i=1}^{l} \sqrt{X[J_i]TL} \\
&\quad + 8\sum_{i=1}^{m} X[I_i] H^2 L + 8\sum_{i=1}^{l} X[J_i] H^2 L \\
&\leq 3\sum_{i=1}^{m} H\sqrt{X[I_i]TL} + 4\sum_{i=1}^{l} \sqrt{X[J_i]TL} + 53m^3 H^2 (\max_i S_i)^{1.5} \max_i X[I_i] L^{2.5} \\
&\quad + 22ml H^2 (\max_i S_i)^{0.5} (\max_i X[I_i])^{0.5} \max_i X[J_i] L^2 \\
&= \tilde{\mathcal{O}}\left(\sum_{i=1}^{m} \sqrt{H^2 X[I_i]T} + \sum_{i=1}^{l} \sqrt{X[J_i]T}\right),
\end{aligned}
$$

where in the last equality we assume that $T \geq \text{poly}(m, l, \max_i S_i, \max_i X[I_i], H)$.

To accommodate the case of known rewards, it suffices to remove the parts related to reward estimation and reward bonuses in both the algorithm and the analysis, which yields the regret bound $\tilde{\mathcal{O}}(\sum_{i=1}^{m} \sqrt{H^2 X[I_i]T})$. $\qquad\square$

# B  Regret analysis of F-EULER

The basic structure of the regret analysis in this section is the same as that of F-UCBVI (Section A), e.g., the notion of the good sets, including the definitions and lemmas (specifically, Definition 5 and Lemmas 13, 14, 15 and 16), carries through here. The key difference is a more refined analysis using Bernstein-style concentrations (Lemmas 38 and 39).

## B.1  Failure event

Recall that $L = \log(16mlSXT/\delta)$. We define the failure event for F-EULER as follows.

**Definition 6** (Failure events). *Define the events*

$$\mathcal{B}_1 := \Bigg\{ \exists (i \in [m], k \in [K], h \in [H], x \in \mathcal{X}),$$

$$\left| \left\langle \hat{P}_{i,k}(x) - P_i(x), \mathbb{E}_{P_{-i}(x)}[V_{h+1}^*] \right\rangle \right| > \sqrt{\frac{2\mathrm{Var}_{P_i(x)}\mathbb{E}_{P_{-i}(x)}[V_{h+1}^*]L}{N_{i,k}(x)}} + \frac{2HL}{3N_{i,k}(x)} \Bigg\},$$

$$\mathcal{B}_2 := \mathcal{F}_2,$$

$$\mathcal{B}_3 := \left\{ \exists (i \in [l], k \in [K], x \in \mathcal{X}), \quad \left| \hat{R}_{i,k}(x) - R_i(x) \right| > \sqrt{\frac{2\mathbb{S}[\hat{r}_i(x)]L}{M_{i,k}(x)}} + \frac{14L}{3M_{i,k}(x)} \right\},$$

$$\mathcal{B}_4 := \Bigg\{ \exists (i \in [m], k \in [K], h \in [H], x \in \mathcal{X}, s[-i] \in S_{-i}),$$

$$\left| \left\langle \hat{P}_{i,k}(x) - P_i(x), V_{h+1}^*(s[-i]) \right\rangle \right| > H \sqrt{\frac{L}{2N_{i,k}(x)}} \Bigg\},$$

$$\mathcal{B}_5 := \Bigg\{ \exists (i \in [m], k \in [K], h \in [H], x \in \mathcal{X}),$$

$$\left| \sqrt{\mathrm{Var}_{\hat{P}_{i,k}(x)}\mathbb{E}_{P_{-i}(x)}[V_h^*]} - \sqrt{\mathrm{Var}_{P_i(x)}\mathbb{E}_{P_{-i}(x)}[V_h^*]} \right| > 3H\sqrt{\frac{L}{N_{i,k}(x)}} \Bigg\},$$

$$\mathcal{B}_6 := \left\{ \exists (i \in [m], k \in [K], h \in [H], x \in \mathcal{X}), \quad \left| \sqrt{\mathbb{S}[\hat{r}_i(x)]} - \sqrt{\mathrm{Var}(r_i(x))} \right| > \sqrt{\frac{4L}{M_{i,k}(x)}} \right\},$$

$$\mathcal{B}_7 := \mathcal{F}_4, \quad \mathcal{B}_8 := \mathcal{F}_5, \quad \mathcal{B}_9 := \mathcal{F}_6,$$

*where in $\mathcal{B}_3$ and $\mathcal{B}_5$ we assume $N_{i,k} \geq 2$ (true for $x$ in the good set $L_k$), and in $\mathcal{B}_6$ we assume $M_{i,k} \geq 2$ (true for $x$ in the good set $\Lambda_k$). Then the failure event for F-EULER is defined by $\mathcal{B} := \bigcup_{i=1}^{9} \mathcal{B}_i$.*

The following lemma shows that the failure event $\mathcal{B}$ happens with low probability.

**Lemma 23** (Failure probability). *For any FMDP specified by* (2.1)*, during the running of F-EULER for $K$ episodes, the failure event $\mathcal{B}$ happens with probability at most $\delta$.*

*Proof.* By Bernstein's inequality (Lemma 38) and the union bound, $\mathcal{B}_1$ happens with probability at most $\delta/8$. By empirical Bernstein's inequality (Lemma 39) [27] and the union bound, $\mathcal{B}_3$ happens with probability at most $\delta/8$, where $1/(N_{i,k}(x) - 1)$ is replaced by $2/N_{i,k}(x)$ for $N_{i,k}(x) \geq 2$. By Hoeffding's inequality (Lemma 37) and the union bound, $\mathcal{B}_4$ happens with probability at most $\delta/8$. By Theorem 10 in [27] and the union bound, $\mathcal{B}_6$ happens with probability at most $\delta/8$, where $1/(M_{i,k}(x) - 1)$ is replaced by $2/M_{i,k}(x)$ for $M_{i,k}(x) \geq 2$. By the proof of Lemma 8, $\mathcal{B}_2$ and $\mathcal{B}_9$ happen with probability at most $\delta/8$, respectively; $\mathcal{B}_7$ and $\mathcal{B}_8$ happen with probability at most $\delta/16$, respectively. The argument on $\mathcal{B}_5$ is more involved, which is stated as follows.

For $k \in [K], h \in [H], x \in \mathcal{X}$, let $\mathbb{S}_{\hat{P}_{i,k}(x)}\mathbb{E}_{P_{-i}(x)}[V_h^*]$ denote the sample variance of $\mathbb{E}_{P_{-i}(x)}[V_h^*]$, whose relationship with the empirical variance is given by

$$\mathbb{S}_{\hat{P}_{i,k}(x)}\mathbb{E}_{P_{-i}(x)}[V_h^*] = \frac{N_{i,k}(x)}{N_{i,k}(x) - 1}\mathrm{Var}_{\hat{P}_{i,k}(x)}\mathbb{E}_{P_{-i}(x)}[V_h^*].$$

Hence, for $N_{i,k}(x) \geq 2$,

$$\mathbb{S}_{\hat{P}_{i,k}(x)}\mathbb{E}_{P_{-i}(x)}[V_h^*] - \mathrm{Var}_{\hat{P}_{i,k}(x)}\mathbb{E}_{P_{-i}(x)}[V_h^*] = \frac{1}{N_{i,k}(x) - 1}\mathrm{Var}_{\hat{P}_{i,k}(x)}\mathbb{E}_{P_{-i}(x)}[V_h^*] \leq \frac{2H^2}{N_{i,k}(x)}.$$

By Theorem 10 in [27] and the union bound, with probability at least $1 - \delta/8$, for all $i \in [m], k \in [K], h \in [H], x \in \mathcal{X}$ and $N_{i,k}(x) \geq 2$,

$$\left|\sqrt{\mathbb{S}_{\hat{P}_{i,k}(x)}\mathbb{E}_{P_{-i}(x)}[V_h^*]} - \sqrt{\mathrm{Var}_{P_i(x)}\mathbb{E}_{P_{-i}(x)}[V_h^*]}\right| \leq H\sqrt{\frac{2L}{N_{i,k}(x) - 1}} \leq H\sqrt{\frac{4L}{N_{i,k}(x)}},$$

which yields that

$$\left|\sqrt{\mathrm{Var}_{\hat{P}_{i,k}(x)}\mathbb{E}_{P_{-i}(x)}[V_h^*]} - \sqrt{\mathrm{Var}_{P_i(x)}\mathbb{E}_{P_{-i}(x)}[V_h^*]}\right|$$
$$\leq \left|\sqrt{\mathrm{Var}_{\hat{P}_{i,k}(x)}\mathbb{E}_{P_{-i}(x)}[V_h^*]} - \sqrt{\mathbb{S}_{\hat{P}_{i,k}(x)}\mathbb{E}_{P_{-i}(x)}[V_h^*]}\right.$$
$$\left. + \sqrt{\mathbb{S}_{\hat{P}_{i,k}(x)}\mathbb{E}_{P_{-i}(x)}[V_h^*]} - \sqrt{\mathrm{Var}_{P_i(x)}\mathbb{E}_{P_{-i}(x)}[V_h^*]}\right|$$
$$\leq \sqrt{\frac{2H^2}{N_{i,k}(x)}} + H\sqrt{\frac{4L}{N_{i,k}(x)}} \leq 3H\sqrt{\frac{L}{N_{i,k}(x)}},$$

where in the second inequality we use $|\sqrt{a} - \sqrt{b}| \leq \sqrt{|a - b|}$ for all $a, b \geq 0$ and in the third inequality we use $L = \log(16mlSXT/\delta) \geq 2$. Therefore, $\mathcal{B}_5$ happens with probability at most $\delta/8$.

Finally, applying the union bound on $\mathcal{B}_i$ for $i \in [9]$ yields that the failure event $\mathcal{F}$ happens with probability at most $\delta$. $\qquad\square$

The deduction in the rest of this section and hence the regret bound hold outside the failure event $\mathcal{B}$, with probability at least $1 - \delta$. From the above derivation, note that we can actually use a smaller

$$L_1 = \log(16ml\max\{S\max_i X[I_i], \max_i X[J_i]\}T/\delta)$$

to replace $L$ for F-EULER. We use $L$ for simplicity.

## B.2 Upper confidence bound on the optimal state-value function

The goal of this section is to show that $\overline{V}_{k,h}$ is an entrywise UCB of $V_h^*$ for all $k \in [K], h \in [H]$. Analogously to the analysis of F-UCBVI, this optimism is achieved by setting the transition (reward, respectively) bonus as the UCB of the transition (reward, respectively) estimation error. While the reward estimation error is direct to bound (failure event $\mathcal{B}_3$), to bound the transition estimation error, we need some properties of the relevant functions.

Recall that in (4.1), for $i \in [m], P_i \in \Delta(\mathcal{S}_i), P = \prod_{i=1}^m P_i \in \Delta(\mathcal{S})$ and $V \in \mathbb{R}^{\mathcal{S}}$, we define $g_i(P, V) := 2\sqrt{L}\sqrt{\mathrm{Var}_{P_i}\mathbb{E}_{P_{-i}}[V]}$. Now for $i \in [m], k \in [K], P \in \Delta(\mathcal{S}), V \in \mathbb{R}^{\mathcal{S}}$ and $x \in \mathcal{X}$, we define

$$\phi_{i,k}(P, V, x) := \frac{g_i(P, V)}{\sqrt{N_{i,k}(x)}} + \frac{2HL}{3N_{i,k}(x)}.$$

The following lemma establishes some "Lipschitzness" properties of $g_i$, which are then used to prove a similar property of $\phi_{i,k}$.

**Lemma 24** (Properties of $g_i$). *Outside the failure event $\mathcal{B}$, for any index $i \in [m]$, episode $k \in [K]$, step $h \in [H]$, state-action pair $x \in \mathcal{X}$ and vectors $V_1, V_2 \in \mathbb{R}^S$,*

$$|g_i(P(x), V_1) - g_i(P(x), V_2)| \leq \sqrt{2L}\|V_1 - V_2\|_{2,P(x)}, \tag{B.1}$$

$$\left|g_i(\hat{P}_k(x), V_h^*) - g_i(P(x), V_h^*)\right| \leq 3\sqrt{2}HL\sum_{j=1}^m \frac{1}{\sqrt{N_{j,k}(x)}}. \tag{B.2}$$

*Proof.* For any finite set $\mathcal{S}$, $P \in \Delta(\mathcal{S})$ and vectors $V_1, V_2 \in \mathbb{R}^{\mathcal{S}}$, we have that $\sqrt{\operatorname{Var}_P[V_1]} \leq \sqrt{\operatorname{Var}_P[V_2]} + \sqrt{\operatorname{Var}_P[V_1 - V_2]}$ (see [44, Section D.3] for a proof). Then for the first property (B.1),

$$
\begin{aligned}
|g_i(P(x), V_1) - g_i(P(x), V_2)| &\leq \sqrt{2L}\sqrt{\operatorname{Var}_{P_i(x)}\mathbb{E}_{P_{1:i-1}(x)}\mathbb{E}_{P_{i+1:m}(x)}[V_1 - V_2]} \\
&\leq \sqrt{2L}\sqrt{\mathbb{E}_{P_i(x)}\left(\mathbb{E}_{P_{1:i-1}(x)}\mathbb{E}_{P_{i+1:m}(x)}[V_1 - V_2]\right)^2} \\
&\leq \sqrt{2L}\sqrt{\mathbb{E}_P[(V_1 - V_2)^2]} := \sqrt{2L}\|V_1 - V_2\|_{2,P(x)}.
\end{aligned}
$$

For the second property (B.2),

$$
\begin{aligned}
&\left| g_i(\hat{P}_k(x), V_h^*) - g_i(P(x), V_h^*) \right| \\
=& \sqrt{2L}\left| \sqrt{\operatorname{Var}_{\hat{P}_{i,k}(x)}\mathbb{E}_{\hat{P}_{-i,k}(x)}[V_h^*]} - \sqrt{\operatorname{Var}_{P_i(x)}\mathbb{E}_{P_{-i}(x)}[V_h^*]} \right| \\
=& \sqrt{2L}\left| \sqrt{\operatorname{Var}_{\hat{P}_{i,k}(x)}\mathbb{E}_{\hat{P}_{-i,k}(x)}[V_h^*]} - \sqrt{\operatorname{Var}_{\hat{P}_{i,k}(x)}\mathbb{E}_{P_{-i}(x)}[V_h^*]} \right. \\
&\left. + \sqrt{\operatorname{Var}_{\hat{P}_{i,k}(x)}\mathbb{E}_{P_{-i}(x)}[V_h^*]} - \sqrt{\operatorname{Var}_{P_i(x)}\mathbb{E}_{P_{-i}(x)}[V_h^*]} \right| \\
\leq& \sqrt{2L}\left| \sqrt{\operatorname{Var}_{\hat{P}_{i,k}(x)}\mathbb{E}_{\hat{P}_{-i,k}(x)}[V_h^*]} - \sqrt{\operatorname{Var}_{\hat{P}_{i,k}(x)}\mathbb{E}_{P_{-i}(x)}[V_h^*]} \right| \tag{B.3} \\
&+ \sqrt{2L}\left| \sqrt{\operatorname{Var}_{\hat{P}_{i,k}(x)}\mathbb{E}_{P_{-i}(x)}[V_h^*]} - \sqrt{\operatorname{Var}_{P_i(x)}\mathbb{E}_{P_{-i}(x)}[V_h^*]} \right|. \tag{B.4}
\end{aligned}
$$

To bound (B.3), omitting the dependence of $\hat{P}_{i,k}(x)$, $P_i(x)$ on $x$, outside the failure event $\mathcal{B}$ (specifically, $\mathcal{B}_4$), by an inverse telescoping argument,

$$
\begin{aligned}
&\left| \sqrt{\operatorname{Var}_{\hat{P}_{i,k}}\mathbb{E}_{\hat{P}_{-i,k}}[V_h^*]} - \sqrt{\operatorname{Var}_{\hat{P}_{i,k}}\mathbb{E}_{P_{-i}}[V_h^*]} \right| \\
\leq& \sum_{j=1, j\neq i}^m \left| \sqrt{\operatorname{Var}_{\hat{P}_{i,k}}\mathbb{E}_{P_{1:j-1\setminus i,k}}\mathbb{E}_{\hat{P}_{j:m\setminus i,k}}[V_h^*]} - \sqrt{\operatorname{Var}_{\hat{P}_{i,k}}\mathbb{E}_{P_{1:j\setminus i,k}}\mathbb{E}_{\hat{P}_{j+1:m\setminus i,k}}[V_h^*]} \right| \\
\leq& \sum_{j=1, j\neq i}^m \sqrt{\operatorname{Var}_{\hat{P}_{i,k}}\mathbb{E}_{P_{1:j-1\setminus i}}\mathbb{E}_{\hat{P}_{j+1:m\setminus i,k}}(\mathbb{E}_{\hat{P}_{j,k}} - \mathbb{E}_{P_j})[V_h^*]} \\
\leq& \sum_{j=1, j\neq i}^m \sqrt{\mathbb{E}_{\hat{P}_{i,k}}\left(\mathbb{E}_{P_{1:j-1\setminus i}}\mathbb{E}_{\hat{P}_{j+1:m\setminus i,k}}(\mathbb{E}_{\hat{P}_{j,k}} - \mathbb{E}_{P_j})[V_h^*]\right)^2} \leq \sum_{j=1, j\neq i}^m H\sqrt{\frac{L}{2N_{j,k}(x)}},
\end{aligned}
$$

where "$\cdot \setminus i$" denotes excluding $i$. For (B.4), outside the failure event $\mathcal{B}$ (specifically, $\mathcal{B}_5$),

$$
\left| \sqrt{\operatorname{Var}_{\hat{P}_{i,k}(x)}\mathbb{E}_{P_{-i}(x)}[V_h^*]} - \sqrt{\operatorname{Var}_{P_i(x)}\mathbb{E}_{P_{-i}(x)}[V_h^*]} \right| \leq 3H\sqrt{\frac{L}{N_{i,k}(x)}},
$$

Combining the above bounds on (B.3) and (B.4) yields

$$
\left| g_i(\hat{P}_k(x), V_h^*) - g_i(P(x), V_h^*) \right| \leq 3\sqrt{2}HL\sum_{j=1}^m \frac{1}{\sqrt{N_{j,k}(x)}}.
$$

$\square$

Then $\phi_{i,k}$ satisfies the following "Lipschitzness" property.

**Lemma 25** (Property of $\phi_{i,k}$). *Outside the failure event $\mathcal{B}$, for any index $i \in [m]$, episode $k \in [K]$, step $h \in [H]$, state-action pair $x \in \mathcal{X}$ and vector $V \in \mathbb{R}^S$,*

$$
\begin{aligned}
&\left| \phi_{i,k}(\hat{P}_k(x), V, x) - \phi_{i,k}(P(x), V_{h+1}^*, x) \right| \\
\leq& \frac{\sqrt{2L}\left\|V - V_{h+1}^*\right\|_{2,\hat{P}_k(x)}}{\sqrt{N_{i,k}(x)}} + \frac{3\sqrt{2}HL}{\sqrt{N_{i,k}(x)}}\sum_{j=1}^m \frac{1}{\sqrt{N_{j,k}(x)}}.
\end{aligned}
$$

*Proof.* By Lemma 24, outside the failure event $\mathcal{B}$,

$$\left|\phi_{i,k}(\hat{P}_k(x), V, x) - \phi_{i,k}(P(x), V_{h+1}^*, x)\right|$$

$$\leq \frac{1}{\sqrt{N_{i,k}(x)}} \left(\left|g_i(\hat{P}_k(x), V) - g_i(\hat{P}_k(x), V_{h+1}^*)\right| + \left|g_i(\hat{P}_k(x), V_{h+1}^*) - g_i(P(x), V_{h+1}^*)\right|\right)$$

$$\leq \frac{\sqrt{2L}\left\|V - V_{h+1}^*\right\|_{2,\hat{P}_k(x)}}{\sqrt{N_{i,k}(x)}} + \frac{3\sqrt{2}HL}{\sqrt{N_{i,k}(x)}} \sum_{j=1}^m \frac{1}{\sqrt{N_{j,k}(x)}}. \qquad \square$$

Now we are ready to present the bounds on transition estimation error in the following lemma.

**Lemma 26** (Transition estimation error, Bernstein-style). *Outside the failure event $\mathcal{B}$, for any episode $k \in [K]$, step $h \in [H]$ and state-action pair $x \in \mathcal{X}$,*

$$\left|\left\langle \hat{P}_k(x) - P_k(x), V_{h+1}^*\right\rangle\right| \leq \sum_{i=1}^m \phi_{i,k}(P(x), V_{h+1}^*, x) + \sum_{i=1}^m \sum_{j=i+1}^m 2HL\sqrt{\frac{S_i S_j}{N_{i,k}(x) N_{j,k}(x)}}.$$

*And for a given $k \in [K]$ and a given $h \in [H]$, if $\underline{V}_{k,h+1} \leq V_{h+1}^* \leq \overline{V}_{k,h+1}$ entrywise, then the above inequality yields*

$$\left|\left\langle \hat{P}_k(x) - P_k(x), V_{h+1}^*\right\rangle\right|$$

$$\leq \sum_{i=1}^m \phi_{i,k}(\hat{P}_k(x), \overline{V}_{k,h+1}, x) + \sum_{i=1}^m \frac{\sqrt{2L}\left\|\overline{V}_{k,h+1} - \underline{V}_{k,h+1}\right\|_{2,\hat{P}_k(x)}}{\sqrt{N_{i,k}(x)}}$$

$$+ \sum_{i=1}^m \frac{3\sqrt{2}HL}{\sqrt{N_{i,k}(x)}} \sum_{j=1}^m \frac{1}{\sqrt{N_{j,k}(x)}} + \sum_{i=1}^m \sum_{j=i+1}^m 2HL\sqrt{\frac{S_i S_j}{N_{i,k}(x) N_{j,k}(x)}}.$$

*Proof.* Recall that in Lemma 9, for all $k \in [K], h \in [H], x \in \mathcal{X}$, omitting the dependence of $\hat{P}_k(x), P(x), \hat{P}_{i,k}(x), P_i(x)$ on $x$, we decompose the transition estimation error as

$$\left\langle \hat{P}_k - P, V_{h+1}^*\right\rangle = \sum_{i=1}^m \left\langle \hat{P}_{i,k} - P_i, \mathbb{E}_{P_{1:i-1}}\mathbb{E}_{P_{i+1:m}}[V_{h+1}^*]\right\rangle \tag{B.5}$$

$$+ \sum_{i=1}^m \left\langle \hat{P}_{i,k} - P_i, \mathbb{E}_{P_{1:i-1}}(\mathbb{E}_{\hat{P}_{i+1:m,k}} - \mathbb{E}_{P_{i+1:m}})[V_{h+1}^*]\right\rangle. \tag{B.6}$$

Outside the failure event $\mathcal{B}$ (specifically, $\mathcal{B}_1$), (B.5) is bounded by

$$\left|\left\langle \hat{P}_{i,k}(x) - P_i(x), \mathbb{E}_{P_{1:i-1}(x)}\mathbb{E}_{P_{i+1:m}(x)}[V_{h+1}^*]\right\rangle\right|$$

$$\leq \sqrt{\frac{2\mathrm{Var}_{P_i(x)}\mathbb{E}_{P_{1:i-1}(x)}\mathbb{E}_{P_{i+1:m}(x)}[V_{h+1}^*]L}{N_{i,k}(x)}} + \frac{2HL}{3N_{i,k}(x)}$$

$$= \phi_{i,k}(P(x), V_{h+1}^*, x).$$

If $\underline{V}_{k,h+1} \leq V_{h+1}^* \leq \overline{V}_{k,h+1}$ entrywise, then by Lemma 25, (B.5) is further bounded by

$$\phi_{i,k}(P(x), V_{h+1}^*, x)$$

$$\leq \phi_{i,k}(\hat{P}_k(x), \overline{V}_{h+1}, x) + \frac{\sqrt{2L}\left\|\overline{V}_{k,h+1} - V_{h+1}^*\right\|_{2,\hat{P}_k(x)}}{\sqrt{N_{i,k}(x)}} + \frac{3\sqrt{2}HL}{\sqrt{N_{i,k}(x)}} \sum_{j=1}^m \frac{1}{\sqrt{N_{j,k}(x)}}$$

$$\leq \phi_{i,k}(\hat{P}_k(x), \overline{V}_{h+1}, x) + \frac{\sqrt{2L}\left\|\overline{V}_{k,h+1} - \underline{V}_{k,h+1}\right\|_{2,\hat{P}_k(x)}}{\sqrt{N_{i,k}(x)}} + \frac{3\sqrt{2}HL}{\sqrt{N_{i,k}(x)}} \sum_{j=1}^m \frac{1}{\sqrt{N_{j,k}(x)}},$$

For (B.6), the Holder's argument (Lemma 10) yields

$$\left| \left\langle \hat{P}_{i,k}(x) - P_i(x), \mathbb{E}_{P_{1:i-1}(x)} (\mathbb{E}_{\hat{P}_{i+1:m,k}(x)} - \mathbb{E}_{P_{i+1:m}(x)})[V_{h+1}^*] \right\rangle \right|$$
$$\leq \sum_{j=i+1}^{m} 2HL \sqrt{\frac{S_i S_j}{N_{i,k}(x) N_{j,k}(x)}}.$$

Combining the above bounds on (B.5) and (B.6) completes the proof. $\qquad\square$

The first Bernstein-style bound on the transition estimation error (Lemma 26) depends on the unknown $V_{h+1}^*$, which is why we derive the second bound in terms of the UCB $\overline{V}_{k,h}$ and LCB $\underline{V}_{k,h}$. Now simplify the second bound to the form of the transition bonus. Expanding $\phi_{i,k}$ by definition,

$$\sum_{i=1}^{m} \phi_{i,k}(\hat{P}_k(x), \overline{V}_{k,h+1}, x) + \sum_{i=1}^{m} \frac{\sqrt{2L} \left\| \overline{V}_{k,h+1} - \underline{V}_{k,h+1} \right\|_{2,\hat{P}_k(x)}}{\sqrt{N_{i,k}(x)}}$$
$$+ \sum_{i=1}^{m} \frac{3\sqrt{2}HL}{\sqrt{N_{i,k}(x)}} \sum_{j=1}^{m} \frac{1}{\sqrt{N_{j,k}(x)}} + \sum_{i=1}^{m} \sum_{j=i+1}^{m} 2HL \sqrt{\frac{S_i S_j}{N_{i,k}(x) N_{j,k}(x)}}$$
$$= \sum_{i=1}^{m} \frac{g_i(\hat{P}_k(x), \overline{V}_{k,h+1})}{\sqrt{N_{i,k}(x)}} + \sum_{i=1}^{m} \frac{2HL}{3N_{i,k}(x)} + \sum_{i=1}^{m} \frac{\sqrt{2L} \left\| \overline{V}_{k,h+1} - \underline{V}_{k,h+1} \right\|_{2,\hat{P}_k(x)}}{\sqrt{N_{i,k}(x)}}$$
$$+ \sum_{i=1}^{m} \frac{3\sqrt{2}HL}{\sqrt{N_{i,k}(x)}} \sum_{j=1}^{m} \frac{1}{\sqrt{N_{j,k}(x)}} + \sum_{i=1}^{m} \sum_{j=i+1}^{m} 2HL \sqrt{\frac{S_i S_j}{N_{i,k}(x) N_{j,k}(x)}}$$
$$\leq \sum_{i=1}^{m} \frac{g_i(\hat{P}_k(x), \overline{V}_{k,h+1})}{\sqrt{N_{i,k}(x)}} + \sum_{i=1}^{m} \frac{\sqrt{2L} \left\| \overline{V}_{k,h+1} - \underline{V}_{k,h+1} \right\|_{2,\hat{P}_k(x)}}{\sqrt{N_{i,k}(x)}}$$
$$+ \sum_{i=1}^{m} \sum_{j=i+1}^{m} 11HL \sqrt{\frac{S_i S_j}{N_{i,k}(x) N_{j,k}(x)}} + \sum_{i=1}^{m} \frac{5HL}{N_{i,k}(x)},$$

which is precisely the transition bonus $b_{k,h}(x)$ in (4.2). Therefore, for any $k \in [K], h \in [H]$,

$$\left| \left\langle \hat{P}_k(x) - P_k(x), V_{h+1}^* \right\rangle \right| \leq b_{k,h}(x),$$

if $\underline{V}_{k,h+1} \leq V_{h+1}^* \leq \overline{V}_{k,h+1}$ holds entrywise, which we soon prove in Lemma 27. Note that our choice of the reward bonus upper bounds the reward estimation error outside the failure event $\mathcal{B}$ (specifically, $\mathcal{B}_3$), i.e.,

$$\left| \hat{R}_k(x) - R(x) \right| \leq \sum_{i=1}^{l} \left| \hat{R}_{i,k}(x) - R_i(x) \right| \leq \sum_{i=1}^{l} \sqrt{\frac{2\mathbb{S}[\hat{r}_i(x)]L}{M_{i,k}(x)}} + \sum_{i=1}^{l} \frac{14L}{3M_{i,k}(x)} := \beta_k(x).$$

With $x_{k,h} = (s, \pi_k(s,h))$, recall the optimistic and pessimistic value iterations are defined as

$$\overline{V}_{k,h}(s) = \min \left\{ H - h + 1, \hat{R}(x_{k,h}) + \left\langle \hat{P}_k(x_{k,h}), \overline{V}_{k,h+1} \right\rangle + b_{k,h}(x_{k,h}) + \beta_k(x_{k,h}) \right\},$$
$$\underline{V}_{k,h}(s) = \max \left\{ 0, R(x_{k,h}) + \left\langle \hat{P}_k(x_{k,h}), \underline{V}_{k,h+1} \right\rangle - b_{k,h}(x_{k,h}) - \beta_k(x_{k,h}) \right\}.$$

The following lemma indicates that the Bernstein-style bonuses and the above value iterations ensure optimism and pessimism. Specifically, $\overline{V}_{k,h}$ and $\underline{V}_{k,h}$ are entrywise upper and lower confidence bounds of $V_h^*$ for all $k \in [K], h \in [H]$.

**Lemma 27** (Upper-lower confidence bounds). *Outside the failure event $\mathcal{B}$, for the choices of bonuses in (4.2) and (4.3), for any episode $k \in [K]$, step $h \in [H]$ and state $s \in \mathcal{S}$,*

$$\underline{V}_{k,h}(s) \leq V_h^*(s) \leq \overline{V}_{k,h}(s). \tag{B.7}$$

*Proof.* For $h = H+1$, $\underline{V}_{k,H+1}(s) = V^*_{H+1}(s) = \overline{V}_{k,H+1}(s) = 0$ for all $s \in \mathcal{S}, k \in [K]$. We proceed by backward induction. For $h \in [H]$, assume (B.7) holds for $h+1$. The transition bonus then satisfies $|\langle \hat{P}_k(x) - P_k(x), V^*_{h+1}\rangle| \le b_{k,h}(x)$ for all $x \in \mathcal{X}$. For all $s \in \mathcal{S}$, with $x_{k,h} = (s, \pi_k(s,h))$ and $x^*_h = (s, \pi^*(s,h))$, $\overline{V}_{k,h}$ satisfies that

$$\overline{V}_{k,h}(s) - V^*_h(s)$$
$$= \hat{R}(x_{k,h}) + \beta_k(x_{k,h}) + \left\langle \hat{P}_k(x_{k,h}), \overline{V}_{k,h+1}\right\rangle + b_k(x_{k,h}) - R(x^*_h) - \left\langle P(x^*_h), V^*_{h+1}\right\rangle$$
$$\ge \hat{R}(x^*_h) + \beta_k(x^*_h) + \left\langle \hat{P}_k(x^*_h), \overline{V}_{k,h+1}\right\rangle + b_k(x^*_h) - R(x^*_h) - \left\langle P(x^*_h), V^*_{h+1}\right\rangle$$
$$\ge \left\langle \hat{P}_k(x^*_h), \overline{V}_{k,h+1} - V^*_{h+1}\right\rangle + \left\langle \hat{P}_k(x^*_h) - P(x^*_h), V^*_{h+1}\right\rangle + b_k(x^*_h) \ge 0,$$

and $\underline{V}_{k,h}$ satisfies that

$$V^*_h(s) - \underline{V}_{k,h}(s)$$
$$\ge R(x^*_h) + \left\langle P(x^*_h), V^*_{h+1}\right\rangle - \hat{R}_k(x_{k,h}) - \beta_k(x_{k,h}) - \left\langle \hat{P}_k(x_{k,h}), \underline{V}_{k,h+1}\right\rangle + b_{k,h}(x_{k,h})$$
$$\ge R(x_{k,h}) + \left\langle P(x_{k,h}), V^*_{h+1}\right\rangle - \hat{R}_k(x_{k,h}) - \beta_k(x_{k,h}) - \left\langle \hat{P}_k(x_{k,h}), \underline{V}_{k,h+1}\right\rangle + b_{k,h}(x_{k,h})$$
$$\ge \left\langle \hat{P}_k(x_{k,h}), V^*_{h+1} - \underline{V}_{k,h+1}\right\rangle + \left\langle P(x_{k,h}) - \hat{P}_k(x_{k,h}), V^*_{h+1}\right\rangle + b_{k,h}(x_{k,h}) \ge 0.$$

Inductively, $\underline{V}_{k,h} \le V^*_h \le \overline{V}_{k,h}$ holds entrywise for all $k \in [K]$ and $h \in [H]$. $\qquad\square$

For F-EULER, we refer to the difference between the optimistic value function and the pessimistic value function as the *confidence radius*, which we bound in the following lemma. For here and below, we use "$\lesssim, \approx$" to denote "$\le, =$" neglecting constants. Different from the main text, we make lower-order terms explicit in the appendices.

**Lemma 28** (Confidence radius, Bernstein-style). *Let $F_1 := mH \max_i S_i L$ be a lower-order term. Let $s_{k,t} \in \mathcal{S}$ denote the state at step $t$ of episode $k$ and $x_{k,t} = (s_{k,t}, \pi_k(s_{k,t}, t))$. Outside the failure event $\mathcal{B}$, for any episode $k \in [K]$, step $h \in [H]$ and state $s \in \mathcal{S}$, the confidence radius of F-EULER satisfies that*

$$\overline{V}_{k,h}(s) - \underline{V}_{k,h}(s) \lesssim \min\left\{\sum_{t=h}^{H} \mathbb{E}_{\pi_k}\left[\sum_{i=1}^{m} \frac{F_1}{\sqrt{N_{i,k}(x_{k,t})}} + \sum_{i=1}^{l} \frac{L}{\sqrt{M_{i,k}(x_{k,t})}}\,\middle|\, s_{k,h} = s\right], H\right\}.$$

*Proof.* By definition, for any $k \in [K], h \in [H], s \in \mathcal{S}$,

$$\overline{V}_{k,h}(s) - \underline{V}_{k,h}(s)$$
$$\le \hat{R}_k(x_{k,h}) + \left\langle \hat{P}_k(x_{k,h}), \overline{V}_{k,h+1}\right\rangle + b_{k,h}(x_{k,h}) + \beta_k(x_{k,h})$$
$$\quad - \hat{R}_k(x_{k,h}) - \left\langle \hat{P}_k(x_{k,h}), \underline{V}_{k,h+1}\right\rangle + b_{k,h}(x_{k,h}) + \beta_k(x_{k,h})$$
$$= \left\langle \hat{P}_k(x_{k,h}), \overline{V}_{k,h+1} - \underline{V}_{k,h+1}\right\rangle + 2b_{k,h}(x_{k,h}) + 2\beta_k(x_{k,h})$$
$$= \left\langle P(x_{k,h}), \overline{V}_{k,h+1} - \underline{V}_{k,h+1}\right\rangle + \left\langle P(x_{k,h}) - \hat{P}_k(x_{k,h}), \overline{V}_{k,h+1} - \underline{V}_{k,h+1}\right\rangle$$
$$\quad + 2b_{k,h}(x_{k,h}) + 2\beta_k(x_{k,h})$$
$$\le \left\langle P(x_{k,h}), \overline{V}_{k,h+1} - \underline{V}_{k,h+1}\right\rangle + \sum_{i=1}^{m} H\sqrt{\frac{2S_i L}{N_{i,k}(x_{k,h})}} + 2b_{k,h}(x_{k,h}) + 2\beta_k(x_{k,h}), \qquad \text{(B.8)}$$

where the last inequality results from Holder's inequality and holds outside the failure event $\mathcal{B}$ (specifically, $\mathcal{B}_2$). We apply the following loose bounds on the transition bonus and the reward bonus

that for any $k \in [K], h \in [H], x \in \mathcal{X}$,

$$
\begin{aligned}
b_{k,h}(x) &= \sum_{i=1}^{m} \frac{g_i(\hat{P}_k(x), \overline{V}_{k,h+1})}{\sqrt{N_{i,k}(x)}} + \sum_{i=1}^{m} \frac{\sqrt{2L} \left\| \overline{V}_{k,h+1} - \underline{V}_{k,h+1} \right\|_{2, \hat{P}_k(x)}}{\sqrt{N_{i,k}(x)}} \\
&\quad + \sum_{i=1}^{m} \sum_{j=i+1}^{m} 8HL \sqrt{\frac{S_i S_j}{N_{i,k}(x) N_{j,k}(x)}} + \sum_{i=1}^{m} \frac{4HL}{N_{i,k}(x)} \\
&\leq \frac{2\sqrt{L} \sqrt{\mathrm{Var}_{\hat{P}_{i,k}} \mathbb{E}_{\hat{P}_{-i,k}}[\overline{V}_{k,h+1}]}}{\sqrt{N_{i,k}(x)}} + \sum_{i=1}^{m} \frac{\sqrt{2L}H}{\sqrt{N_{i,k}(x)}} \\
&\quad + \sum_{i=1}^{m} \sum_{j=i+1}^{m} 8HL \max_i S_i \frac{1}{\sqrt{N_{i,k}(x)}} + \sum_{i=1}^{m} \frac{4HL}{N_{i,k}(x)} \\
&\lesssim mH \max_i S_i L \sum_{i=1}^{m} \frac{1}{\sqrt{N_{i,k}(x)}},
\end{aligned}
$$

and

$$
\beta_k(x) = \sum_{i=1}^{l} \sqrt{\frac{2\mathbb{S}[\hat{r}_i(x)]L}{M_{i,k}(x)}} + \sum_{i=1}^{l} \frac{14L}{3M_{i,k}(x)} \lesssim L \sum_{i=1}^{l} \frac{1}{\sqrt{M_{i,k}(x)}}.
$$

Substituting the above bounds on $b_{k,h}(x)$ and $\beta_k(x)$ at $x = x_{k,h}$ into (B.8) yields

$$
\overline{V}_{k,h}(s) - \underline{V}_{k,h}(s) \lesssim \langle P(x_{k,h}), \overline{V}_{k,h+1} - \underline{V}_{k,h+1} \rangle + \sum_{i=1}^{m} \frac{F}{\sqrt{N_{i,k}(x_{k,h})}} + \sum_{i=1}^{l} \frac{L}{\sqrt{M_{i,k}(x_{k,h})}}.
$$

Inductively, we have

$$
\overline{V}_{k,h}(s) - \underline{V}_{k,h}(s) \lesssim \min \left\{ \sum_{t=h}^{H} \mathbb{E}_{\pi_k} \left[ \sum_{i=1}^{m} \frac{F_1}{\sqrt{N_{i,k}(x_{k,t})}} + \sum_{i=1}^{l} \frac{L}{\sqrt{M_{i,k}(x_{k,t})}} \middle| s_{k,h} = s \right], H \right\}.
$$

$\square$

As noted above, the notion of the good sets in the analysis of F-UCBVI carries over here, which again plays an important role in showing sum-over-time bounds. Analogous to Lemma 17, the following lemma bounds the sum over time of the squared confidence radius, which is later used to bound the cumulative correction term (Lemma 34).

**Lemma 29** (Cumulative confidence radius, Bernstein-style). *Define the lower-order term*

$$
G_1 := m^4 H^4 (\max_i S_i)^2 \max_i X[I_i] L^3 + l^2 H^3 \max_i X[J_i] L^3.
$$

*Then outside the failure event $\mathcal{B}$, for all $i \in [m]$, the sum over time of the following expected squared confidence radius of F-EULER satisfies that*

$$
\begin{aligned}
&\sum_{k=1}^{K} \sum_{h=1}^{H} \sum_{x \in \mathcal{X}} w_{k,h}(x) \left( \mathbb{E}_{P_i} \left( \mathbb{E}_{P_{-i}} [\overline{V}_{k,h+1} - \underline{V}_{k,h+1}] \right)^2 \right) \\
&\leq \sum_{k=1}^{K} \sum_{h=1}^{H} \sum_{x \in \mathcal{X}} w_{k,h}(x) \left( \mathbb{E}_P [(\overline{V}_{k,h+1} - \underline{V}_{k,h+1})^2] \right) \lesssim G_1.
\end{aligned}
$$

*Proof.* Replacing the failure event $\mathcal{F}$ and confidence radius bound (Lemma 12) of F-UCBVI by the failure event $\mathcal{B}$ and confidence radius bound (Lemma 28) of F-EULER, the proof of Lemma 17 carries over here. Note that there is a minor difference in the order of $L$ between the second terms of $G_1$ (F-EULER) and $G_0$ (F-UCBVI), resulting from the $\sqrt{L}$ difference in the corresponding confidence radius bounds. $\square$

Like the analysis of EULER [44], we show two problem-dependent regret bounds of F-EULER. The following lemma bridges one bound to the other.

**Lemma 30** (Bound bridge). *Outside the failure event $\mathcal{B}$, for F-EULER, we have*

$$\sum_{k=1}^{K}\sum_{h=1}^{H}\sum_{x\in L_k} w_{k,h}(x)\frac{g_i(P(x),V_{h+1}^*)-g_i(P(x),V_{h+1}^{\pi_k})}{\sqrt{N_{i,k}(x)}} \leq 2\sqrt{2L}H\sqrt{X[I_i]L}\sqrt{\text{Regret}(K)}.$$

*Proof.* Outside the failure event $\mathcal{B}$, by the properties of $g_i$ (Lemma 24),

$$\sum_{k=1}^{K}\sum_{h=1}^{H}\sum_{x\in L_k} w_{k,h}(x)\frac{g_i(P(x),V_{h+1}^*)-g_i(P(x),V_{h+1}^{\pi_k})}{\sqrt{N_{i,k}(x)}}$$

$$\leq\sqrt{2L}\sum_{k=1}^{K}\sum_{h=1}^{H}\sum_{x\in L_k} w_{k,h}(x)\frac{\|V_{h+1}^*-V_{h+1}^{\pi_k}\|_{2,P(x)}}{\sqrt{N_{i,k}(x)}}$$

$$\leq\sqrt{2L}\sqrt{\sum_{k=1}^{K}\sum_{h=1}^{H}\sum_{x\in L_k}\frac{w_{k,h}(x)}{N_{i,k}(x)}}\cdot\sqrt{\sum_{k=1}^{K}\sum_{h=1}^{H}\sum_{x\in L_k} w_{k,h}(x)\left\langle P(x),(V_{h+1}^*-V_{h+1}^{\pi_k})^2\right\rangle}$$

$$\leq 2\sqrt{2L}\sqrt{X[I_i]L}\cdot\sqrt{H^2\text{Regret}(K)}=2\sqrt{2L}H\sqrt{X[I_i]L}\sqrt{\text{Regret}(K)},$$

where in the second inequality we use the Cauchy-Schwarz inequality, and the third inequality is due to Lemma 15 in this work and Lemma 16 in [44]. $\qquad\square$

### B.3  Bounds on the individual terms in regret

Replacing the failure event $\mathcal{F}$ by $\mathcal{B}$, the regret decomposition in Lemma 18 carries over here. Hence, outside the failure event $\mathcal{B}$,

$\text{Regret}(K)$

$$\leq\sum_{k=1}^{K}\sum_{h=1}^{H}\sum_{x\in L_k} w_{k,h}(x)\left(\underbrace{\left\langle\hat{P}_k(x)-P(x),V_{h+1}^*\right\rangle}_{\text{transition estimation error}}+b_{k,h}(x)+\underbrace{\left\langle\hat{P}_k(x)-P(x),\overline{V}_{k,h+1}-V_{h+1}^*\right\rangle}_{\text{correction term}}\right)$$

$$+\sum_{k=1}^{K}\sum_{h=1}^{H}\sum_{x\in\Lambda_k} w_{k,h}(x)2\beta_k(x)+8H^2\sum_{i=1}^{m}X[I_i]L+8H^2\sum_{i=1}^{l}X[J_i]L,$$

where the $b_{k,h}(x)$ term is referred to as "transition optimism" and $2\beta_k(x)$ term is referred to as "reward estimation error and optimism". In this subsection, we present the bounds on the above individual terms for F-EULER.

**Lemma 31** (Cumulative transition estimation error, Bernstein-style). *Define*

$$\mathbb{C}_i^* := \frac{1}{T}\sum_{k=1}^{K}\sum_{h=1}^{H}\sum_{x\in\mathcal{X}} w_{k,h}(x)g_i^2(P(x),V_{h+1}^*)=\frac{1}{T}\sum_{k=1}^{K}\sum_{h=1}^{H}\mathbb{E}_{\pi_k}[g_i^2(P,V_{h+1}^*)|s_{k,1}],$$

$$\mathbb{C}_i^\pi := \frac{1}{T}\sum_{k=1}^{K}\sum_{h=1}^{H}\sum_{x\in\mathcal{X}} w_{k,h}(x)g_i^2(P(x),V_{h+1}^{\pi_k})=\frac{1}{T}\sum_{k=1}^{K}\sum_{h=1}^{H}\mathbb{E}_{\pi_k}[g_i^2(P,V_{h+1}^{\pi_k})|s_{k,1}],$$

*where $s_{k,1}$ denotes the initial state in the $k$th episode. Then for F-EULER, outside the failure event $\mathcal{B}$, the sum over time of the transition estimation error satisfies that*

$$\sum_{k=1}^{K}\sum_{h=1}^{H}\sum_{x\in L_k} w_{k,h}(x)\left\langle\hat{P}_k(x)-P(x),V_{h+1}^*\right\rangle \lesssim \sum_{i=1}^{m}\sqrt{\mathbb{C}_i^*X[I_i]T}+m^2H\max_i S_i\max_i X[I_i]L^2,$$

*and that*

$$\sum_{k=1}^{K}\sum_{h=1}^{H}\sum_{x\in L_k} w_{k,h}(x)\left\langle\hat{P}_k(x)-P(x),V_{h+1}^*\right\rangle$$

$$\lesssim \sum_{i=1}^{m}\sqrt{\mathbb{C}_i^\pi X[I_i]TL}+\sum_{i=1}^{m}H\sqrt{X[I_i]L}\sqrt{\text{Regret}(K)}+m^2H\max_i S_i\max_i X[I_i]L^2.$$

*Proof.* Outside the failure event $\mathcal{B}$, by lemma 26,

$$\sum_{k=1}^{K}\sum_{h=1}^{H}\sum_{x\in L_k} w_{k,h}(x)\left\langle \hat{P}_k(x) - P(x), V_{h+1}^* \right\rangle$$

$$\leq \sum_{k=1}^{K}\sum_{h=1}^{H}\sum_{x\in L_k} w_{k,h}(x)\left(\sum_{i=1}^{m}\phi_{i,k}(P(x),V_{h+1}^*,x) + \sum_{i=1}^{m}\sum_{j=i+1}^{m} 2HL\sqrt{\frac{S_i S_j}{N_{i,k}(x)N_{j,k}(x)}}\right)$$

$$= \sum_{k=1}^{K}\sum_{h=1}^{H}\sum_{x\in L_k} w_{k,h}(x)\left(\sum_{i=1}^{m}\frac{g_i(P(x),V_{h+1}^*)}{\sqrt{N_{i,k}(x)}}\right. \tag{B.9}$$

$$\left. + \sum_{i=1}^{m}\frac{2HL}{3N_{i,k}(x)} + \sum_{i=1}^{m}\sum_{j=i+1}^{m} 2HL\sqrt{\frac{S_i S_j}{N_{i,k}(x)N_{j,k}(x)}}\right). \tag{B.10}$$

By the Cauchy-Schwarz inequality and Lemma 15, the term (B.9) is bounded by

$$\sum_{k=1}^{K}\sum_{h=1}^{H}\sum_{x\in L_k} w_{k,h}(x)\left(\sum_{i=1}^{m}\frac{g_i(P(x),V_{h+1}^*)}{\sqrt{N_{i,k}(x)}}\right)$$

$$\leq \sum_{i=1}^{m}\sqrt{\sum_{k=1}^{K}\sum_{h=1}^{H}\sum_{x\in L_k} w_{k,h}(x)g_i^2(P(x),V_{h+1}^*)} \cdot \sqrt{\sum_{k=1}^{K}\sum_{h=1}^{H}\sum_{x\in L_k}\frac{w_{k,h}(x)}{N_{i,k}(x)}}$$

$$\leq \sum_{i=1}^{m} 2\sqrt{\mathbb{C}_i^* X[I_i]TL}.$$

By Lemma 15 and Lemma 16, the terms in (B.10) are bounded by

$$\sum_{k=1}^{K}\sum_{h=1}^{H}\sum_{x\in L_k} w_{k,h}(x)\left(\sum_{i=1}^{m}\frac{2HL}{3N_{i,k}(x)} + \sum_{i=1}^{m}\sum_{j=i+1}^{m} 2HL\sqrt{\frac{S_i S_j}{N_{i,k}(x)N_{j,k}(x)}}\right)$$

$$\leq \frac{2}{3}HL\sum_{i=1}^{m} 4X[I_i]L + 2HL\max_i S_i\sum_{i=1}^{m}\sum_{j=i+1}^{m} 4\sqrt{X[I_i]X[I_j]}L$$

$$\lesssim m^2 H\max_i S_i\max_i X[I_i]L^2.$$

Combining the above bounds on (B.9) and (B.10) yields the first bound in this lemma. To show the second bound, we bound (B.9) otherwise, by

$$\sum_{k=1}^{K}\sum_{h=1}^{H}\sum_{x\in L_k} w_{k,h}(x)\left(\sum_{i=1}^{m}\frac{g_i(P(x),V_{h+1}^*)}{\sqrt{N_{i,k}(x)}}\right)$$

$$= \sum_{i=1}^{m}\sum_{k=1}^{K}\sum_{h=1}^{H}\sum_{x\in L_k} w_{k,h}(x)\left(\frac{g_i(P(x),V_{h+1}^{\pi_k})}{\sqrt{N_{i,k}(x)}} + \frac{g_i(P(x),V_{h+1}^*) - g_i(P(x),V_{h+1}^{\pi_k})}{\sqrt{N_{i,k}(x)}}\right)$$

$$\leq \sum_{i=1}^{m} 2\sqrt{\mathbb{C}_i^{\pi} X[I_i]TL} + \sum_{i=1}^{m} 2\sqrt{2L}H\sqrt{X[I_i]L}\sqrt{\mathrm{Regret}(K)},$$

where the inequality is due to the bound bridge (Lemma 30). $\qquad\square$

**Lemma 32** (Cumulative transition optimism, Bernstein-style). *For F-EULER, outside the failure event $\mathcal{B}$, the sum over time of the transition optimism satisfies that*

$$\sum_{k=1}^{K}\sum_{h=1}^{H}\sum_{x\in L_k} w_{k,h}(x)b_{k,h}(x) \lesssim \sum_{i=1}^{m}\sqrt{\mathbb{C}_i^* X[I_i]T} + m^3 H^2\max_i S_i\max_i X[I_i]L^2$$

$$+ m^{1.5}lH^{1.5}(\max_i S_i)^{0.25}(\max_i X[I_i])^{0.75}(\max_i X[J_i])^{0.5}L^2,$$

*and that*

$$\sum_{k=1}^{K}\sum_{h=1}^{H}\sum_{x \in L_k} w_{k,h}(x) b_{k,h}(x) \lesssim \sum_{i=1}^{m} \sqrt{\mathbb{C}_i^{\pi} X[I_i] T L} + \sum_{i=1}^{m} H \sqrt{X[I_i] L} \sqrt{\text{Regret}(K)}$$
$$+ m^3 H^2 \max_i S_i \max_i X[I_i] L^2$$
$$+ m^{1.5} l H^{1.5} (\max_i S_i)^{0.25} (\max_i X[I_i])^{0.75} (\max_i X[J_i])^{0.5} L^2.$$

*Proof.* Outside the failure event $\mathcal{B}$, for all $k \in [K], h \in [H]$ and $x \in \mathcal{X}$, by definition,

$$b_{k,h}(x) = \sum_{i=1}^{m} \frac{g_i(\hat{P}_k(x), \overline{V}_{k,h+1})}{\sqrt{N_{i,k}(x)}} + \sum_{i=1}^{m} \frac{\sqrt{2L} \left\| \overline{V}_{k,h+1} - \underline{V}_{k,h+1} \right\|_{2,\hat{P}_k(x)}}{\sqrt{N_{i,k}(x)}}$$

$$+ \sum_{i=1}^{m} \sum_{j=i+1}^{m} 11HL \sqrt{\frac{S_i S_j}{N_{i,k}(x) N_{j,k}(x)}} + \sum_{i=1}^{m} \frac{5HL}{N_{i,k}(x)}$$

$$= \sum_{i=1}^{m} \phi_{i,k}(\hat{P}_k(x), \overline{V}_{h+1}, x) + \sum_{i=1}^{m} \frac{\sqrt{2L} \left\| \overline{V}_{k,h+1} - \underline{V}_{k,h+1} \right\|_{2,\hat{P}_k(x)}}{\sqrt{N_{i,k}(x)}}$$

$$+ \sum_{i=1}^{m} \sum_{j=i+1}^{m} 11HL \sqrt{\frac{S_i S_j}{N_{i,k}(x) N_{j,k}(x)}} + \sum_{i=1}^{m} \frac{13HL}{3N_{i,k}(x)}$$

$$\leq \sum_{i=1}^{m} \phi_{i,k}(P(x), V_{h+1}^*, x) + \sum_{i=1}^{m} \frac{2\sqrt{2L} \left\| \overline{V}_{k,h+1} - \underline{V}_{k,h+1} \right\|_{2,\hat{P}_k(x)}}{\sqrt{N_{i,k}(x)}}$$

$$+ \sum_{i=1}^{m} \frac{3\sqrt{2}HL}{\sqrt{N_{i,k}(x)}} \sum_{j=1}^{m} \frac{1}{\sqrt{N_{j,k}(x)}} + \sum_{i=1}^{m} \sum_{j=i+1}^{m} 11HL \sqrt{\frac{S_i S_j}{N_{i,k}(x) N_{j,k}(x)}} + \sum_{i=1}^{m} \frac{13HL}{3N_{i,k}(x)}$$

$$\leq \sum_{i=1}^{m} \phi_{i,k}(P(x), V_{h+1}^*, x) + \sum_{i=1}^{m} \frac{2\sqrt{2L} \left\| \overline{V}_{k,h+1} - \underline{V}_{k,h+1} \right\|_{2,\hat{P}_k(x)}}{\sqrt{N_{i,k}(x)}}$$

$$+ \sum_{i=1}^{m} \sum_{j=i+1}^{m} 20HL \sqrt{\frac{S_i S_j}{N_{i,k}(x) N_{j,k}(x)}} + \sum_{i=1}^{m} \frac{9HL}{N_{i,k}(x)}$$

where the first inequality is due to Lemma 25. Substituting the definition of $\phi_{i,k}$ into the above bound on $b_{k,h}(x)$, we have

$$\sum_{k=1}^{K}\sum_{h=1}^{H}\sum_{x \in L_k} w_{k,h}(x) b_{k,h}(x)$$

$$\lesssim \sum_{k=1}^{K}\sum_{h=1}^{H}\sum_{x \in L_k} w_{k,h}(x) \left( \sum_{i=1}^{m} \frac{\sqrt{L} \left\| \overline{V}_{k,h+1} - \underline{V}_{k,h+1} \right\|_{2,\hat{P}_k(x)}}{\sqrt{N_{i,k}(x)}} \right. \tag{B.11}$$

$$\left. + \sum_{i=1}^{m} \frac{g_i(P(x), V_{h+1}^*)}{\sqrt{N_{i,k}(x)}} + \sum_{i=1}^{m} \frac{HL}{N_{i,k}(x)} + \sum_{i=1}^{m} \sum_{j=i+1}^{m} HL \sqrt{\frac{S_i S_j}{N_{i,k}(x) N_{j,k}(x)}} \right). \tag{B.12}$$

By the Cauchy-Schwarz inequality and the inequality that $\sqrt{a+b} \le \sqrt{a} + \sqrt{b}$ for all $a, b \ge 0$, we bound the term in (B.11) by

$$\sum_{k=1}^{K}\sum_{h=1}^{H}\sum_{x \in L_k} w_{k,h}(x)\frac{\left\|\overline{V}_{k,h+1} - \underline{V}_{k,h+1}\right\|_{2,\hat{P}_k(x)}}{\sqrt{N_{i,k}(x)}}$$

$$\le \sqrt{\sum_{k=1}^{K}\sum_{h=1}^{H}\sum_{x \in L_k} \frac{w_{k,h}(x)}{N_{i,k}(x)}} \cdot \sqrt{\sum_{k=1}^{K}\sum_{h=1}^{H}\sum_{x \in L_k} w_{k,h}(x)\left\|\overline{V}_{k,h+1} - \underline{V}_{k,h+1}\right\|_{2,\hat{P}_k(x)}^2}$$

$$\le 2\sqrt{X[I_i]L} \cdot \left( \sqrt{\sum_{k=1}^{K}\sum_{h=1}^{H}\sum_{x \in L_k} w_{k,h}(x)\left\|\overline{V}_{k,h+1} - \underline{V}_{k,h+1}\right\|_{2,P(x)}^2} \right. \tag{B.13}$$

$$\left. + \sqrt{\sum_{k=1}^{K}\sum_{h=1}^{H}\sum_{x \in L_k} w_{k,h}(x)\left\langle \hat{P}_k(x) - P(x), (\overline{V}_{k,h+1} - \underline{V}_{k,h+1})^2 \right\rangle} \right), \tag{B.14}$$

where the term in (B.13) is bounded, due to Lemma 29, by

$$\sum_{k=1}^{K}\sum_{h=1}^{H}\sum_{x \in L_k} w_{k,h}(x)\left\|\overline{V}_{k,h+1} - \underline{V}_{k,h+1}\right\|_{2,P(x)}^2 \lesssim G_1,$$

and the term in (B.14) is bounded, due to Lemma 34, by

$$\sum_{k=1}^{K}\sum_{h=1}^{H}\sum_{x \in L_k} w_{k,h}(x)\left\langle \hat{P}_k(x) - P(x), (\overline{V}_{k,h+1} - \underline{V}_{k,h+1})^2 \right\rangle$$

$$\le H\sum_{k=1}^{K}\sum_{h=1}^{H}\sum_{x \in L_k} w_{k,h}(x)\left|\left\langle \hat{P}_k(x) - P(x), \overline{V}_{k,h+1} - \underline{V}_{k,h+1} \right\rangle\right|$$

$$\lesssim m^3 H^3 (\max_i S_i)^{1.5} \max_i X[I_i]L^{2.5} + mlH^{2.5}(\max_i S_i)^{0.5}(\max_i X[I_i])^{0.5}(\max_i X[J_i])^{0.5}L^{2.5}.$$

By the proof of Lemma 31, the same bounds as those on the cumulative transition estimation error applies to (B.12). Combining the above bounds on (B.11) and (B.12), we obtain the first bound that

$$\sum_{k=1}^{K}\sum_{h=1}^{H}\sum_{x \in L_k} w_{k,h}(x)b_{k,h}(x)$$

$$\lesssim \sum_{i=1}^{m} \sqrt{\mathbb{C}_i^* X[I_i]T} + m^2 H \max_i S_i \max_i X[I_i]L^2 + m\sqrt{\max_i X[I_i]L}\sqrt{G_1}$$

$$\quad + m\sqrt{\max_i X[I_i]L}\sqrt{m^3 H^3 (\max_i S_i)^{1.5} \max_i X[I_i]L^{2.5}}$$

$$\quad + m\sqrt{\max_i X[I_i]L}\sqrt{mlH^{2.5}(\max_i S_i)^{0.5}(\max_i X[I_i])^{0.5}(\max_i X[J_i])^{0.5}L^{2.5}}$$

$$\lesssim \sum_{i=1}^{m} \sqrt{\mathbb{C}_i^* X[I_i]T} + m^2 H \max_i S_i \max_i X[I_i]L^2$$

$$\quad + m(\max_i X[I_i])^{0.5}L^{0.5}m^2 H^2 \max_i S_i(\max_i X[I_i])^{0.5}L^{1.5}$$

$$\quad + m(\max_i X[I_i])^{0.5}L^{0.5}lH^{1.5}(\max_i X[J_i])^{0.5}L^{1.5}$$

$$\quad + m(\max_i X[I_i])^{0.5}L^{0.5}m^{1.5}H^{1.5}(\max_i S_i)^{0.75}(\max_i X[I_i])^{0.5}L^{1.25}$$

$$\quad + m(\max_i X[I_i])^{0.5}L^{0.5}m^{0.5}l^{0.5}H^{1.25}(\max_i S_i)^{0.25}(\max_i X[I_i])^{0.25}(\max_i X[J_i])^{0.25}L^{1.25}$$

$$\lesssim \sum_{i=1}^{m} \sqrt{\mathbb{C}_i^* X[I_i]T} + m^3 H^2 \max_i S_i \max_i X[I_i]L^2$$

$$\quad + m^{1.5}lH^{1.5}(\max_i S_i)^{0.25}(\max_i X[I_i])^{0.75}(\max_i X[J_i])^{0.5}L^2,$$

and the second bound that

$$\sum_{k=1}^{K}\sum_{h=1}^{H}\sum_{x\in L_k} w_{k,h}(x)b_{k,h}(x)$$

$$\lesssim \sum_{i=1}^{m}\sqrt{\mathbb{C}_i^{\pi}X[I_i]T} + \sum_{i=1}^{m}H\sqrt{X[I_i]L}\sqrt{\mathrm{Regret}(K)}$$

$$+ m^3 H^2 \max_i S_i \max_i X[I_i]L^2 + m^{1.5}lH^{1.5}(\max_i S_i)^{0.25}(\max_i X[I_i])^{0.75}(\max_i X[J_i])^{0.5}L^2. \quad \square$$

**Lemma 33** (Cumulative reward estimation error and optimism, Bernstein-style). *For F-EULER, outside the failure event* $\mathcal{B}$*, the sum over time of the reward estimation error and optimism satisfies that*

$$\sum_{k=1}^{K}\sum_{h=1}^{H}\sum_{x\in\Lambda_k} w_{k,h}(x)2\beta_k(x_{k,h}) \lesssim \sum_{i=1}^{l}\sqrt{\mathcal{R}_i X[J_i]T}L + \sum_{i=1}^{l}X[J_i]L^2,$$

*and that*

$$\sum_{k=1}^{K}\sum_{h=1}^{H}\sum_{x\in\Lambda_k} w_{k,h}(x)2\beta_k(x_{k,h}) \lesssim \sum_{i=1}^{l}\sqrt{\mathcal{G}^2 X[J_i]K}L + \sum_{i=1}^{l}X[J_i]L^2.$$

*Proof.* The treatment of the cumulative $\beta_{k,h}(x)$ is essentially the same as the proof of Lemma 8 in [44]. Recall that $\mathcal{R}_i := \max_{x\in\mathcal{X}}\{\mathrm{Var}[r_i(x)]\}$. Outside the failure event $\mathcal{B}$ (specifically, $\mathcal{B}_6$),

$$\sum_{k=1}^{K}\sum_{h=1}^{H}\sum_{x\in\Lambda_k} w_{k,h}(x)\beta_k(x_{k,h})$$

$$= \sum_{k=1}^{K}\sum_{h=1}^{H}\sum_{x\in\Lambda_k} w_{k,h}(x)\sum_{i=1}^{l}\left(\sqrt{\frac{2\mathbb{S}[\hat{r}_i(x)]L}{M_{i,k}(x)}} + \frac{14L}{3M_{i,k}(x)}\right)$$

$$\leq \sum_{k=1}^{K}\sum_{h=1}^{H}\sum_{x\in\Lambda_k} w_{k,h}(x)\sum_{i=1}^{l}\left(\sqrt{\frac{2\mathrm{Var}(r_i(x))L}{M_{i,k}(x)}} + \frac{\sqrt{4L}}{M_{i,k}(x)} + \frac{14L}{3M_{i,k}(x)}\right)$$

$$\lesssim \sqrt{L}\sum_{i=1}^{l}\sqrt{\sum_{k=1}^{K}\sum_{h=1}^{H}\sum_{x\in\Lambda_k}\frac{w_{k,h}(x)}{M_{i,k}(x)}}\sqrt{\sum_{k=1}^{K}\sum_{h=1}^{H}\sum_{x\in\Lambda_k} w_{k,h}(x)\mathrm{Var}(r_i(x))} + X[J_i]L^2 \quad \text{(B.15)}$$

$$\lesssim \sum_{i=1}^{l}\sqrt{\mathcal{R}_i X[J_i]T}L + X[J_i]L^2,$$

where the second inequality is due to the Cauchy-Schwarz inequality, and the last inequality is due to Lemma 15.

Let $s_{k,h} \in \mathcal{S}$ denote the state at step $h$ of episode $k$ and $x_{k,h} = (s_{k,h}, \pi_k(s_{k,h}, h))$. Recall that $\sum_{h=1}^{H} r_i(x_{k,h}) \leq \mathcal{G}$ for any sequence $\{x_{k,h}\}_{h=1}^{H}$. To obtain the second bound, consider

$$\mathrm{Var}\left[\sum_{h=1}^{H} r_i(x_{k,h})\bigg|\{s_{k,h}\}_{h=1}^{H}\right] \leq \mathbb{E}\left[\left(\sum_{h=1}^{H} r_i(x_{k,h})\right)^2\bigg|\{s_{k,h}\}_{h=1}^{H}\right] \leq \mathcal{G}^2.$$

Take the expectation over the trajectory $\{s_{k,h}\}_{h=1}^{H}$ yields

$$\mathbb{E}_{\{s_{k,h}\}_{h=1}^{H}}\left[\mathrm{Var}\left[\sum_{h=1}^{H} r_i(x_{k,h})\bigg|\{s_{k,h}\}_{h=1}^{H}\right]\right] \leq \mathcal{G}^2.$$

Since $r_i(\cdot)$ has independent randomness in different steps, in an alternative notation and taking sum over $k \in [K]$, we have

$$\sum_{k=1}^{K}\sum_{h=1}^{H}\sum_{x\in\Lambda_k} w_{k,h}(x)\mathrm{Var}(r_i(x)) \leq K\mathcal{G}^2.$$

Substituting the above into (B.15) yields

$$\sum_{k=1}^{K}\sum_{h=1}^{H}\sum_{x\in\Lambda_k} w_{k,h}(x)2\beta_k(x_{k,h}) \lesssim \sum_{i=1}^{l}\sqrt{\mathcal{G}^2 X[J_i]K}L + \sum_{i=1}^{l} X[J_i]L^2.$$

$\square$

**Lemma 34** (Cumulative correction term, Bernstein-style). *For F-EULER, outside the failure event $\mathcal{B}$, the sum over time of the correction term satisfies that*

$$\sum_{k=1}^{K}\sum_{h=1}^{H}\sum_{x\in L_k} w_{k,h}(x)\left\langle \hat{P}_k(x) - P(x), \overline{V}_{k,h+1} - V_{h+1}^* \right\rangle$$

$$\leq \sum_{k=1}^{K}\sum_{h=1}^{H}\sum_{x\in L_k} w_{k,h}(x)\left|\left\langle \hat{P}_k(x) - P(x), \overline{V}_{k,h+1} - V_{h+1}^* \right\rangle\right|$$

$$\lesssim m^3 H^2 (\max_i S_i)^{1.5} \max_i X[I_i]L^{2.5} + mlH^{1.5}(\max_i S_i)^{0.5}(\max_i X[I_i])^{0.5}(\max_i X[J_i])^{0.5}L^{2.5}.$$

*Proof.* Replacing the failure event $\mathcal{F}$ by $\mathcal{B}$ and the lower-order term $G_0$ by $G_1$, the proof of the bound on the cumulative correction term for F-UCBVI (Lemma 22) carries over. Note that the second term in the bound here has an extra $\sqrt{L}$ factor due to the difference between $G_0$ and $G_1$. $\square$

## B.4 Regret bounds

The following lemma is useful in deriving the problem-dependent regret bounds of F-EULER.

**Lemma 35** (Component variance bound). *For any finite set $\mathcal{S} = \bigotimes_{i=1}^{m} \mathcal{S}_i$, any probability mass function $P = \prod_{i=1}^{m} P_i \in \Delta(\mathcal{S})$ where $P_i \in \Delta(\mathcal{S}_i)$ for $i \in [m]$, and any vector $V \in \mathbb{R}^{\mathcal{S}}$,*

$$\mathrm{Var}_{P_i}\mathbb{E}_{P_{-i}}[V] \leq \mathrm{Var}_P[V].$$

*Proof.* By the definition of variance,

$$\begin{aligned}\mathrm{Var}_P[V] - \mathrm{Var}_{P_i}\mathbb{E}_{P_{-i}}[V] &= \mathbb{E}_P[V^2] - (\mathbb{E}_P[V])^2 - \mathbb{E}_{P_i}\left[\left(\mathbb{E}_{P_{-i}}[V]\right)^2\right] + (\mathbb{E}_P[V])^2 \\ &= \mathbb{E}_{P_i}\left[\mathrm{Var}_{P_{-i}}[V]\right] \\ &\geq 0. \end{aligned}$$

$\square$

### B.4.1 Proof of Theorem 2

*Proof.* To prove Theorem 2, we actually need to show four combinations of the bounds therein.

Outside the failure event $\mathcal{B}$, combining Lemmas 18, 31, 32, 33 and 34, for F-EULER, we obtain

$$\text{Regret}(K) \lesssim \sum_{i=1}^{m} \sqrt{\mathbb{C}_i^* X[I_i]T} + m^3 H^2 \max_i S_i \max_i X[I_i]L^2$$
$$+ m^{1.5}lH^{1.5}(\max_i S_i)^{0.25}(\max_i X[I_i])^{0.75}(\max_i X[J_i])^{0.5}L^2,$$
$$+ \sum_{i=1}^{l} \sqrt{\mathcal{R}_i X[J_i]T}L + \sum_{i=1}^{l} X[J_i]L^2$$
$$+ m^3 H^2 (\max_i S_i)^{1.5} \max_i X[I_i]L^{2.5}$$
$$+ mlH^{1.5}(\max_i S_i)^{0.5}(\max_i X[I_i])^{0.5}(\max_i X[J_i])^{0.5}L^{2.5}$$
$$+ H^2 \sum_{i=1}^{m} X[I_i]L + H^2 \sum_{i=1}^{l} X[J_i]L$$
$$\lesssim \sum_{i=1}^{m} \sqrt{\mathbb{C}_i^* X[I_i]T} + \sum_{i=1}^{l} \sqrt{\mathcal{R}_i X[J_i]T}L + m^3 H^2 (\max_i S_i)^{1.5} \max_i X[I_i]L^{2.5}$$
$$+ m^{1.5}lH^2(\max_i S_i)^{0.5}(\max_i X[I_i])^{0.75} \max_i X[J_i]L^{2.5},$$

and similarly,

$$\text{Regret}(K) \lesssim \sum_{i=1}^{m} \sqrt{\mathbb{C}_i^* X[I_i]T} + \sum_{i=1}^{l} \sqrt{\mathcal{G}^2 X[J_i]K}L + m^3 H^2 (\max_i S_i)^{1.5} \max_i X[I_i]L^{2.5}$$
$$+ m^{1.5}lH^2(\max_i S_i)^{0.5}(\max_i X[I_i])^{0.75} \max_i X[J_i]L^{2.5}.$$

By definition,

$$\mathbb{C}_i^* = \frac{1}{T} \sum_{k=1}^{K} \sum_{h=1}^{H} \mathbb{E}_{\pi_k}[g_i^2(P, V_{h+1}^*)|s_{k,1}] \le (2\sqrt{L})^2 \mathcal{Q}_i = 4\mathcal{Q}_i L.$$

Substituting the above bound on $\mathbb{C}_i^*$ into the above two regret bounds, we obtain two combinations of the bounds in Theorem 2. Specifically, assuming $T \ge \text{poly}(m, l, \max_i S_i, \max_i X[I_i], H)$, we have

$$\text{Regret}(K) = \tilde{\mathcal{O}}(\sum_{i=1}^{m} \sqrt{\mathcal{Q}_i X[I_i]T} + \sum_{i=1}^{l} \sqrt{\mathcal{R}_i X[J_i]T}),$$
$$\text{Regret}(K) = \tilde{\mathcal{O}}(\sum_{i=1}^{m} \sqrt{\mathcal{Q}_i X[I_i]T} + \sum_{i=1}^{l} \sqrt{\mathcal{G} X[J_i]K}).$$

For other two combinations of the regret bounds of F-EULER, outside the failure event $\mathcal{B}$, combining Lemmas 18, 31, 32, 33 and 34 yet with the alternative bounds in Lemmas 31 and 32, we obtain

$$\text{Regret}(K) \lesssim \sum_{i=1}^{m} \sqrt{\mathbb{C}_i^{\pi} X[I_i]T} + \sum_{i=1}^{m} H\sqrt{X[I_i]L}\sqrt{\text{Regret}(K)} + \sum_{i=1}^{l} \sqrt{\mathcal{R}_i X[J_i]T}L$$
$$+ m^3 H^2 (\max_i S_i)^{1.5} \max_i X[I_i]L^{2.5}$$
$$+ m^{1.5}lH^2(\max_i S_i)^{0.5}(\max_i X[I_i])^{0.75} \max_i X[J_i]L^{2.5}, \tag{B.16}$$

and

$$\text{Regret}(K) \lesssim \sum_{i=1}^{m} \sqrt{\mathbb{C}_i^{\pi} X[I_i]T} + \sum_{i=1}^{m} H\sqrt{X[I_i]L}\sqrt{\text{Regret}(K)} + \sum_{i=1}^{l} \sqrt{\mathcal{G}^2 X[J_i]K}L$$
$$+ m^3 H^2 (\max_i S_i)^{1.5} \max_i X[I_i]L^{2.5}$$
$$+ m^{1.5}lH^2(\max_i S_i)^{0.5}(\max_i X[I_i])^{0.75} \max_i X[J_i]L^{2.5}. \tag{B.17}$$

For inequality $y \lesssim A\sqrt{y} + B$, the solution is

$$\sqrt{y} \lesssim \frac{A + \sqrt{A^2 + 4B}}{2} \lesssim A + \sqrt{B},$$

which yields $y \lesssim A^2 + B + 2A^2\sqrt{B} \lesssim A^2 + B$, where the last inequality is due to the AM-GM inequality. Applying the above solution to (B.16) and (B.17) yields

$$\text{Regret}(K) \lesssim \sum_{i=1}^{m} \sqrt{\mathbb{C}_i^\pi X[I_i]T} + \sum_{i=1}^{l} \sqrt{\mathcal{R}_i X[J_i]T} L + m^3 H^2 (\max_i S_i)^{1.5} \max_i X[I_i] L^{2.5} \tag{B.18}$$

$$+ m^{1.5} l H^2 (\max_i S_i)^{0.5} (\max_i X[I_i])^{0.75} \max_i X[J_i] L^{2.5}. \tag{B.19}$$

and

$$\text{Regret}(K) \lesssim \sum_{i=1}^{m} \sqrt{\mathbb{C}_i^\pi X[I_i]T} + \sum_{i=1}^{l} \sqrt{\mathcal{G}^2 X[J_i]K} L + m^3 H^2 (\max_i S_i)^{1.5} \max_i X[I_i] L^{2.5} \tag{B.20}$$

$$+ m^{1.5} l H^2 (\max_i S_i)^{0.5} (\max_i X[I_i])^{0.75} \max_i X[J_i] L^{2.5}. \tag{B.21}$$

With $x_{k,h} = (s_{k,h}, \pi_k(s_{k,h}, h))$ where $s_{k,h} \in \mathcal{S}$ denotes the state in step $h$ of episode $k$, by definition,

$$\mathbb{C}_i^\pi = \frac{1}{T} \sum_{k=1}^{K} \sum_{h=1}^{H} \sum_{x \in \mathcal{X}} w_{k,h}(x) g_i^2(P(x), V_{h+1}^{\pi_k})$$

$$= \frac{4L}{T} \sum_{k=1}^{K} \sum_{h=1}^{H} \sum_{x \in \mathcal{X}} w_{k,h}(x) \text{Var}_{P_i(x)} \mathbb{E}_{P_{-i}(x)}[V_{h+1}^{\pi_k}]$$

$$\leq \frac{4L}{T} \sum_{k=1}^{K} \sum_{h=1}^{H} \sum_{x \in \mathcal{X}} w_{k,h}(x) \text{Var}_{P(x)}[V_{h+1}^{\pi_k}]$$

$$= \frac{4L}{T} \sum_{k=1}^{K} \sum_{h=1}^{H} \mathbb{E}_{\pi_k} \left[ \text{Var}_{P(x_{k,h})}[V_{h+1}^{\pi_k}|s_{k,h}] \middle| s_{k,1} \right]$$

$$\leq \frac{4L}{T} \sum_{k=1}^{K} \mathbb{E}_{\pi_k} \left[ \left( \sum_{h=1}^{H} R(x_{k,h}) - V_1^{\pi_k}(s_{k,1}) \right)^2 \middle| s_{k,1} \right]$$

$$\leq \frac{4K\mathcal{G}^2 L}{T} = \frac{4\mathcal{G}^2 L}{H},$$

where the first inequality is due to the component variance bound (Lemma 35), the second inequality is due to a law of total variance argument (see Lemma 15 in [44] for a proof). Substituting the above bound on $\mathbb{C}_i^\pi$ into (B.18) and (B.20), we obtain the other two combinations of the bounds in Theorem 2. Specifically, assuming $T \geq \text{poly}(m, l, \max_i S_i, \max_i X[I_i], H)$, we have

$$\text{Regret}(K) = \tilde{\mathcal{O}}(\sum_{i=1}^{m} \sqrt{\mathcal{G}X[I_i]K} + \sum_{i=1}^{l} \sqrt{\mathcal{R}_i X[J_i]T}),$$

$$\text{Regret}(K) = \tilde{\mathcal{O}}(\sum_{i=1}^{m} \sqrt{\mathcal{G}X[I_i]K} + \sum_{i=1}^{l} \sqrt{\mathcal{G}X[J_i]K}).$$

To accommodate the case of known rewards, it suffices to remove the parts related to reward estimation and reward bonuses in both the algorithm and the analysis, which yields the regret bound

$$\min\left\{ \tilde{\mathcal{O}}\left( \sum_{i=1}^{m} \sqrt{\mathcal{Q}_i X[I_i]T} \right), \tilde{\mathcal{O}}\left( \sum_{i=1}^{m} \sqrt{\mathcal{G}X[I_i]K} \right) \right\}.$$

$\square$

### B.4.2 Proof of Corollary 3

*Proof.* Since $r(x) \in [0,1]$ and $R(x) \in [0,1]$ for all $x \in \mathcal{X}$, we have $\mathcal{G} \leq H$ and $\mathcal{R}_i = \max_{x \in \mathcal{X}}[\mathrm{Var}(r_i(x))] \leq 1$ for all $i \in [l]$. Then by Theorem 2, for F-EULER,

$$
\begin{aligned}
\mathrm{Regret}(K) &\lesssim \sum_{i=1}^{m} \sqrt{\mathcal{G}^2 X[I_i] K L} + \sum_{i=1}^{l} \sqrt{\mathcal{R}_i X[J_i] T} L + m^3 H^2 (\max_i S_i)^{1.5} \max_i X[I_i] L^{2.5} \\
&\quad + m^{1.5} l H^2 (\max_i S_i)^{0.5} (\max_i X[I_i])^{0.75} \max_i X[J_i] L^{2.5} \\
&\lesssim \sum_{i=1}^{m} \sqrt{H X[I_i] T L} + \sum_{i=1}^{l} \sqrt{X[J_i] T} L + m^3 H^2 (\max_i S_i)^{1.5} \max_i X[I_i] L^{2.5} \\
&\quad + m^{1.5} l H^2 (\max_i S_i)^{0.5} (\max_i X[I_i])^{0.75} \max_i X[J_i] L^{2.5}.
\end{aligned}
$$

Assuming $T \geq \mathrm{poly}(m, l, \max_i S_i, \max_i X[I_i], H)$, we further have

$$
\mathrm{Regret}(K) = \tilde{\mathcal{O}}\left( \sum_{i=1}^{m} \sqrt{H X[I_i] T} + \sum_{i=1}^{l} \sqrt{X[J_i] T} \right).
$$

In the case of known rewards, again, by removing the parts related to reward estimation and reward bonuses in both the algorithm and the analysis, we obtain the regret bound $\tilde{\mathcal{O}}(\sum_{i=1}^{m} \sqrt{H X[I_i] T})$. □

# C  Lower bounds for FMDPs

## C.1  Proof for degenerate case 1 (Proposition 4)

*Proof.* Without loss of generality, assume $\operatorname{argmax}_i X[{J_i}'] = 1$. Let the reward function depends only on $\mathcal{X}[{J_1}']$. Then for arbitrary transition, learning in this FMDP can be converted to an MAB problem with $X[{J_1}']$ arms, whose regret in $T$ steps has the lower bound $\Omega(\sqrt{X[{J_1}']T})$ [26]. □

## C.2  Proof for degenerate case 2 (Proposition 5)

*Proof.* The construction is essentially the same as that in the proof of Proposition 4. Without loss of generality, assume $\operatorname{argmax}_i X[J_i] = 1$. Let the reward function depends only on $\mathcal{X}[J_1]$. Then for arbitrary transition, learning in this FMDP can be converted to an MAB problem with $X[J_1]$ arms, whose regret in $T$ steps has the lower bound $\Omega(\sqrt{X[J_1]T})$ [26]. □

## C.3  Proof for the nondegenerate case (Theorem 6)

*Proof.* The proof of this lower bound relies on the $\Omega(\sqrt{HSAT})$ regret for nonfactored MDPs. The basic idea to prove the $\Omega(\sqrt{HSAT})$ regret bound in [21] is to construct an MDP with 3 states (an initial state, two states with reward 0 and 1, respectively) and $A$ actions, where the state remains unchanged in the following $H-1$ steps after one step of transition. This MDP is equivalent to an MAB with $A$ arms, which has the lower bound $\Omega((H-1)\sqrt{AK})$. Making $S/3$ copies of this MAB-like MDP and restarting at each copy uniformly at random yield the expected regret lower bound $\Omega(\sqrt{HSAT})$.

Here, without loss of generality, assume $\operatorname{argmax}_i X[I_i] = 1$. Then we only consider the transition of $\mathcal{S}_1$ and neglect the rest. Let the reward function depends only on $\mathcal{S}_1$. Let the transition of $\mathcal{S}_1$ depend only on $\mathcal{S}_1$ and $\mathcal{X}[{I_1}']$, where ${I_1}' = I_1 \cap \{m+1, \cdots, n\}$ contains only the action component indices. Learning in this component can then be converted to learning in a nonfactored MDP with $\mathcal{S}_1$ states and $X[{I_1}']$ actions, which has $\Omega(\sqrt{HS_1X[{I_1}']T})$ regret lower bound.

Now consider the other state components in $\mathcal{X}[I_1]$, which we denote by $\mathcal{S}_1' = \mathcal{X}[I_1 \cap ([m]\backslash\{i\})]$, where "\" denotes set subtraction. Then $X[I_1] = X'[I_1] \cdot S_1' \cdot S_1$. Make $S_1'$ copies of the above FMDP, and restart at each copy uniformly at random, so that each copy is expected to run $T/S_1'$ steps ($K/S_1'$ episodes). The regret lower bound is then given by $\Omega(S_1'\sqrt{HS_1X'[I_1]T/S_1'}) = \Omega(\sqrt{HX[I_1]T})$. □

## C.4  Proof for the general lower bound for the normal factored structure (Theorem 7)

*Proof.* Refer to the number of intermediate state components in an influence loop as its length. For a state component with the loop property, define its minimum influence loop as the one with the minimum length. Without loss of generality, assume $\operatorname{argmax}_{i\in\mathcal{I}} X[I_i] = 1$ and the minimum influence loop of the first state component to be

$$\mathcal{S}_1 \to \mathcal{S}_2 \to \cdots \to \mathcal{S}_{u-1} \to \mathcal{S}_u \to \mathcal{S}_1. \tag{C.1}$$

Since the above influence loop is minimum, we have $i \notin I_1$ for all $i \in \{2, \cdots, u-1\}$. Let $I_{1,s} := I_1 \cap ([m]\backslash\{u\})$ and $I_{1,a} := I_1 \cap \{m+1, \cdots, n\}$ be the state parts (excluding $u$) and action parts of the scope index sets of $\mathcal{S}_1$, respectively. Then $X[I_1] = S_u X[I_{1,s}]X[I_{1,a}]$. Note that in the proof below, we can actually relax the assumption that $S_i \geq 3$ for all $i$ to the assumption that $S_u \geq 3$ and $S_i \geq 2$ for all $i \in [u-1]$. In the case where $\mathcal{S}_1$ has the self-loop property, the proof below carries over by letting $u = 1$.

For the space $\mathcal{S}_i$ in the loop (C.1), we define two special values $s_+^i, s_-^i \in \mathcal{S}_i$ as positive and negative state component values, respectively. Let the reward function be the indicator function of whether at least one of the state components in the loop takes its positive value, i.e., for $x = (s, a)$,

$$R(x) := \max_{i\in[u]}\{\mathbb{I}(s[i] = s_+^i)\}.$$

Construct the transition of $s[1]$ with dependence only on $\mathcal{S}_u$ and $\mathcal{X}[I_{1,a}]$. If $s[u] = s_+^u$ (or $s_-^u$, respectively), then $s[1]$ follows $s[u]$ in the next step and transitions to $s_+^1$ (or $s_-^1$, respectively)

deterministically. Otherwise, $s[1]$ also transitions to $s_+^1$ or $s_-^1$, but with probabilities specified by the action components in $\mathcal{X}[I_{1,a}]$. Construct the transition of the intermediate state components $s[i], i \in \{2, \cdots, u\}$ with dependence only on $s[i-1]$. If $s[i-1] = s_+^{i-1}$, then in the next step $s[i]$ transitions to $s_+^i$; otherwise $s[i]$ transitions to $s_-^i$. The transitions of other state components are arbitrary and irrelevant to the regret. Therefore, after the first step, the positive and negative state component values shift their places in a cyclic way within the influence loop.

For initialization, we choose $s[u]$ to be arbitrary in $\mathcal{S}_u \backslash \{s_+^u, s_-^u\}$, $s[i]$ to be arbitrary in $\mathcal{S}_i \backslash \{s_+^i\}$ for $i \in [u-1]$, and $s[I_{1,s}]$ to be arbitrary in $\mathcal{X}[I_{1,s}]$, where we apply the scope operation (Definition 1) to $s$. The initializations of other state components are arbitrary and irrelevant to the regret. In this way, after the first step of transition, there can be zero or one positive state component values in the influence loop, depending on the action components in $\mathcal{X}[I_{1,a}]$. Moreover, the number of positive state components in the loop remains the same for $H-1$ steps until the end of the episode.

Therefore, for any $s[u] \in \mathcal{S}_u \backslash \{s_+^u, s_-^u\}$ and any $s[I_i] \in \mathcal{X}[I_{1,s}]$, learning in the above FMDP can be converted to an MAB problem with $X[I_{1,a}]$ actions where the reward is $(H-1)$ if $s_+^1$ is reached at the second time step and $0$ otherwise. The regret of such an MAB has the lower bound $\Omega((H-1)\sqrt{X[I_{1,a}]T}) = \Omega(\sqrt{HX[I_{1,a}]T})$ [26]. Splitting $\mathcal{S}_u \backslash \{s_+^u, s_-^u\}$ and $\mathcal{X}[I_{1,s}]$ to make $(S_u - 2)X[I_{1,s}]$ copies of this MAB-like FMDP and restarting at each copy uniformly at random, we obtain the lower bound on regret

$$\Omega\left((S_u - 2)X[I_{1,s}]\sqrt{HX[I_{1,a}]\left\lfloor \frac{T}{(S_u - 2)X[I_{1,s}]}\right\rfloor}\right) = \Omega(\sqrt{HX[I_1]T}).$$

Note that making $S_u/3$ copies is unnecessary since different copies can share $s_+^u$ and $s_-^u$. $\qquad \square$

# D  Lower bound for MDPs via the JAO MDP construction

As pointed out in [1], "they (the $\tilde{\mathcal{O}}(\sqrt{HSAT})$ upper bounds) help to establish the information-theoretic lower bound of reinforcement learning at $\Omega(\sqrt{HSAT})$. ... Moving from this big picture insight to an analytically rigorous bound is non-trivial." A rigorous $\Omega(\sqrt{HSAT})$ lower bound proof is possible [21] by constructing MAB-like MDPs [8]. In the meantime, the MDP literature suggests that the JAO MDP [20], which establishes the minimax lower bound for nonepisodic MDPs, also establishes the minimax lower bound for episodic MDPs. We look into this problem in detail, and find that a direct episodic extension of the JAO MDP actually establishes the lower bound at $\Omega(\sqrt{HSAT/\log T})$, missing a log factor. It is not clear whether this result can be further improved with the same construction. The rest of this section introduces our derivation, which we recommend reading in comparison with [20, Section 6].

Recall that the JAO MDP is an MDP with two states $s_0, s_1$ and $A' = \lfloor (A-1)/2 \rfloor$ actions. The transition probability from $s_1$ to $s_0$ is $\delta$ for all actions, and the transition probability from $s_0$ to $s_1$ is $\delta$ for all actions except that it is $\delta + \epsilon$ for one special action $a^*$. The reward is 1 for each step at $s_1$ and 0 otherwise. By making $S' = \lfloor S/2 \rfloor$ copies of the JAO MDP one extends the construction to $S$ or $(S-1)$ states, which then reduces to a two-state JAO MDP with $S'A'$ actions. In the episodic setting, we simply start the MDP at $s_0$ for each episode. The symbols $\delta, \epsilon, A'$ have the same meanings as they do in [20], and we use $S'$ to replace $k$ in [20]. Let $\mathbb{E}_a, \mathbb{E}_{\text{unif}}$ denote the expectation under $a \in [S'A']$ being the better action $a^*$ and there being no better action respectively, and $\mathbb{P}_a, \mathbb{P}_{\text{unif}}$ denote the corresponding probability measures. Let $\mathbb{E}_*$ denote the expectation under a uniformly random choice of the better action. Hence, $\mathbb{E}_*[f] = \frac{1}{S'A'} \sum_{a=1}^{S'A'} \mathbb{E}_a[f]$. Let $N_1, N_0, N_0^*$ denote the number of visits to state $s_1$, to state $s_0$ and to state-action pair $(s_0, a^*)$ respectively. We use another subscript $k$ to denote the corresponding quantity in the $k$th episode.

**Step 1**  Let $H' = 1/\delta$.

$$\mathbb{E}_a[N_{1,k}] \le \mathbb{E}_a[N_{0,k}] + \epsilon H' \mathbb{E}_a[N_{0,k}^*] - \frac{1}{2\delta} + \frac{(1-2\delta)^H}{2\delta}.$$

*Proof.* Here we adopt a more refined analysis compared to that in [20], where we consider the probability of the last state, because a constant deviation in each episode results in a relaxation on the order of $\mathcal{O}(K)$ in total.

$$
\begin{aligned}
\mathbb{E}_a[N_{1,k}] &= \sum_{h=2}^{H} \mathbb{P}_a(s_{k,h} = s_1 | s_{k,h-1} = s_0) \cdot \mathbb{P}_a(s_{k,h-1} = s_0) \\
&\quad + \sum_{h=2}^{H} \mathbb{P}_a(s_{k,h} = s_1 | s_{k,h-1} = s_1) \cdot \mathbb{P}_a(s_{k,h-1} = s_1) \\
&= \delta \sum_{h=2}^{H} \mathbb{P}_a(s_{k,h-1} = s_0, a_{k,h-1} \ne a) + (\delta + \epsilon) \sum_{h=2}^{H} \mathbb{P}_a(s_{k,h-1} = s_0, a_{k,h-1} = a) \\
&\quad + (1-\delta) \sum_{h=2}^{H} \mathbb{P}_a(s_{k,h-1} = s_1) \\
&= \delta \mathbb{E}_a[N_{0,k} - N_{0,k}^*] + (\delta + \epsilon)\mathbb{E}_a[N_{0,k}^*] + (1-\delta)\mathbb{E}_a[N_{1,k}] - \delta \mathbb{P}_a(s_{k,H} = s_0) - (1-\delta)\mathbb{P}_a(s_{k,H} = s_1).
\end{aligned}
$$

Note that

$$\mathbb{P}_a(s_{k,H} = s_1) \ge \mathbb{P}_{\text{unif}}(s_{k,H} = s_1) = \frac{1}{2} - \frac{1}{2}(1-2\delta)^{H-1}.$$

Then for $\delta < \frac{1}{2}$,

$$
\begin{aligned}
\delta \mathbb{P}_a(s_{k,H} = s_0) + (1-\delta)\mathbb{P}_a(s_{k,H} = s_1) &= \delta(1 - \mathbb{P}_a(s_{k,H} = s_1)) + (1-\delta)\mathbb{P}_a(s_{k,H} = s_1) \\
&\ge \delta(\frac{1}{2} + \frac{1}{2}(1-2\delta)^{H-1}) + (1-\delta)\frac{1}{2} - \frac{1}{2}(1-2\delta)^{H-1} \\
&= \frac{1}{2} - \frac{(1-2\delta)^H}{2}.
\end{aligned}
$$

Hence,

$$\mathbb{E}_a[N_{1,k}] \leq \delta\mathbb{E}_a[N_{0,k} - N_{0,k}^*] + (\delta + \epsilon)\mathbb{E}_a[N_{0,k}^*] + (1-\delta)\mathbb{E}_a[N_{1,k}] - \frac{1}{2} + \frac{(1-2\delta)^H}{2}.$$

By rearranging the terms,

$$\mathbb{E}_a[N_{1,k}] \leq \mathbb{E}_a[N_{0,k}] + \epsilon H'\mathbb{E}_a[N_{0,k}^*] - \frac{1}{2\delta} + \frac{(1-2\delta)^H}{2\delta},$$

where we use $H' = \frac{1}{\delta}$. $\qquad\square$

**Step 2** Let $R$ denote the cumulative reward by a given algorithm through $K$ episodes. Then assuming $\mathbb{E}_a[N_0] \leq \mathbb{E}_{\text{unif}}[N_0]$, by Step 1 we have

$$\mathbb{E}_a[R] = \mathbb{E}_a[N_1] = \sum_{k=1}^{K}\mathbb{E}_a[N_{1,k}] \leq KH - \sum_{k=1}^{K}\mathbb{E}_{\text{unif}}[N_{1,k}] + \epsilon H'\sum_{k=1}^{K}\mathbb{E}_a[N_{0,k}^*] - \frac{K}{2\delta} + K\frac{(1-2\delta)^H}{2\delta}.$$

$$(D.1)$$

**Step 3** Independent of the above two steps,

$$\mathbb{E}_{\text{unif}}[N_{1,k}] = \sum_{t=1}^{H}\mathbb{P}_{\text{unif}}(s_t = s_1) = \sum_{t=1}^{H}\frac{1}{2} + (1-2\delta)^{t-1}(0 - \frac{1}{2}) = \frac{H}{2} - \frac{1-(1-2\delta)^H}{4\delta} \geq \frac{H - H'}{2}.$$

$$(D.2)$$

Therefore,

$$\mathbb{E}_{\text{unif}}[N_{0,k}] \leq \frac{H + H'}{2}. \qquad (D.3)$$

**Step 4** Substituting (D.2) in Step 3 into (D.1) in Step 2 yields

$$\mathbb{E}_a[R] \leq \frac{KH}{2} + \epsilon H'\sum_{k=1}^{K}\mathbb{E}_a[N_{0,k}^*] - \frac{K}{4\delta} + K\frac{(1-2\delta)^H}{4\delta}.$$

Therefore,

$$\mathbb{E}_*[R] = \frac{1}{S'A'}\sum_{a=1}^{S'A'}\mathbb{E}_a[R] \leq \frac{KH}{2} + \frac{\epsilon H'}{S'A'}\sum_{a=1}^{S'A'}\sum_{k=1}^{K}\mathbb{E}_a[N_{0,k}^*] - \frac{K}{4\delta} + K\frac{(1-2\delta)^H}{4\delta}.$$

**Step 5**

$$\mathbb{E}_*[R] \leq \frac{KH}{2} + \frac{\epsilon KH'}{S'A'}\left(\frac{H+H'}{2} + \frac{\epsilon H\sqrt{H'}}{2}\sqrt{S'A'(H'+H)K}\right) - \frac{K}{4\delta} + K\frac{(1-2\delta)^H}{4\delta}.$$

*Proof.* To be more explicit, let $N_{0,k,a}$ denote the number where action $a$ is chosen in state $s_0$ at the $k$th episode. Then by Lemma 13 in [20], we have

$$\sum_{k=1}^{K}\sum_{a=1}^{S'A'}\mathbb{E}_a[N_{0,k}^*] = \sum_{a=1}^{S'A'}\mathbb{E}_a\left[\sum_{k=1}^{K}N_{0,k,a}\right] \leq \sum_{a=1}^{S'A'}\mathbb{E}_{\text{unif}}\left[\sum_{k=1}^{K}N_{0,k,a}\right] + \sum_{a=1}^{S'A'}\epsilon KH\sqrt{H'}\sqrt{2\mathbb{E}_{\text{unif}}\left[\sum_{k=1}^{K}N_{0,k,a}\right]}.$$

Since $\sum_{a=1}^{S'A'}\mathbb{E}_{\text{unif}}[N_{0,k,a}] \leq \frac{H+H'}{2}$ by (D.2) in Step 3,

$$\sum_{k=1}^{K}\sum_{a=1}^{S'A'}\mathbb{E}_a[N_{0,k}^*] \leq K\frac{H+H'}{2} + \frac{\epsilon KH\sqrt{H'}}{2}\sqrt{S'A'K(H+H')}.$$

Substituting the above into the bound on $\mathbb{E}_*[R]$ in Step 4 concludes the proof. $\qquad\square$

**Step 6** Now we determine the optimal value $V_1^*(s_0)$ we aim to compare. Let $V_{H+1}^* = [0,0]^\top$. Then the optimal value function is given by the iteration $V_h^* = B + AV_{h+1}^*$, where

$$A = \begin{bmatrix} 1 - \delta - \epsilon & \delta + \epsilon \\ \delta & 1 - \delta \end{bmatrix}, \quad B = \begin{bmatrix} 0 \\ 1 \end{bmatrix}.$$

Iteratively, by matrix diagonalization,

$$V_1^* = B + AV_2^* = B + A(B + AV_3^*) = \cdots = (\sum_{h=0}^{H-1} A^h)B = U(\sum_{h=0}^{H-1} \Lambda^h)UB,$$

where $U = [1, -\frac{\delta+\epsilon}{\delta}; 1, 1]$ and $\Lambda = \mathrm{diag}(1, 1 - 2\delta - \epsilon)$. Hence,

$$V_1^*(s_0) = \frac{\delta + \epsilon}{2\delta + \epsilon} H - \frac{\delta + \epsilon}{(2\delta + \epsilon)^2} \left(1 - (1 - 2\delta - \epsilon)^H\right).$$

Note that here we compute the exact optimal value because the episodic resetting causes a constant difference than the stationary optimal value in the infinite-horizon setting in each episode, which accumulates to the order of $\mathcal{O}(K)$ in total, similar to Step 1.

**Step 7** Since

$$\frac{\delta + \epsilon}{(2\delta + \epsilon)^2} = \frac{\delta + \epsilon}{4\delta(\delta + \epsilon) + \epsilon^2} \leq \frac{1}{4\delta},$$

we have

$$\mathrm{Regret}(K) = KV_1^*(s_0) - \mathbb{E}_*[R]$$

$$\geq \underbrace{\frac{\delta + \epsilon}{2\delta + \epsilon} KH - \frac{KH}{2} - \frac{\epsilon KH'}{S'A'}\left(\frac{H + H'}{2} + \frac{\epsilon H \sqrt{H'}}{2}\sqrt{S'A'K(H' + H)}\right)}_{\text{standard as in [20]}}$$

$$\underbrace{- K\left(\frac{(\delta + \epsilon)\left(1 - (1 - 2\delta - \epsilon)^H\right)}{(2\delta + \epsilon)^2} - \frac{1}{4\delta} + \frac{(1 - 2\delta)^H}{4\delta}\right)}_{\text{new challenge in the episodic setting}}$$

$$\geq \underbrace{\frac{\epsilon}{4\delta + 2\epsilon}T - \frac{\epsilon TH'}{2S'A'}\left(1 + \frac{H'}{H}\right) - \frac{\epsilon^2 TH'}{2S'A'}\sqrt{H'S'A'KH}\left(\sqrt{1 + \frac{H'}{H}}\right)}_{\Theta(\sqrt{H'S'A'T})}$$

$$\underbrace{- \frac{K}{4\delta}\left((1 - 2\delta)^H - (1 - 2\delta - \epsilon)^H\right)}_{\Theta(KH'(1-\frac{2}{H'})^H)},$$

where $\Theta(\sqrt{H'S'A'T})$ in the last line is obtained by taking $\epsilon = \Theta(\sqrt{\frac{S'A'}{H'T}}) = \Theta(\sqrt{\frac{\delta S'A'}{T}})$. The logic of taking such an $\epsilon$ is to let the first term be on the order of the third term, i.e.,

$$\frac{\epsilon}{4\delta + 2\epsilon}T = \Theta\left(\frac{\epsilon^2 TH'}{2S'A'}\sqrt{H'S'A'KH}\right).$$

The new challenge in the episodic setting results from the episodic resetting of each episode and a different definition of regret. By taking $H = H'\log KH = H'\log T$, we have

$$KH'(1 - \frac{2}{H'})^H \leq KH'\frac{1}{e^{\log KH}} = \frac{H'}{H} = \frac{1}{\log T} = o(\sqrt{H'S'A'T}).$$

Hence, we obtain the $\Omega(\sqrt{HSAT/\log T}) = \tilde{\Omega}(\sqrt{HSAT})$ lower bound. Note that taking $H' = H/\log T$ is reasonable, because none of the parameters is exponential in others in our consideration.

# E    Concentration inequalities

In this section, we provide a summary of some important concentration inequalities that are frequently invoked in this work. In what follows, $\mathbb{P}(\cdot)$ denotes an appropriate probability measure.

**Lemma 36** (Concentration on $L_1$-norm of probability distributions). *Let $P$ be a probability mass function on a finite set $\mathcal{Y}$ with cardinality $Y$. Let $\boldsymbol{y} = [y_1, \cdots, y_n]$ be $n$ i.i.d. samples from $P$. Let $\hat{P}_{\boldsymbol{y}}$ be the empirical distribution based on the observed samples. Then for all $\epsilon > 0$*

$$\mathbb{P}(\|P - \hat{P}_{\boldsymbol{y}}\|_1 \geq \epsilon) \leq 2^Y \exp\left\{-\frac{n\epsilon^2}{2}\right\}.$$

*Alternatively, with probability at least $1 - \delta$,*

$$\|P - \hat{P}_{\boldsymbol{y}}\|_1 \leq \sqrt{\frac{2Y\log 2 + 2\log\frac{1}{\delta}}{n}} \leq \sqrt{\frac{2Y}{n}\log\frac{2}{\delta}}$$

*Proof.* This lemma is a relaxation of Theorem 2.1 in [41].    $\square$

**Lemma 37** (Hoeffding's inequality). *Given $n$ independent random variables such that $x_i \in [a, b]$ a.s., then for all $\epsilon \geq 0$,*

$$\mathbb{P}\left(\left|\sum_{i=1}^{n}(x_i - \mathbb{E}[x_i])\right| \geq \epsilon\right) \leq 2\exp\left\{-\frac{2\epsilon^2}{n(b-a)^2}\right\}.$$

*Alternatively, with probability at least $1 - \delta$,*

$$\left|\frac{1}{n}\sum_{i=1}^{n}(x_i - \mathbb{E}[x_i])\right| \leq \sqrt{\frac{(b-a)^2}{2n}\log\frac{2}{\delta}}.$$

*Proof.* See e.g., Proposition 2.5 in [40] for the one-sided Hoeffding's inequality, applying which to $-x_i$ yields the other side.    $\square$

**Lemma 38** (One-sided Bernstein's inequality). *Given $n$ independent random variables such that $x_i \leq b$ a.s., then for all $\epsilon \geq 0$,*

$$\mathbb{P}\left(\sum_{i=1}^{n}(x_i - \mathbb{E}[x_i]) \geq n\epsilon\right) \leq \exp\left\{-\frac{n\epsilon^2}{2(V + \frac{b\epsilon}{3})}\right\},$$

*where $V = \frac{1}{n}\sum_{i=1}^{n}\mathbb{E}[x_i^2]$. Alternatively, with probability at least $1 - \delta$,*

$$\frac{1}{n}\sum_{i=1}^{n}(x_i - \mathbb{E}[x_i]) \leq \frac{b\log\frac{1}{\delta}}{3n} + \sqrt{(\frac{b\log\frac{1}{\delta}}{3n})^2 + \frac{2V\log\frac{1}{\delta}}{n}} \leq \frac{2b\log\frac{1}{\delta}}{3n} + \sqrt{\frac{2V\log\frac{1}{\delta}}{n}}. \qquad \text{(E.1)}$$

*Proof.* See e.g., Proposition 2.14 in [40].    $\square$

For random variables $x_i \in [0, b]$, applying the one-sided Bernstein's inequality to $x_i - \mathbb{E}[x_i]$, we can replace $V$ in (E.1) by the average variance $\sigma^2 = \frac{1}{n}\sum_{i=1}^{n}\text{Var}(x_i)$. Moreover, applying the one-sided Bernstein's inequality to $-x_i + \mathbb{E}[x_i]$ yields the other side of the bound in terms of $\sigma^2$. By the union bound, we have that with probability at least $1 - \delta$,

$$\left|\frac{1}{n}\sum_{i=1}^{n}(x_i - \mathbb{E}[x_i])\right| \leq \frac{2b\log\frac{1}{\delta}}{3n} + \sqrt{\frac{2\sigma^2\log\frac{1}{\delta}}{n}},$$

which is what we actually use in this work.

**Lemma 39** (Empirical Bernstein's inequality)**.** *Given $n$ independent random variables such that* $x_i \leq 1$ *a.s.. Let* $S = \frac{1}{n(n-1)} \sum_{i=1}^{n} \sum_{j=1}^{n} \frac{(x_i - x_j)^2}{2}$ *be the sample variance. Then with probability at least* $1 - \delta$,

$$\frac{1}{n} \sum_{i=1}^{n} (\mathbb{E}[x_i] - x_i) \leq \sqrt{\frac{2S \log \frac{2}{\delta}}{n}} + \frac{7 \log \frac{2}{\delta}}{3(n-1)}.$$

*Proof.* See Theorem 11 in [27]. $\qquad\square$