[Reviews · NeurIPS 2020]

Review 1

Summary and Contributions: This paper looks at the problem of efficient exploration within factored MDPs where the conditional independence structure is already known. The authors do a good job reviewing the literature in this area, providing insight into the structures and mechanisms in this problem and propose two new algorithms for exploration in this setting. In short, these algorithms extend the recent work in non-factored MDPs (e.g. Azar et al, Zanetter + Brunskill) to factored MDPs.. The main support for this paper comes in the form of theoretical bounds, which seem reasonable according to "standard machinery". However, with a 31 page appendix and dense/technical issues I have to admit i did not verify this during the time pressure of NeurIPS reviews... this should be a significant concern and something that we need to review if all reviewers end in the same boat!

Strengths: There are several things to like about this paper: - It is extremely well-written and easy to follow, even though this is a dense/techincal subject matter. - It seems to elegantly combine (mostly known) tools and techniques to provide results that are genuinely new (if somewhat unsurprising). - The discussions of lower bounds and "degenerate structures" provides some valuable insight into the area that can be confusing... this is probably one of the most clear pieces of work I've seen in this area!

Weaknesses: However, there are some places this paper still has some weaknesses: - The main regret theorems appear to be mostly an exercise in mathematics without too much new insight... it's fine, but I don't think these results are particularly groundbreaking. (I do think the other lower bound results are nice though). - The proofs/appendix are a total of 30+ dense pages of algebra... at some point this feels like we are heading the wrong direction in the field with the "standard machine" of Bernstein-bounds? I think/hope that someone will come along with new tools/insights that can help with something a bit more elegant..

Correctness: Again, I have to admit I did not check this carefully. The general "feel" of the proof and sketches are reasonable, but I hope that someone else has done the hard work of checking this. I don't think it is possible to fully verify these claims without going through the appendix... and even then that could take at absolute minimum a full day of careful inspection.

Clarity: This paper is extremely well written.

Relation to Prior Work: Yes, the relation to prior work and placement of algorithms is top notch!

Reproducibility: Yes

Additional Feedback: Given my comments on review, I am going to say I think this is a good paper, but keep a relatively low confidence. ============ Post rebuttal: I probably stand by that the new insight in the regret is not hugely significant, yes the result per se is new... but similar flavour of results adapted previous UCRL results to factored MDP: https://arxiv.org/abs/1403.3741 So, in this sense I don't find it particularly surprising that a similar result would translate from the UCBVI work: https://arxiv.org/abs/1703.05449 In a sense, this feels more like "filling in the gaps" than truly striking new territory. That said, I maintain my recommendation that this is a good paper and think it deserves acceptance. My worry is that the 30+ pages of appendix could hide a subtle bug... the good news is that because we have a few reviewers here + the general "feel" of the result seems right I am not too worried. I do have a more general feeling that this line of research maybe needs some new breakthrough/insight to really scale things up... That said, great paper, really well written and think it will be great for the conference.


Review 2

Summary and Contributions: The paper analyzes the regret of algorithms in finite-horizon MDPs from a fixed start state, specifically when the transition dynamics are captured by a known graphical model structure. It provides lower bounds and roughly matching upper bounds using a cross-component exploration approach.

Strengths: The paper presents a well-scoped problem and provides state of the art results for this problem.

Weaknesses: The paper describes its principal conceptual advance as follows: "The key algorithmic technique that we needed to develop was a careful design of “cross-component” bonuses to ensure optimism and guide the exploration." But, very little time is spent conveying the insight and the intuitions behind this advance in a way that would help the reader use the idea in other contexts.

Correctness: Claims/methods appear correct.

Clarity: Paper is adequately clear.

Relation to Prior Work: Prior work on analyzing algorithms for finite-horizon MDPs is summarized in a useful way.

Reproducibility: Yes

Additional Feedback: Response to author feedback: From the informal discussion about the cross-component counters, I'm getting that it's somehow bad if different components have been explored unevenly and therefore encouraging more balanced exploration (pairwise) reduces overall variance in the amount of exploration between components. I'm sure there's a lot I'm not getting, but that helps a bit. Regarding "factored MDP", I'm still not buying it. I think it should be the case that you recover an object when you multiply its factors together (for the appropriate definition of "multiply"). There are papers (well, just one I can think of) that deal with truly factored MDPs that are the product of simpler MDPs. They correctly call their MDPs factored. What you (and, I know, many before you) call "factored MDPs" do not have this property. The transition probabilities are indeed factored, as the probability of an outcome can be constructed by multiplying the probabilities of the outcomes of their components. But, that's just part of an MDP, so the MDP object itself is NOT factored. I understand that I'm fighting a losing battle here, but one tries where one can... Original review follows. I object to the term "Factored Markov Decision Processes" because the MDP isn't factored, just its state-space representation is. (Edit: "and transition probabilities".) "develop near-optimal regret bound of FMDPs" -> "develop a near-optimal regret bound for FMDPs". "A key benefit of FMDPs": I think there are a lot of caveats that belong here. FMDPs CAN result in lower representational, computational and sample complexity, but they need not (worst case). I think you need to acknowledge that fact to avoiding leaving in inaccurate impression. "size of the scope of the i-th component": "Scope" is (yet) not defined. "or essentially": Can you be more specific? Is it "equivalently" or "which implies"? Or something else? To be "essentially" seems like "approximately", which isn't a very formal thing to say. "nondegerate" -> "nondegenerate"? Throughout the bibliography capitalization is an issue (mdp, bernstein, more, q, etc.) Please review and fix.


Review 3

Summary and Contributions: The paper presents two algorithms for finite horizon factored MDPs with a rigorous regret analysis. Lower bound results suggest that one of the algorithms has an optimal regret bound and the other has a slightly worse bound.

Strengths: The paper makes strong theoretical contributions to the literature on factored MDPs where both states and actions are factored.

Weaknesses: The work does not consider function approximation and assumes an oracle solver for for the MDPs. It also assumes the factors of the state and reward variables are given.

Correctness: The claims and proofs in the paper appear to be correct.

Clarity: The paper is well written in spite of the technical depth and the notation.

Relation to Prior Work: The prior work is described well and covers the relevant papers.

Reproducibility: Yes

Additional Feedback: Equation 1 and the last line in page 3, shouldn't r_i's be R_i's? Please give an intuitive explanation of why the cross-component bonuses are needed. If you need them, why is it sufficient to stop there and not consider higher order interactions, i.e,., between triples of features. I have read the authors' response. I stand by my review and support the paper. Line 197. I believe you mean "unknown" here. Line 200. What do you mean by the bounds do not assume the knowledge of actual Q_i, R_i and G - since they occur in the bound.


Review 4

Summary and Contributions: This paper introduces two reinforcement algorithms for exploration in factored MDPs. The authors also provide worst case and problem dependant regret bounds for each algorithm. Lower bounds for regret in factored MDPs are also provided in the paper.

Strengths: This work shows that Bernstein-type exploration bonuses can be applied to the setting of factored MDPs which allows to regret bounds for factored MDPs bby a factor \sqrt(H). The author made the effort to provide both worst-case and problem specific bounds for their algorithm. The provided lower bounds are also useful to understand the complexity of factored MDPs.

Weaknesses: My main issue with this work is the lack of novelty. This work simply extends Bernstein type bonuses to factored MDPs and appear incremental. The analysis is not trivial and quite technical and relies mostly on a clever telescoping trick. However the paper does not provide new proof techniques that could be applied in other setting. In its current state I believe the work is incremental and I cannot recommend acceptance. The setting also seems a bit restrictive given that the factorization must be known. At the moment it is not clear if factored MDPs are useful without a lot of prior knowledge about the environment. Post rebuttal update: After reading the author response I agree that the regret bound provided is indeed interesting given that the transition and reward factorizations may differ arbitrarily, this is something I had missed. I updated my score accordingly.

Correctness: I was not able to do a detailed review of the proof in the appendix.

Clarity: The paper is well written and easy to follow. The proof sketch was helpful to understand the analysis of the paper.

Relation to Prior Work: Related work is appropriately discussed.

Reproducibility: Yes

Additional Feedback:

[Author Response · NeurIPS 2020]

We thank the reviewers for their efforts. We will make sure to fix the typos. We address below the main questions.

**Response to Reviewer 1.**  Thank you for the supportive comments!

*1.1 Not too much new insight in the main regret theorems?*  Our analysis shows that the regret of learning in an
FMDP is on the same order as the sum of the regret of learning in each transition component (suppose the reward is
known). This relationship is obvious if the transition and reward factorizations are the same, namely $\mathcal{X}[I_i] = \mathcal{X}[J_i]$
for all $i \in [m]$, in which case the FMDP has $m$ independent components. *The remarkable aspect here is that such*
*a relationship holds, even if the transition and reward factorizations differ arbitrarily.* Moreover, there is additional
insight in the algorithm design to guarantee such theorems, as discussed in our *Response 2.1* below.

**Response to Reviewer 2.**  Thank you for the thoughtful feedback! We address the key points below for space reasons.
*2.1 Lacking in conveying the intuition and insight behind the cross-component bonuses?*  We thank the reviewer for
pointing it out, and are happy to add some of the following discussion in the final paper.

**Intuition.** *An FMDP is not a simple collection of some independent transition components with corresponding reward*
*components*, for which the sum of component-wise bonuses *suffices* to ensure UCB exploration. Instead, for an FMDP,
since the transition and reward factorizations may differ arbitrarily, the naive sum of component-wise bonuses may *not*
*suffice* (discussed around Line 230), and thus we may expect some additional bonuses. A contribution of this paper is
to identify that cross-component bonuses (CCBs) can fulfill the role of the needed additional bonuses.

**Insight.** *The CCBs help stablize early-stage exploration.* Concretely, consider the four phases. **1:** initially all counters
are small, and the CCBs encourage exploration of the $i$th and $j$th components with large $S_i S_j$. **2:** the counters
of some components (say, the $i$th) grow faster such that $N_i(x) \geq S_i$, and the CCBs (say, with the $j$th) satisfy
$HL\sqrt{S_i/N_i(x) \cdot S_j/N_j(x)} \leq HL\sqrt{S_j/N_j(x)}$, which prioritize exploration of underexplored components until all coun-
ters satisfy $N_i(x) \geq S_i$. **3:** the CCBs balance the exploration among different components until all counters satisfy
$\max\{N_i(x), N_j(x)\} \geq S_i S_j$. **4:** the component-wise bonuses dominate the overall bonus and exploration stabilizes.

To summarize the insight, in the long run, different growth rates of the counters reflect different importance of the
components towards maximizing cumulative rewards, and early on, their growth can suffer large variance. In this
sense, the CCBs play a role of *variance reduction*. The intuition and insight here may also be helpful in future
research on FMDPs, e.g., the function approximation setting and the structure-agnostic setting.

*2.2 Is the term "factored MDP" proper?*  Although the term "factored MDP" is not our invention, we think that it is
indeed accurate. In a factored MDP, in addition to the state space, the state-action space, the transition and the reward
can all be factored. In fact, *the transition factorization turns out to be essential in reducing the regret* compared with
a nonfactored MDP, since the regret depends on $X[I_i]$, the size of the scope of a transition component.

**Response to Reviewer 3.**  Thank you for the kind and positive comments!

*3.1 Not considering function approximation setting?*  Even for nonfactored MDPs, the regret bounds in the function
approximation setting are only established under stringent assumptions, e.g., linear MDPs. Our work can be seen as a
concrete first step towards understanding more general settings.

*3.2 Assuming an oracle solver for FMDPs?*  Actually, neither F-UCBVI nor F-EULER assumes an oracle solver. The
solver, based on value iteration, is explicitly described in Algorithm 2.

*3.3 Other issues?*  **Intuition:** please see our *Response 2.1* for an intuitive explanation regarding why we need the
cross-component bonuses. These bonuses suffice to guarantee minimax optimal regret up to log factors, though it is
possible that higher-order interactions could provide other benefits. **Line 200:** what we meant to say is that while the
F-EULER algorithm does not assume the knowledge of $\mathcal{Q}_i, \mathcal{R}_i, \mathcal{G}$, its regret guarantees automatically adapt to these
quantities. We shall rephrase our expression in the final paper.

**Response to Reviewer 4.**  Thank you for the constructive comments!

*4.1 Lacking novelty?*  We would like to emphasize three noteworthy contributions of this paper. 1) As mentioned in our
*Response 1.1*, our main regret theorems show that the regret of learning in an FMDP is on the same order of the sum
of those in each transition component. This clean result, however, is nontrivial and even surprising, since the transition
and reward factorizations may differ arbitrarily. 2) We identify a correct form (cross-component) of bonuses that
are novel and *highlight the key difference towards achieving UCB exploration compared with the nonfactored setting*.
Moreover, these cross-component bonuses offer *new insight* (see our *Response 2.1*). The insight and techniques we
develop might also be helpful for future research on FMDPs. 3) We are *the first to unveil the intricacy of proving lower*
*bounds* in the factored setting and *to establish nontrivial lower bounds* for some fairly general factored structures.

*4.2 Assuming known factorization?*  Although in some practical problems the factorization is indeed known (see [12]
for an example), we admit that this assumption can be a bit restrictive. On the other hand, the primary aim of this
paper is to promote our understanding about the fundamental theoretic limits, and in this sense, the result here can be
seen as a solid first step towards our understanding about the more general structure-agnostic setting.

[Meta-Review · NeurIPS 2020]

While this paper initially had some mild divergence of opinion among the reviewers, after the author response and some detailed discussion, it was agreed that this paper makes a solid contribution (please see the revised reviews). It is certainly is of relevance to NeuRIPS. After discussion, there was agreement on the significance of the conceptual contribution, namely the treatment of the cross-component bonuses. Several reviewers note that the mathematics is fairly “standard” (Bernstein-bound machinery), though in the end that should not be considered a drawback. At least one reviewer notes that the 31pp appendix means that it is not possible to verify the mathematical results during the review period. (That said, I’m not sure that much can be done to address this is in the revised paper!) The reviews, including their initial pre-rebuttal critique, suggest several ways on which the paper can be improved to maximize its impact. The author response did a good job of conveying the significance of the results, and the author(s) is/are strongly encouraged to revise the paper to ensure these points are not lost on future readers. Please incorporate other reviewer suggestions as appropriate also. Solid contribution!